# Age-related epithelial defects limit thymic function and regeneration

The thymus is essential for establishing adaptive immunity yet undergoes age-related involution that leads to compromised immune responsiveness. The thymus is also extremely sensitive to acute insult and although capable of regeneration, this capacity declines with age for unknown reasons. We applied single-cell and spatial transcriptomics, lineage-tracing and advanced imaging to define age-related changes in nonhematopoietic stromal cells and discovered the emergence of two atypical thymic epithelial cell (TEC) states. These age-associated TECs (aaTECs) formed high-density peri-medullary epithelial clusters that were devoid of thymocytes; an accretion of nonproductive thymic tissue that worsened with age, exhibited features of epithelial-to-mesenchymal transition and was associated with downregulation of FOXN1. Interaction analysis revealed that the emergence of aaTECs drew tonic signals from other functional TEC populations at baseline acting as a sink for TEC growth factors. Following acute injury, aaTECs expanded substantially, further perturbing trophic regeneration pathways and correlating with defective repair of the involuted thymus. These findings therefore define a unique feature of thymic involution linked to immune aging and could have implications for developing immune-boosting therapies in older individuals.

Thymic T cell differentiation requires the close interaction between thymocytes and the supporting stromal microenvironment, which is composed of highly specialized TECs, endothelial cells (ECs), mesenchymal cells like fibroblasts (FBs), dendritic cells, innate lymphoid cells and macrophages[1]; however, thymic function is not static over lifespan with a well-described decline in function that accelerates upon puberty and is characterized by tissue atrophy, disrupted stromal architecture, reduced export of new naive T cells and, ultimately, diminished responsiveness to new antigens[2]. In addition to its chronic functional decline with age, the thymus is also extremely sensitive to acute damage induced by routine insults such as stress and infection, but also more severe damage such as that caused by common cancer therapies such as cytoreductive chemo- or radiation therapy[2,3]. Although the thymus harbors tremendous capacity for endogenous repair, this regenerative capacity also declines with age[2,3]. This deficiency manifests most prominently after the thymic damage caused by the conditioning regimens required for hematopoietic cell transplantation (HCT), which

leads to prolonged T cell lymphopenia, an important contributor to transplant-related morbidity and mortality due to infections and malignant relapse[2,3]. In fact, both pre- and post-transplant thymic function can be positive prognostic indicators of HCT outcomes[4]. Recent advances in single-cell technology have provided new insights into the heterogeneity of TECs in young and aged mice and in humans[5–20]; how TECs orchestrate T cell differentiation and how dysfunction in these processes is linked to autoimmunity and immunodeficiency; however, perhaps as a consequence of this complexity, the mechanisms underlying thymic involution and regeneration after damage remain poorly understood[2,3].

Using single-cell and spatial transcriptomics, lineage-tracing and advanced imaging, we report here the age-associated emergence of unique TEC states linked with tissue degeneration. aaTECs formed atypical high-density epithelial clusters that were devoid of thymocytes; an accretion of nonfunctional thymic tissue that worsened with age and exhibited features of partial epithelial-to-mesenchymal

e-mail: dgray@wehi.edu.au; mvandenbrink@coh.org; jdudakov@fredhutch.org

transition (EMT). The accumulation of aaTECs in the involuted thymus was exacerbated by acute injury and was associated with diminished regenerative capacity compared to young mice. We found evidence that aaTECs drew tonic signals away from other TECs, acting as a 'sink' for epithelial regeneration cues such as FGF and BMP signaling. These structural and functional changes to the thymic epithelium could be linked to molecular changes in the fibroblast compartment and specifically, their age-related upregulation of programs associated with inflammaging. These data define a key feature of the involuted thymus that limits organ function and restricts regenerative capacity after acute injury; findings that may have important implications for the development of therapeutic strategies to improve thymic function in aged individuals.

## Emergence of atypical epithelial cell populations with age

Consistent with its well-described decline in function over lifespan[2,3,21,22], we found that total thymic cellularity declined in female mice across 18 months of age (Extended Data Fig. 1a) coincident with morphological changes, such as a relative decrease in the cortical-to-medullary ratio (Extended Data Fig. 1b,c). Quantification of the major structural cell lineages (TECs, ECs and FBs) by flow cytometry revealed little alteration in ECs or FBs, but a diminished TEC compartment that mirrors the overall loss of thymic cellularity, with a more severe loss in medullary TECs (mTECs) compared to cortical TECs (cTECs) despite the observed cortical thinning (Extended Data Fig. 1b–d).

Recent reports have resolved the remarkable heterogeneity of stromal subsets[5–7,13,20,23,24]. To investigate the stromal changes in the thymic microenvironment associated with age-related thymic involution, we performed single-cell sequencing of nonhematopoietic stromal cells from 2-month-old (2-mo) or 18-month-old (18-mo) female mice (22,932 CD45− cells). To define various cell compartments and subsets, gene signatures of published thymic single-cell sequencing datasets were mapped to our combined 2-mo and 18-mo steady-state dataset (Fig. 1a, Extended Data Fig. 2 and Supplementary Fig. 1)[5–8,13]. We also integrated this with all published thymic sequencing datasets and generated a tool called ThymoSight (www.thymosight.org) that allows for their interrogation (Fig. 1b and Extended Data Fig. 3a)[5–18].

Primary stromal cell lineages were defined based on transcription of lineage-specific genes: TECs with *Epcam*, *H2-Aa*; ECs with *Pecam1*, *Cdh5*; FBs with *Pdgfra*, alongside less abundant stromal cell types, including mesothelial cells (MECs) (*Upk3b* and *Nkain4*); vascular smooth muscle cells (vSMCs) (*Acta2* and *Myl9*); pericytes (PCs) (*Myl9* and *Acta2*) (Fig. 1a and Extended Data Fig. 2a); and extremely rare nonmyelinating Schwann cells (nmSCs) (*Gfap*, *Ngfr* (p75) and *S100b*)[25]. To more precisely define the heterogeneity of the major CD45− structural compartments (epithelial, endothelial and fibroblast), we then subsampled and reanalyzed each major stromal cell population separately and used public subset signatures (Supplementary Table 1) provided collectively within ThymoSight to precisely define compartment heterogeneity. Using this approach, we annotated steady-state clusters within each cell lineage with respect to publicly available datasets (Fig. 1c,d and Extended Data Fig. 2b–d). Unsupervised clustering analysis distinguished the *Pdgfra*-expressing mesenchyme into three main groups, two of which were consistent with mouse capsular or human interlobular FB signatures (capsFB; including but not limited to expression of *Dpp4*, *Smpd3* and *Pi16*) and mouse medullary or human perilobular FBs (medFB; *Ptn* and *Postn*)[9,23,26] (Fig. 1c,d, Extended Data Fig. 2b and Supplementary Table 1). We also identified an intermediary subset of FB (intFB) marked by *Inmt* and *Gpx3*, which did not map to the public gene signatures (Fig. 1c,d and Extended Data Fig. 2b).

Using gene signatures from an organ-wide murine EC atlas[27], within the endothelium three main clusters were identified that could be mapped to arterial (aEC), capillary (capEC) or venous EC (vEC) (Fig. 1c,d, Extended Data Fig. 2c and Supplementary Table 1). Less than 1% of

all ECs showed enrichment for lymphatic markers (Supplementary Fig. 3c), consistent with the scarcity of lymphatic ECs observed in sections or flow cytometric analysis of thymi from adult mice[28]. Expression of *Plvap* and *Cldn5* delineated capEC[29], whereas vECs demonstrated the highest expression of *Vwf* and *Vcam1* (Fig. 1c,d, Extended Data Fig. 2c and Supplementary Table 1), genes that correlate with the vessel diameter. Venous ECs expressed high levels of P-selectin (*Selp*) (Fig. 1c,d, Extended Data Fig. 2c and Supplementary Table 1), indicative of thymic portal ECs (TPECs), which have been linked to homing of hematopoietic progenitors in the thymus[30]. Notably, *Bmp4*, which is produced by thymic ECs and is involved in regeneration after acute insult[31], was expressed highest by vECs (Fig. 1c,d, Extended Data Fig. 2c and Supplementary Table 1).

Finally, TEC clusters were annotated based on the nomenclature and signatures derived from previously reported studies[5–8,10–18] and mapped to ten subsets (Fig. 1c,d, Extended Data Fig. 2d and Supplementary Table 1): cTEC (based on, among others, expression of *Prss16*, *Psmb11* and *Ly75*), mTEC1 (*Ccl21a*, *Itgb4* and *Ly6a*), a proliferating mTEC subset (mTEC^prol; *Ccnd2*) and mTEC2 (*Aire*). A distinct subset with similarities to mTEC1 cells could also be identified that exhibited a gene signature consistent with a recently identified 'early' TEC progenitor (early^prog)[8]. We could also identify recently described mimetic populations[13,14] with basal (TEC^basal, *Ly6d* and *Spink5*), neural (TEC^neuro and *Car8*), tuft (TEC^tuft; *Avil* and *L1cam*), goblet (TEC^goblet; *Spink5* and *Wfdc2*) and microfold (M)-like cells of the small intestine (TEC^M-cell; *Ccl20*) (Fig. 1c,d, Extended Data Fig. 2d and Supplementary Table 1).

Comparison of the populations from young (2-mo) and aged (18-mo) mice revealed considerable overlay, with no major new cell populations emerging with age in the endothelial or fibroblast compartments, although transcriptional differences were observed within existing cell populations suggesting an age-associated change in cell state rather than cell type (Fig. 1e and Supplementary Table 2). In contrast, within the TEC compartment we observed two distinct age-associated epithelial cell types, referred henceforth as aaTEC1 and aaTEC2 (Fig. 1e). These were apparent only in samples from 18-mo mice and mapped to only one previously published dataset[12], likely due to the age and cell selection protocols used (Extended Data Fig. 3b). Assessing quantitative changes in cell subsets with age, we found that the two aaTEC populations exhibited the greatest changes with age, but we also observed a considerable increase in early^prog (Fig. 1f). Using our sequencing dataset as well as previous publications to design flow cytometry panels to examine these stromal populations[6], we found little change in the frequencies of EC subsets or their number with age, although there were increased numbers of medFBs, PCs and vSMCs (Extended Data Fig. 4a–c). This age-related increase in mesenchymal-lineage cells may reflect that observed in other tissues where fibrosis is a key driver of functional decline[32]. Within the epithelial compartment, we observed decreases in all mTECs, including TEC^tuft cells, but in contrast we observed a marked increase of an atypical TEC population with age that expressed EpCAM but was negative for Ly51 and UEA1 (features of cTECs or mTECs, respectively) (Fig. 1g), consistent with the loss of mTEC- or cTEC-specific markers within the aaTEC1 subset. This population of Ly51/UEA1 double negative (DN)-TECs gradually emerges and expands across lifespan (Fig. 1h and Extended Data Fig. 4d). Differential expression analysis between each aaTEC subset versus all other TECs identified claudin-3 (*Cldn3*) and podoplanin (*Pdpn*) as potentially distinctive, in conjunction with established markers (Fig. 1i and Extended Data Fig. 4e). Flow cytometric analysis confirmed their utility, resolving populations of both aaTEC1 (Epcam+MHCII+Ly51−UEA1−Cldn3+) and aaTEC2 (Epcam−MHCII+PDGFRa−Pdpn+) subsets that increased with age (Fig. 1j,k and Extended Data Fig. 4f). Overall, these data reveal that the most prominent shift among stromal cell types during thymic involution was the emergence of two unique epithelial cell types that lack typical TEC features.

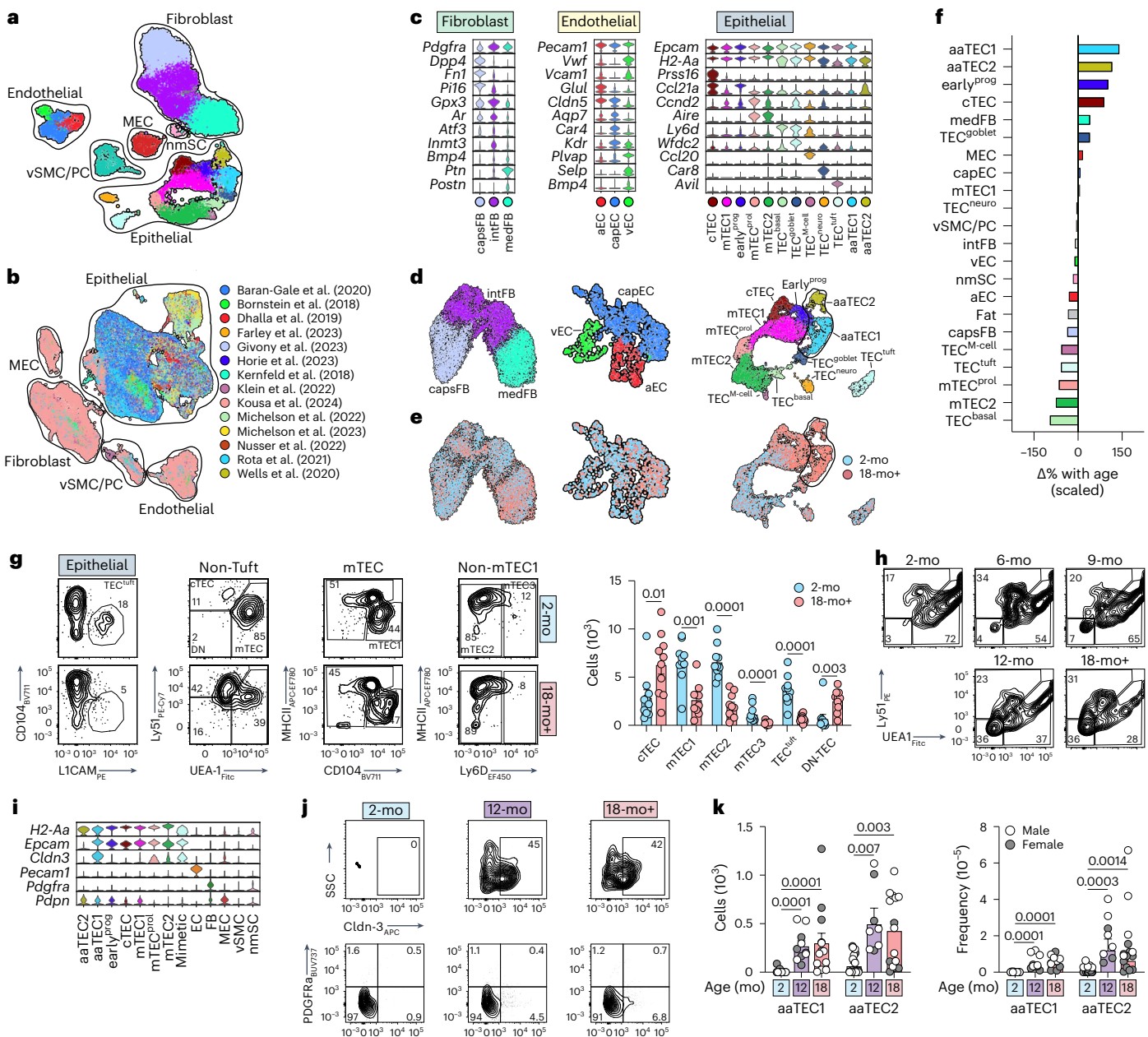

**Fig. 1 | Emergence of atypical epithelial populations with age. a**, Uniform Manifold Approximation and Projection (UMAP) of 22,932 CD45⁻ thymic cells from 2-mo and 18-mo female C57BL/6 mice, annotated by cell type subset and outlined by cell compartment (epithelial; fibroblast; endothelial; MEC; vSMC/PC; nmSC). **b**, ThymoSight integration of public data for murine nonhematopoietic thymic stromal cells, including our own dataset (*n* = 297,988) annotated by publication source and outlined by cell type and compartment. **c**, Violin plots highlighting key genes marking individual subsets within individual structural compartments (fibroblast, endothelium and epithelium). **d,e**, UMAPs of individual structural compartments color-coded by cell type subset (**d**) and age cohort (**e**). *n*ᴇᴄ = 1,661; *n*ꜰʙ = 13,240; *n*ᴛᴇᴄ = 6,175. **f**, Scaled change in frequency for each individual structural cell subset with age. **g**, Gating strategy and quantities for cell populations within the epithelial lineage (based on previous work[6]) in 2-mo (*n* = 10) and 18-mo (*n* = 10) mice. First, based on a CD45⁻EpCAM⁺ parent

gate, tuft cells were identified by expression of L1CAM, then all other TECs were assessed for expression of conventional TEC markers UEA1 and Ly51. Within the UEA1ʰⁱLy51ˡᵒ mTEC population CD104⁺MHCIIˡᵒ cells were identified as mTEC1. Cells that were deemed as non-mTEC1 were then fractionated based on MHCII and Ly6D. **h**, Concatenated flow cytometry plots and graphs highlighting the frequency of Ly51⁻UEA1⁻ (DN-TECs) across lifespan (gated on CD45⁻EpCAM⁺MHCII⁺ cells). **i**, Violin plots of aaTEC1 and aaTEC2 novel markers. **j,k**, Flow cytometry plots (**j**) and quantities (**k**) for aaTEC1 and aaTEC2 populations in 2-mo (*n* = 15), 12-mo (*n* = 10) or 18-mo (*n* = 13) male and female C57BL/6 mice. Summary data represent mean ± s.e.m.; each dot represents an individual biological replicate. Statistics were generated using a two-tailed Mann–Whitney test comparing within individual subsets (**g**) or Kruskal–Wallis (**k**) test with Dunn's correction.

## aaTECs form high-density clusters of epithelial cells

To study epithelial structures in the involuted thymus, we created a reporter strain, *Foxn1*^Cre × *Rosa26*^nTnG (*Foxn1*^nTnG), where all cells express a nuclear-localized tdTomato reporter, except those that have activated *Foxn1* (a master transcription factor in TEC[33]), which instead, express a nuclear-localized green fluorescent protein (GFP) (Extended Data Fig. 5a). The nuclear localization of the *Foxn1*^nTnG reporter enabled

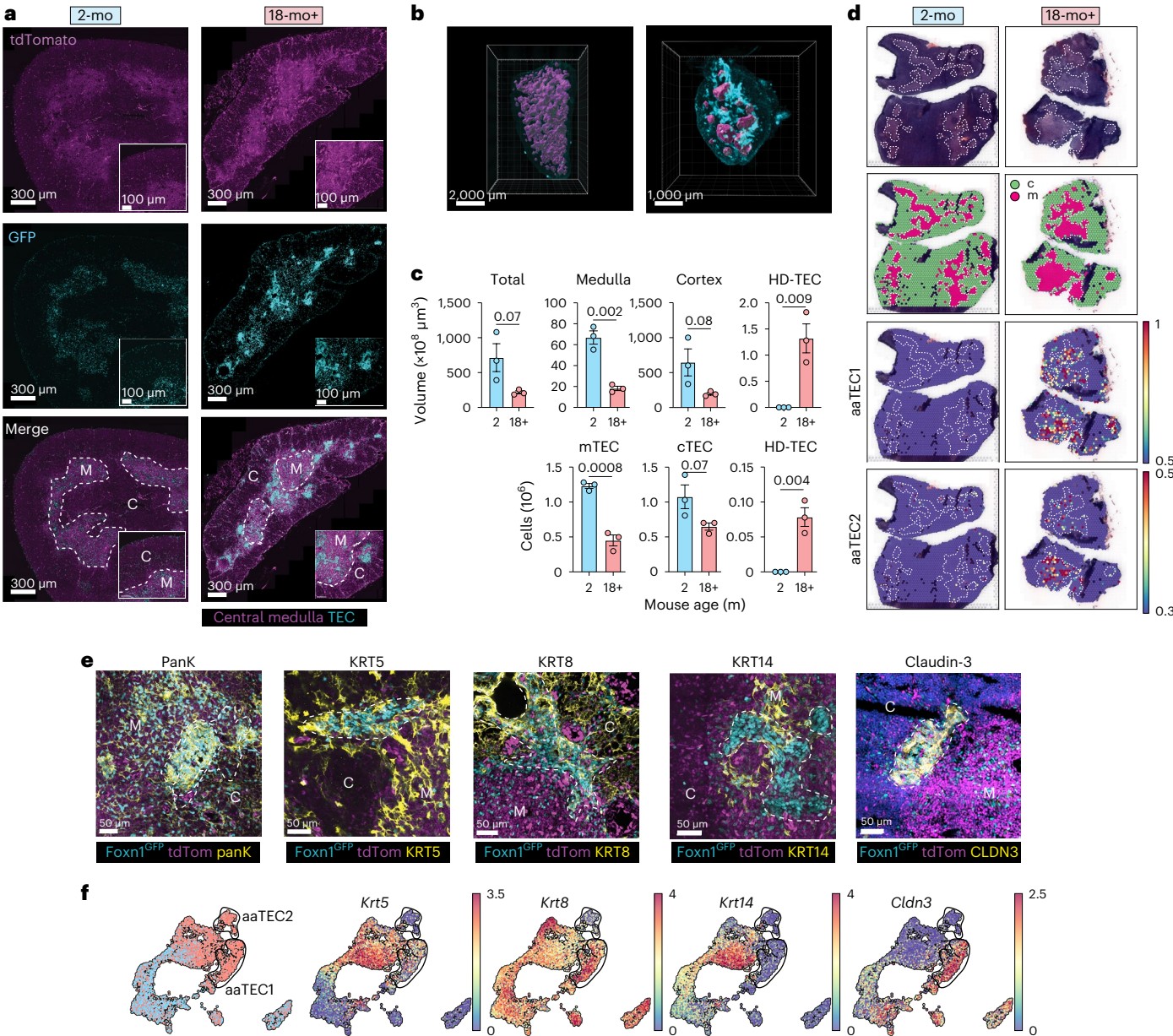

**Fig. 2 | Age-associated TECs form distinct high-density peri-medullary 'scars'.**
**a**, Representative images of thymus in 2-mo and 24-mo male *Foxn1*^nTnG mice with medulla marked by dotted line. HD-TEC regions are apparent in aged but not young mice. **b**, Images of whole-tissue light-sheet imaging showing central medulla (magenta) in 2-mo and 24-mo male *Foxn1*^nTnG mice with HD-TEC (cyan) only apparent in aged mice. Medulla surface was defined on the basis of the frequency of GFP⁺ cells and tdTomato expression (with high tdTom correlating with medullary regions) and confirmed by confocal imaging of KRT14. **c**, Quantification of total thymus volume, volume of cortex, medulla and HD-TEC regions and the number of mTECs, cTECs and HD-TECs calculated from whole-tissue and confocal imaging in 2-mo (*n* = 3) and 18-mo (*n* = 3) mice. Summary data

represent the mean ± s.e.m.; statistics were derived from independent biological replicates (individual animals) using a two-tailed unpaired *t*-test. **d**, Visium spatial sequencing performed on 2-mo or 18-mo C57BL/6 thymus. Displayed are H&E sections, cortex and medulla identified by Leiden clustering and heatmaps of aaTEC1 and aaTEC2 signatures (top 20 differentially expressed genes for each subset versus other TECs; Extended Data Fig. 3 and Supplementary Table 3) overlaid. **e**, Expression of pan-keratin and keratin subunits 8, 5, 14 and claudin-3 on HD-TEC regions of thymus from 12–18-mo male *Foxn1*^nTnG mice. Scale bar, 50 μm. **f**, Transcriptional expression of keratin subunits and claudin-3 in the epithelial scRNA-seq dataset. *n*_TEC = 6,175.

cellular resolution of TEC by light-sheet imaging of whole cleared thymic lobes from young and old mice. As expected, we observed a relatively low density of GFP⁺ cells in the thymic cortex compared to the subcapsular and medullary regions (Fig. 2a). At the whole-tissue level, the medulla formed a highly complex and interconnected structure in the young thymus that degenerated into isolated islets upon involution (Fig. 2b and Supplementary Video 1a,b). Another notable feature of the involuted thymus, entirely absent in the young, was the emergence of zones of very high-density GFP⁺ TEC (HD-TEC) clusters

(Fig. 2a,b). These HD-TEC clusters formed band-like structures that were associated with the medulla of the involuted thymus (Fig. 2b, Extended Data Fig. 5b,c and Supplementary Video 1b). Although the volume of cortex and medulla, and the number of cTECs and mTECs (calculated from whole-tissue imaging) declined with age, HD-TEC clusters emerged to comprise a substantial volume and number of TECs (Fig. 2c).

Spatial transcriptomics using Visium comparing thymi from 2-mo to 18-mo mice demonstrated aaTEC1 and aaTEC2 signatures (generated from the 20 genes most differentially expressed within each subset

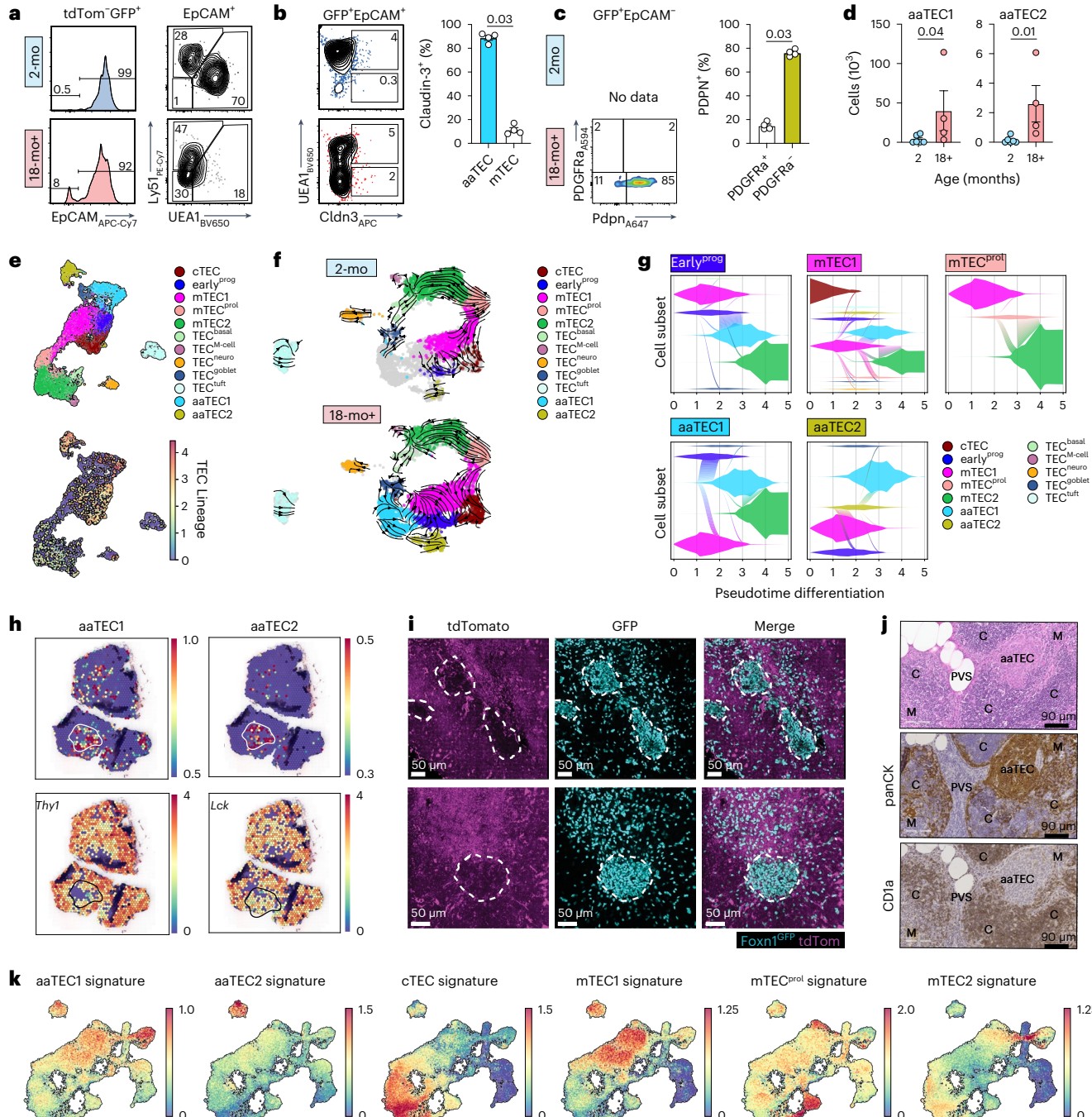

**Fig. 3 | Age-associated TECs are derived from FOXN1-expressing cells.**
**a**, Representative flow cytometry plots from 12–18-mo male and female *Foxn1*[nTnG] mice at the indicated ages, gated on tdTom⁻GFP⁺ cells. tdTom⁻GFP⁺EpCAM⁺ cells were then assessed for expression of the conventional TEC markers UEA1 and Ly51. **b**, Claudin-3 and UEA1 expression on tdTom⁻GFP⁺EpCAM⁺ cells in *Foxn1*[nTnG] mice, and quantification of claudin-3 on UEA1⁺ mTEC and UEA1⁻Ly51⁻ DN-TECs (*n* = 4 biological replicates representing individual mice). **c**, Podoplanin (Pdpn) and PDGFRa expression on tdTom⁻GFP⁺EpCAM⁻ cells (*n* = 4 biological replicates representing individual mice). **d**, Number of GFP⁺EpCAM⁺UEA1⁻Ly51⁻Cldn3⁺ aaTEC1 and GFP⁺EpCAM⁻PDPN⁺PDGFRα⁻ aaTEC2 cells in 2-mo (*n* = 6) and 18-mo (*n* = 4) mice. **e**, scRNA-seq was performed on CD45⁻ cells isolated from male and female 20-mo *Foxn1*[tdTom] and age-matched WT mice, and integrated into the epithelial data described in Fig. 1c–e. UMAP of 8,505 cells of the epithelial compartment in the integrated data showing the TEC annotated subsets (top) and overlaid expression of tdTomato (bottom). Scale represents log-transformed average expression of the tdTomato-WPRE element. **f**, RNA velocity on selected TEC populations in 2-mo (top) or 18-mo (bottom) mice. $n_{2mo}$ = 1,989; $n_{18mo}$ = 3,382.

**g**, Vein plots describing the continuous transition of 18-mo early[prog], mTEC1, mTEC[prol] and aaTEC subsets to their predicted descendants (represented by diagonal flows) and the dynamic relative frequencies (vein width on the *y* axis) of these TEC subsets in the thymus over the binned pseudotime. **h**, Expression of thymocyte markers *Thy1* and *Lck* overlaid on the 18-mo spatial transcriptomics dataset. Outline represents thymocyte-poor area overlaid onto heatmap showing aaTEC1 or aaTEC2 signatures. **i**, Two representative images in 12–18-mo male and female *Foxn1*[nTnG] mice showing tdTomato and GFP expression with HD-TEC areas highlighted, with few or no tdTomato⁺ cells. Scale bar, 50 μm. **j**, Human tissue sections from a 50-year-old woman. Shown are consecutive sections with H&E, cytokeratin or CD1a staining. **k**, aaTEC1 and aaTEC2 gene signatures (top 20 marker genes from our mouse data converted to human orthologs; Supplementary Fig. 3 and Supplementary Table 3) were overlaid on human thymic epithelial cells ($n_{TEC}$ = 40,144) from single-cell sequencing datasets generated and published elsewhere[9,19,20]. Summary data represents mean ± s.e.m. and each dot represents an individual biological replicate. Statistics were generated (**b**–**d**) using a two-tailed Mann–Whitney test.

compared to all other TECs) formed clusters that were only detected in the involuted thymus and the medullary distribution (as assessed by hematoxylin and eosin (H&E) staining) of these signatures resembled that of HD-TEC clusters (Fig. 2d and Supplementary Table 3). To determine whether the HD-TEC structures were composed of aaTECs, we surveyed the expression of keratin subunits distinguishing aaTEC from other TEC populations (Fig. 2e,f). We found that most GFP+ TEC in these regions were K5+, some were K8+ and none were K14+ (Fig. 2e,f). Notably, the prospective aaTEC1 marker claudin-3 was capable of clearly demarcating HD-TEC regions in the aged thymus (Fig. 2e,f). Taken together, these data strongly suggest that the DN-TECs identified through flow cytometry, HD-TECs identified by imaging approaches and aaTECs identified through single-cell RNA sequencing (scRNA-seq) are the same cell populations.

## aaTECs are derived from FOXN1-expressing epithelial cells

Given the paucity of expression of TEC genes by aaTECs, yet their clear association with other TEC populations (Fig. 1a), we sought to establish the lineage relationship between aaTECs and TECs. Flow cytometric analysis of $Foxn1^{nTnG}$ mice confirmed the emergence of two atypical GFP+ TEC-derived populations with age: one population that was EpCAM+ but lacked canonical cTEC or mTEC markers, mirroring the DN-TEC/aaTEC1 population described above, and another that was EpCAM−, consistent with the aaTEC2 phenotype (Fig. 3a). Further analysis of these populations in $Foxn1^{nTnG}$ mice confirmed that claudin-3 expression was increased in EpCAM+UEA1− cells in 18-mo mice, while podoplanin was capable of marking EpCAM− aaTEC2 (Fig. 3b–d). These findings were supported by single-cell sequencing analysis of 20-mo nonhematopoietic thymic stroma from $Foxn1^{Cre}$ × R26-fl-Stop-fl tdTomato ($Foxn1^{tdTom}$) reporter mice, in which tdTomato is expressed in all cells with a history of $Foxn1$ expression. Integration of scRNA-seq data of 1,093 cells from 20-mo $Foxn1^{tdTom}$ TECs with the broader dataset of 7,412 (18-mo and 20-mo) wild-type (WT) cells revealed that transcription of the tdTomato reporter was detected in all TEC clusters, including both aaTEC1 and aaTEC2 (Fig. 3e). Together, these orthogonal lineage-tracing approaches confirm the thymic epithelial origin of both aaTEC1 and aaTEC2 subsets. We next assessed the lineage relationships among TEC subsets by performing unbiased RNA velocity analysis. Consistent with previous reports, this approach demonstrated a clear lineage trajectory in young mice stemming from the mTEC^prol and continuing into differentiated mTEC2 and mimetic cell lineages[5,7] (Fig. 3f,g, Extended Data Fig. 2d and Supplementary Video 2a,b). This relationship was preserved in the involuted thymus, with the additional progression of mTEC1 and early^prog cells into aaTEC1, whereas aaTEC2 were derived from early^prog and aaTEC1 (Fig. 3f,g, Extended Data Fig. 2d and Supplementary Video 2a,b)[8]. Although these data suggest that aaTECs are terminally differentiated subsets, given their expression of TEC precursor markers such as claudin-3 and Plet1 (Extended Data Fig. 4e), an alternative hypothesis is that aaTECs could instead represent a stalled progenitor cell differentiation stage.

## aaTEC niches do not support T cell development

Closer inspection of our Visium sequencing data highlighted a marked lack of thymocyte transcripts in aaTEC zones (Fig. 3h). Consistent with this observation, HD-TEC clusters in the involuted thymus of $Foxn1^{nTnG}$ mice excluded other tdTomato+ cells (Fig. 3i, Extended Data Fig. 5b and Supplementary Video 3), which given its ubiquitous expression in all non-TECs, will largely mark thymocytes. These data suggest that aaTEC microenvironments were thymocyte 'deserts' that do not support thymocyte differentiation and are deprived of thymic crosstalk factors. Notably, staining of sections of aged human thymus also reveals the presence of epithelial-rich thymocyte-poor regions (Fig. 3j), consistent with previously published findings[34]. Finally, we found that aaTEC gene signatures were apparent in previously generated human TEC scRNA-seq datasets, but the utility of prospective markers such as claudin-3 and podoplanin for the identification of aaTECs on human tissue is still to be determined[9,19,20] (Fig. 3k, Extended Data Fig. 5d–f and Supplementary Table 3).

## aaTEC form microenvironmental 'scars' associated with EMT

We next sought to understand the molecular alterations that may underlie thymic involution and how these relate to aaTECs. Despite the profound dysregulation that occurs during thymic involution, we found only relatively minor changes in the expression of key epithelial and thymocyte growth factors, largely restricted to fibroblasts (Extended Data Fig. 6a). cTECs are a crucial population involved with early thymocyte development, as well as regulating thymic size via expression of $Foxn1$ and its downstream targets such as $Dll4$ (refs. 1,3,21,35). We did not observe any change in expression with age of $Foxn1$, $Dll4$ or other genes involved in cTEC identity and function, including $Psmb11$, $Prss16$, $Enpep$ (which encodes Ly51) or $Ly75$ (which encodes CD205); although this is likely a technical limitation due to the low number of cTECs captured through single-cell sequencing of the whole CD45− compartment. To identify concerted age-dependent transcriptional changes within each stromal population, we used gene set enrichment analysis (GSEA)[36,37] coupled with network enrichment analysis using Cytoscape[38] to integrate the GSEA results into networks sharing common gene sets. Pathways with significant enrichment across at least one population with age were broadly grouped into eight biological categories corresponding with the hallmarks of aging (Fig. 4a, Extended Data Fig. 6b and Supplementary Tables 4 and 5)[39–41]. Consistent with reports demonstrating a link between mitochondrial function, aging and senescence[42,43], we found a broad decrease in the transcription of genes within pathways associated with mitochondrial function and metabolism across most cell populations and in fibroblasts upregulation of genes associated with immune function (including antigen processing and presentation). Together, these findings support the role of these cells and pathways in the induction of inflammaging or the senescence-associated secretory phenotype (SASP)[44,45]. Notably, changes in pathways associated with proteostasis were largely restricted to epithelial cell populations (Fig. 4a, Extended Data Fig. 6b and Supplementary Tables 4 and 5).

**Fig. 4 | Age-associated TEC regions are non-functional and associated with EMT. a**, GSEA pathway analysis was performed for each subset based on differentially expressed genes in 18-mo versus 2-mo mice (Supplementary Tables 4 and 5) and Cytoscape network analysis was used to integrate enriched pathways (false discovery rate (FDR) ≤ 0.05) sharing a core set of genes. Dotplot of top five pathways within each category (Supplementary Table 5). **b**, GSEA pathway enrichment within aaTEC1 or aaTEC2 subsets (generated by comparing aaTEC1 and aaTEC2 to all other TECs; Supplementary Table 6). **c**, Heatmap of 8,795 genes within cTEC, mTEC1, aaTEC1, aaTEC2 and medFB subsets ranked by cadherin-1 (encoded by $Cdh1$) expression. **d**, Scatter-plot of $Cdh1$ and $Vim$ with cTEC, mTEC1, aaTEC1, aaTEC2 and medFB subsets. **e**, Scatter-plot of $Cdh1$ and $Vim$ transcription overlaid with expression of epithelial and mesenchymal genes and EMT known regulators. **f**, Expression of key epithelial genes and thymopoietic factors by various 18-mo TEC subsets, including aaTECs. **g**, Heatmap of AIRE-(left) and FEZF2-dependent/independent (right) genes reported previously[49]. Heatmap shows scaled normalized gene expression. **h**, CellChat interaction overview summarizing number of interactions between grouped populations in 2-mo and 18-mo mice. **i**, CellChat interaction analysis between stromal cell populations with early^prog, mTEC1, aaTEC1 or aaTEC2 as cellular receivers (see also Extended Data Fig. 8b). Matrix represents all significantly enriched pathways targeting either early^prog, mTEC1, aaTEC1 or aaTEC2 (color-coded by the receiver population) and split by the type of CellChat signaling (secreted, cell–cell and ECM). **j**, Levels of PTN and MK in thymus at 2-mo ($n = 5$), 12-mo ($n = 5$) or 18-mo ($n = 5$) female C57BL/6 mice. Summary data represents mean ± s.e.m. and each dot represents an individual biological replicate. Statistics for **j** were generated using the Kruskal–Wallis test with Dunn's correction. ECM, extracellular matrix.

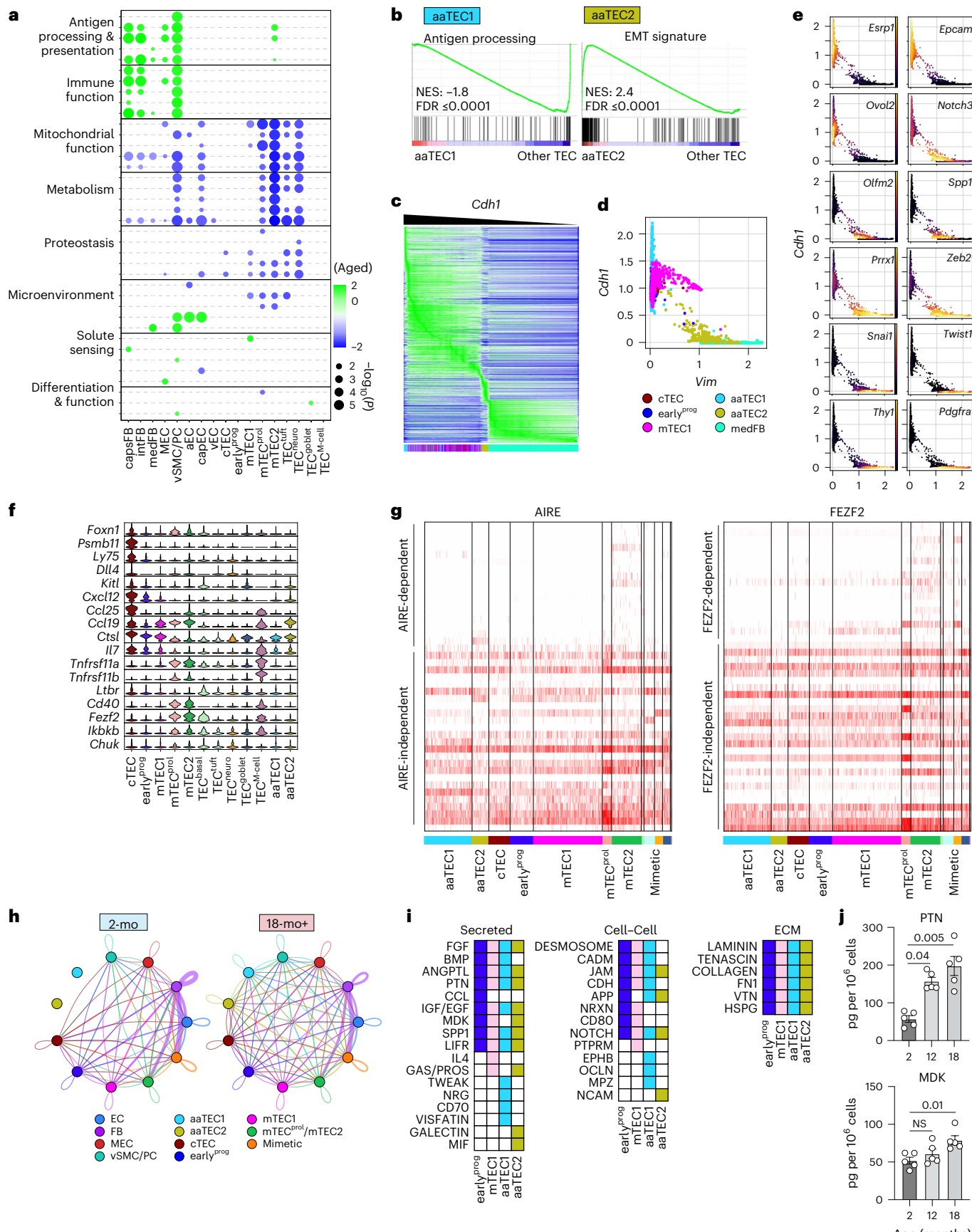

Because the approach above involved direct comparison of cell populations from young and aged mice, aaTECs (which are not present in significant numbers in young mice) could not be assayed. Therefore, we inspected enriched pathways of aaTECs versus all other TECs within the aged setting. GSEA revealed loss of antigen presentation within aaTEC1s (Fig. 4b and Supplementary Table 6), consistent with the progressive loss of thymic epithelial function upon differentiation into aaTECs. For aaTEC2s, one of the most highly enriched pathways was the hallmark EMT gene signature (Fig. 4b and Supplementary Table 6). Analysis of gene expression across 8,795 EMT-related genes in descending order according to expression of E-cadherin (*Cdh1*), as previously described[46], further suggested that aaTEC2 lay in a liminal zone between epithelial and mesenchymal identity (Fig. 4c). Scatter-plots based on E-cadherin (*Cdh1*) and vimentin (*Vim*) transcriptional expression (as prototypical epithelial and mesenchymal markers, respectively) overlaid with expression of archetypal epithelial or mesenchymal genes, further suggested that aaTEC2 lost epithelial features and partially gained mesenchymal traits (Fig. 4d,e). These observations are consistent with a partial EMT (pEMT) state that parallels a senescence-associated pEMT tubular epithelial population identified in kidney fibrosis[47], although we do not exclude the possibility that some aaTEC2s undergo a full EMT[48].

The unusual morphology and microenvironment formed by aaTECs prompted us to further assess their function relative to typical TECs. Consistent with scRNA-seq analyses, imaging and flow cytometry confirmed that aaTECs did not represent prominent, known mimetic cells, lacking expression of markers of tuft cells, M cells or corneocytes (Extended Data Fig. 7a–c). We found that aaTECs expressed low levels of key TEC mediators of T cell differentiation, including *Dll4*, *Psmb11*, *Kitl*, *Ccl25*, *Cxcl12* and *Il7* (Fig. 4f). Neither aaTEC populations expressed *Fezf2*, *Aire*, key NF-κB target genes or the receptors *Tnfrsf11a* (which encodes RANK), *Ltbr* or *Cd40*, which are associated with driving mTEC differentiation (Fig. 4f and Extended Data Fig. 7d,e). Accordingly, we found little or no expression of AIRE-dependent or FEZF2-dependent tissue-restricted antigen (TRA) expression[49] in aaTEC1s or aaTEC2s, being largely restricted to mTEC2 and mTEC^prol populations (Fig. 4g and Extended Data Fig. 8a).

Using CellChat[50] to identify cell–cell interactions between stromal populations, we found a clear shift with age where signals were redistributed with the emergence of the two aaTEC populations (Fig. 4h). Focusing on specific signaling pathways influencing aaTEC1s and aaTEC2s, as well as their putative upstream mTEC1s and early^prog precursors (as well as their sources), we found enrichment for growth factors such as FGF, BMP and EGF, as well as factors associated with promoting EMT such as PTN (pleiotrophin), MDK (midkine) and ANGPTL (angiopoietin-like) coming from across the stromal compartment (Fig. 4i and Extended Data Fig. 8b)[51–56]. Consistent with this observation, the amounts of PTN and MDK increased in the thymus with age (Fig. 4j) and, along with many other TECs, aaTECs expressed abundant EMT-related integrins and *Sdc4*, which can act as a receptor for PTN, MDK and ANGPTL4 (ref. 57) (Extended Data Fig. 8c). Given the link between EMT, fibrosis and senescence, and the well-described loss of TRA expression with age[5], these data suggest that aaTECs form unique nonfunctional microenvironments in the involuted thymus associated with senescence and EMT[32,58,59]. Although not fibrotic themselves, these aberrant high-density aaTEC 'scars' mirror the mesenchymal scarring found in other tissues with age and may be responsible for diminishing overall thymic function[32,41].

## aaTECs limit thymic repair following acute injury in aged mice

The thymus is extremely sensitive to injury but has substantial capacity for repair. The ability of the thymus to regenerate is thought to decline with age yet the mechanisms of this deficit are poorly understood[2,3,60]. We found that 1–2-mo mice subjected to sublethal total

body irradiation (TBI) had fully recovered thymic cellularity by day 28 after TBI (Fig. 5a). By contrast, aged mice exhibited a significant delay in the restoration of thymic cellularity and did not approach pre-damage levels until approximately day 42 (Fig. 5a). Plotting thymic size after damage relative to baseline cellularity showed that restoration of thymic cellularity during the regenerative phase was impaired in aged mice (Fig. 5b). Histomorphological analysis correlated with these findings, showing similar depletion of thymocytes, accumulation of cellular debris and granulation tissue formation observed 1 day after injury in both cohorts, but evidence of sustained fibrosis, dystrophic calcification and occasional dyskeratotic epithelial cells only in 18-mo mice, suggesting a relative dysfunction in the regenerative process with age (Extended Data Fig. 9a). To assess the stromal compartments that orchestrate thymic regeneration, we performed flow cytometric analysis of endothelial, mesenchymal and epithelial stromal cell populations before and after regeneration. Although there were few major differences in the early response of young versus aged mice in most populations (Extended Data Fig. 9b), one notable exception was the re-emergence of EpCAM+MHCII+Ly51−UEA1− DN aaTEC1s and EpCAM−MHCII+PDGFRα−PDPN+ aaTEC2s, which are depleted but rapidly restored as a prominent feature of the regenerated thymus of aged mice (Fig. 5c,d and Extended Data Fig. 9b). Similarly, whole-organ imaging analysis of aged *Foxn1*^nTnG mice 28 days after TBI revealed prominent high-density aaTEC regions after damage (Fig. 5e and Supplementary Video 4). Quantification revealed that, despite the thymus remaining smaller than pre-damage, aaTECs comprised an increased volume and number, indicating a substantial preferential increase in aaTECs coincident with the impaired thymic regeneration of aged mice (Fig. 5e,f).

To explore comprehensively the mechanisms of the damage response and impaired regeneration in aged thymic stroma, we analyzed 58,309 CD45− nonhematopoietic cells from the thymus of 2-mo and 18-mo by scRNA-seq at days 1, 4 and 7 after TBI and annotated them based on steady-state (day 0) subset signatures before integration (Fig. 5g–i, Extended Data Fig. 10a, Supplementary Fig. 1 and Supplementary Table 3). Age-associated TEC populations remained a major feature of the aged thymus after damage (Fig. 5j). RNA velocity analysis implied that, in 2-mo mice, the re-emergence of differentiated conventional mTEC populations (mTEC2, mimetic cells) stemmed largely from the mTEC^prol populations (Fig. 5k). In contrast, after damage in 18-mo mice there was a skewing in the inferred direction and rather than mTEC^prol cells driving differentiation toward conventional differentiated mTECs, mTEC1 and early^prog cells drove differentiation toward aaTEC1 and aaTEC2 (Fig. 5k), consistent with findings by others[5]. Therefore, the emergence of aaTECs may perturb normal TEC differentiation in the early stages of regeneration.

## aaTECs co-opt growth signals at baseline and during repair

After acute damage, we and others have demonstrated the importance of epithelial growth factors such as BMP4 and keratinocyte growth factor (KGF) for endogenous thymic repair[31,61]. Indeed, we found a broad upregulation of endogenous epithelial regenerative factors after damage such as *Fgf7*, *Fgf10*, *Fgf21* and *Bmp4*, as well as thymopoietic factors such as *Flt3l* and *Kitl* (Fig. 6a and Supplementary Tables 7–10). In contrast, stroma from aged mice did not upregulate these regenerative programs to the same extent or, in many instances, at all (Fig. 6a and Supplementary Tables 7–10). cTECs are crucial as targets for epithelial regenerative cues via expression of proteins such as FOXN1 and its downstream targets such as *Dll4*, *Ccl25* and *Cxcl12* (ref. 31). Consistent with this, *Foxn1* expression increased in cTECs after damage, which was maintained in aged mice, which was also reflected in its downstream chemokine targets *Ccl25* and *Cxcl12* (Fig. 6a,b and Supplementary Tables 7–10). In contrast, increased expression of the key regeneration mediator *Dll4* and *Psmb11* (a gene encoding for β5t, a proteasome subunit critical for CD8+ T cell selection), were abrogated or decreased,

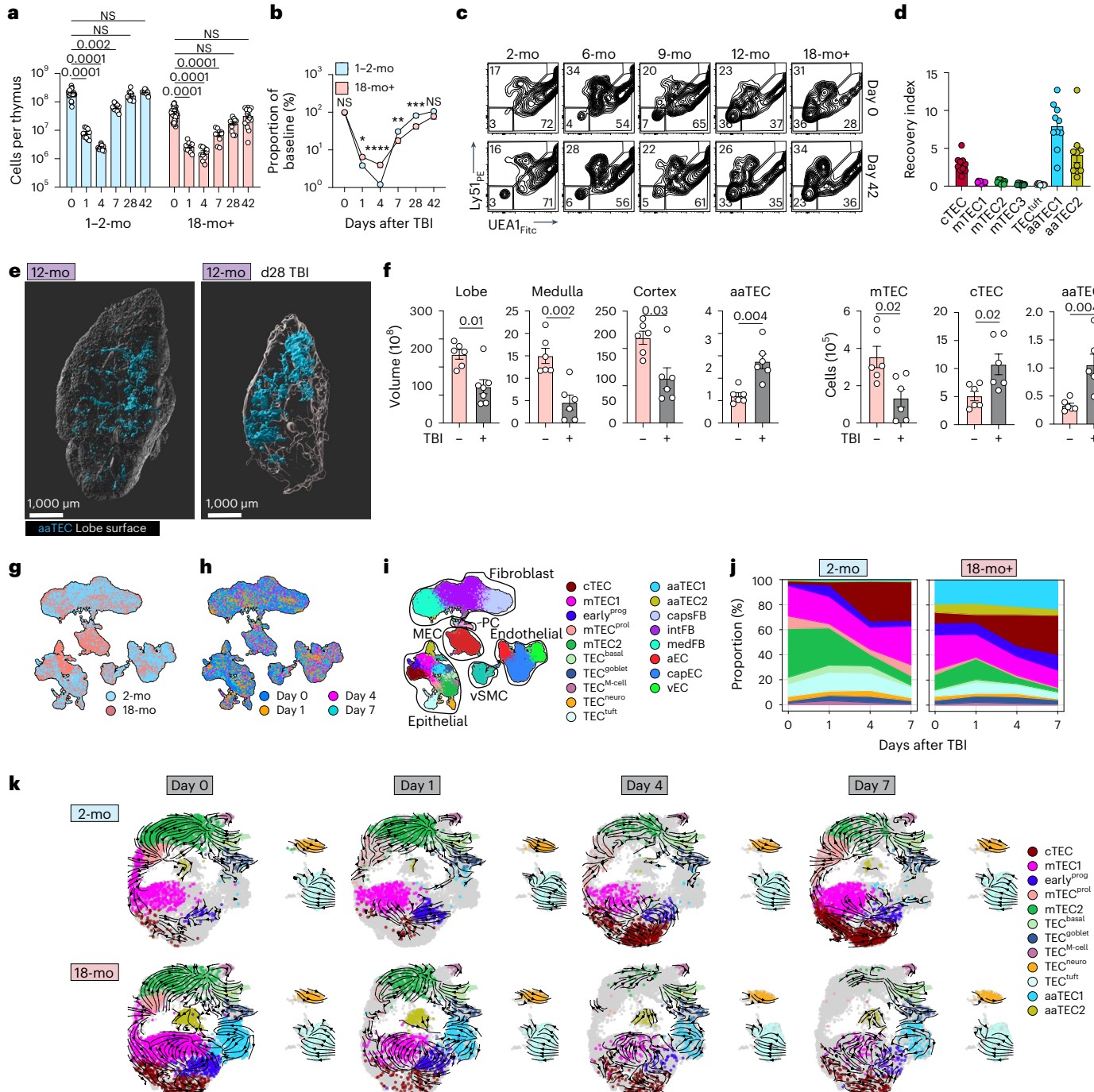

**Fig. 5 | Aging negatively impacts thymic regeneration. a,b,** 2-mo or 18-mo female C57BL/6 mice were given a sublethal dose of TBI (550 cGy) and the thymus was assessed at the indicated time points ($n = 30$, 2-mo day 0; 10, 2-mo day 1; 10, 2-mo day 4; 10, 2-mo day 7; 10, 2-mo day 28; 10, 2-mo day 42; and 35, 18-mo day 0; 10, 18-mo day 1; 10, 18-mo day 4; 10, 18-mo day 7; 10, 18-mo day 28; 15, 18-mo day 42). Total thymic cellularity (**a**). Proportion of total thymic cellularity at the indicated time points as a function of steady-state age-matched cellularity (**b**). **c,** Flow cytometry plots (gated on CD45⁻EpCAM⁺) showing DN-TECs at day 42 after TBI in C57BL/6 mice at the indicated ages. **d,** Depletion and recovery of indicated populations were quantified by flow cytometry over the first 7 days after TBI in 2-mo or 18-mo mice and area under the curve was calculated (Extended Data Fig. 9b). An aging index was generated by calculating the ratio of aged to young AUC for each indicated population ($n = 10$ per cell type). **e,** Whole-tissue imaging of 12-mo male *Foxn1*ⁿᵀⁿᴳ mice at baseline or 28 days after

TBI (550 cGy). **f,** Total volume and volume of cortex, medulla and aaTEC regions, as well as the number of cTECs, mTECs and aaTECs of the thymic right lobe. **g–j,** scRNA-seq was performed on CD45⁻ cells isolated from 2-mo or 18-mo thymus at baseline (day 0) and days 1, 4 and 7 after TBI. UMAP of 81,241 CD45⁻ cells annotated by age cohort (**g**), day after TBI (**h**) or structural cell subset mapped from Fig. 1c and Extended Data Fig. 10a (**i**). **j,** Associated frequency analysis of all TEC subsets after TBI within each age cohort. **k,** RNA velocity analysis on all TEC subsets at days 0, 1, 4 and 7 after TBI in 2-mo or 18-mo mice. Summary data represent mean ± s.e.m. and each dot represents an individual biological replicate. Statistics were generated using the Kruskal–Wallis test with Dunn's correction (**a,d**) and two-tailed Mann–Whitney test (**b,f**). For **b**, statistics represent a comparison of 2-mo to 18-mo mice within each time point. *$P = 0.02$; **$P = 0.004$; ***$P = 0.002$; ****$P < 0.0001$.

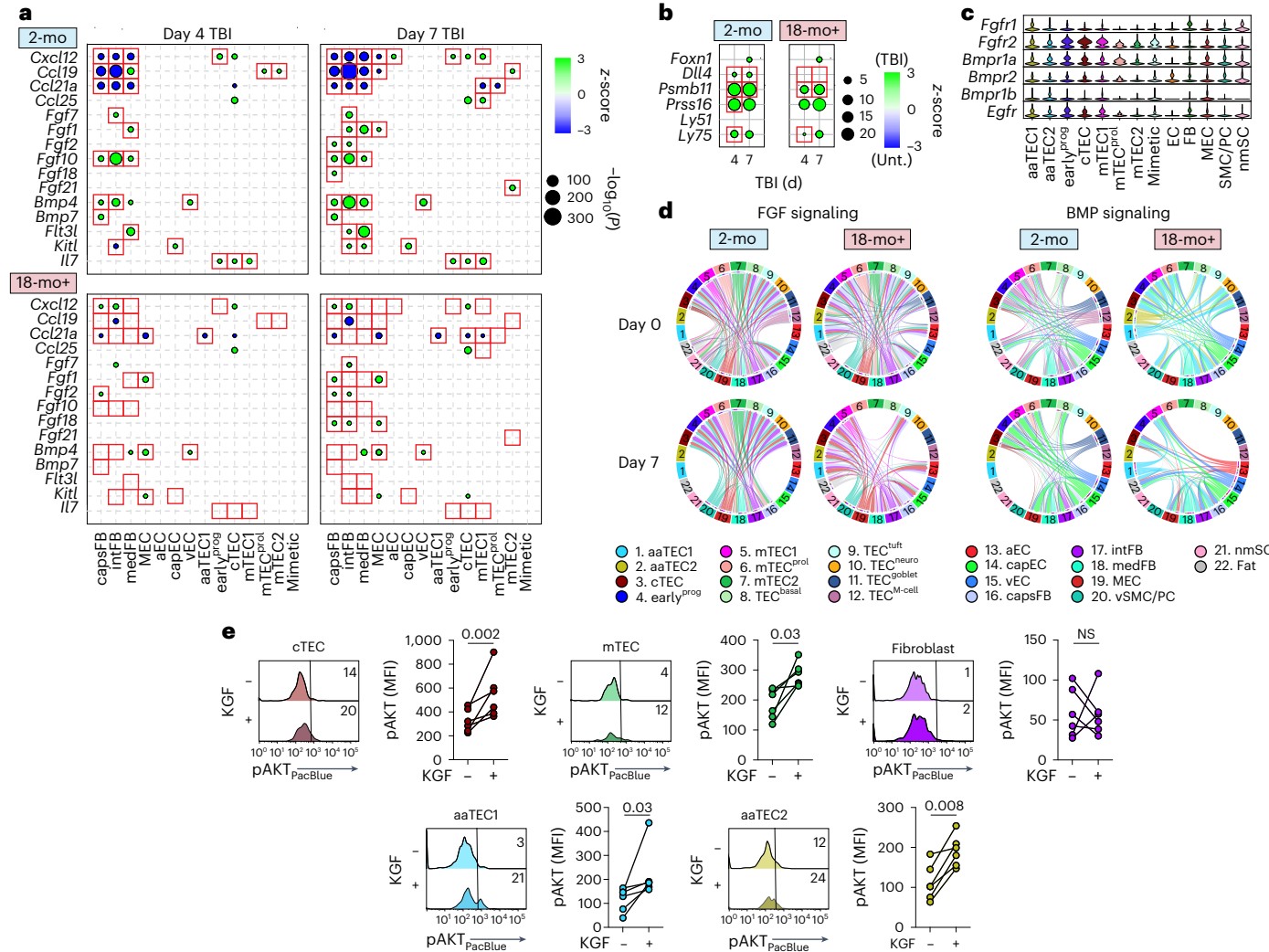

**Fig. 6 | Muted transcriptional response to irradiation in the aged thymic stroma linked to aaTEC disruption of trophic signals. a**, Differential expression after damage within stromal cell subsets of key thymopoietic and epithelial growth factors in 2-mo or 18-mo female C57BL/6 mice at days 4 and 7 after TBI (compared to day 1). **b**, Differential expression of cTECs associated genes at days 4 and 7 after TBI in 2-mo and 18-mo mice. **c**, Expression of receptors for epithelial growth factors across stromal cell subsets. **d**, Chord diagram interaction analysis of FGF and BMP signaling pathways in 2-mo or 18-mo mice at baseline and at day 7 after TBI. **e**, CD45⁻ nonhematopoietic stromal cells were isolated from 18-mo female mice and incubated for 5 min with KGF (100 ng ml⁻¹) when cells were stained for phosphorylated AKT by flow cytometry (*n* = 6 thymuses isolated from individual mice). Summary data represent mean ± s.e.m. and each dot represents an individual biological replicate. Statistics were generated using a two-tailed paired *t*-test.

respectively, in aged mice (Fig. 6b and Supplementary Tables 7–10). Receptors for epithelial growth factors were broadly expressed across TEC subsets, including both aaTEC1s and aaTEC2s (Fig. 6c); however, using CellChat[50] to infer interactions based on expression of receptor–ligand pairs, as well as downstream signaling effectors, demonstrated a clear skewing of interactions with age toward aaTECs for two prominent growth cues for TECs: FGF and BMP signaling (Fig. 6d). These data suggest that aaTECs draw these pro-growth factors away from conventional TECs with age. This was supported by active signaling through FGF receptors in response to KGF in cTECs, mTECs and aaTECs, but not fibroblasts, which should not respond to KGF (Fig. 6e).

## Reduction in FOXN1 activity favors aaTEC differentiation

FOXN1 is the master regulator of thymic epithelial lineage, crucial for TEC differentiation and function. It is also important for thymic regeneration after damage, acting as a key target downstream of these regeneration pathways[2,3]. FOXN1 protein could be readily detected in cTEC and mTEC populations but not aaTEC1s or aaTEC2s (Fig. 7a).

Consistent with this, there was little transcription of *Foxn1* by aaTEC subsets compared to putative precursor populations (cTEC, early^prog and mTEC1; Figs. 4f and 7b,c). Using Dynamo[62] to predict the differentiation potential (negative stem cell potential) of cells within the 18-mo TEC dataset, we found that *Foxn1* expression correlated only with differentiation away (and not toward) aaTECs (Fig. 7d,e).

To reveal quantitative insights into the regulatory capacity of *Foxn1*, we first computed the RNA Jacobian using *Foxn1* as both regulator and target gene (Fig. 7f). The response heatmap showed that self-activation of *Foxn1* followed an almost linear trend with few intermediate plateaus, suggesting endogenous *Foxn1* self-induction and differential regulation of *Foxn1* targets based on expression level, consistent with previously published reports of the self-regulation of FOXN1 (ref. 63). Next, we applied in silico perturbation analysis of *Foxn1* to assess the impact of this major regulator on cell fate outcomes. We first simulated the impact of *Foxn1* deactivation and found that it diverted differentiation away from mTEC^prol and mTEC2 subsets and toward the aaTECs (Fig. 7g). In contrast, computationally activating *Foxn1* expression reinforced the transition of cTEC, early^prog and mTEC1

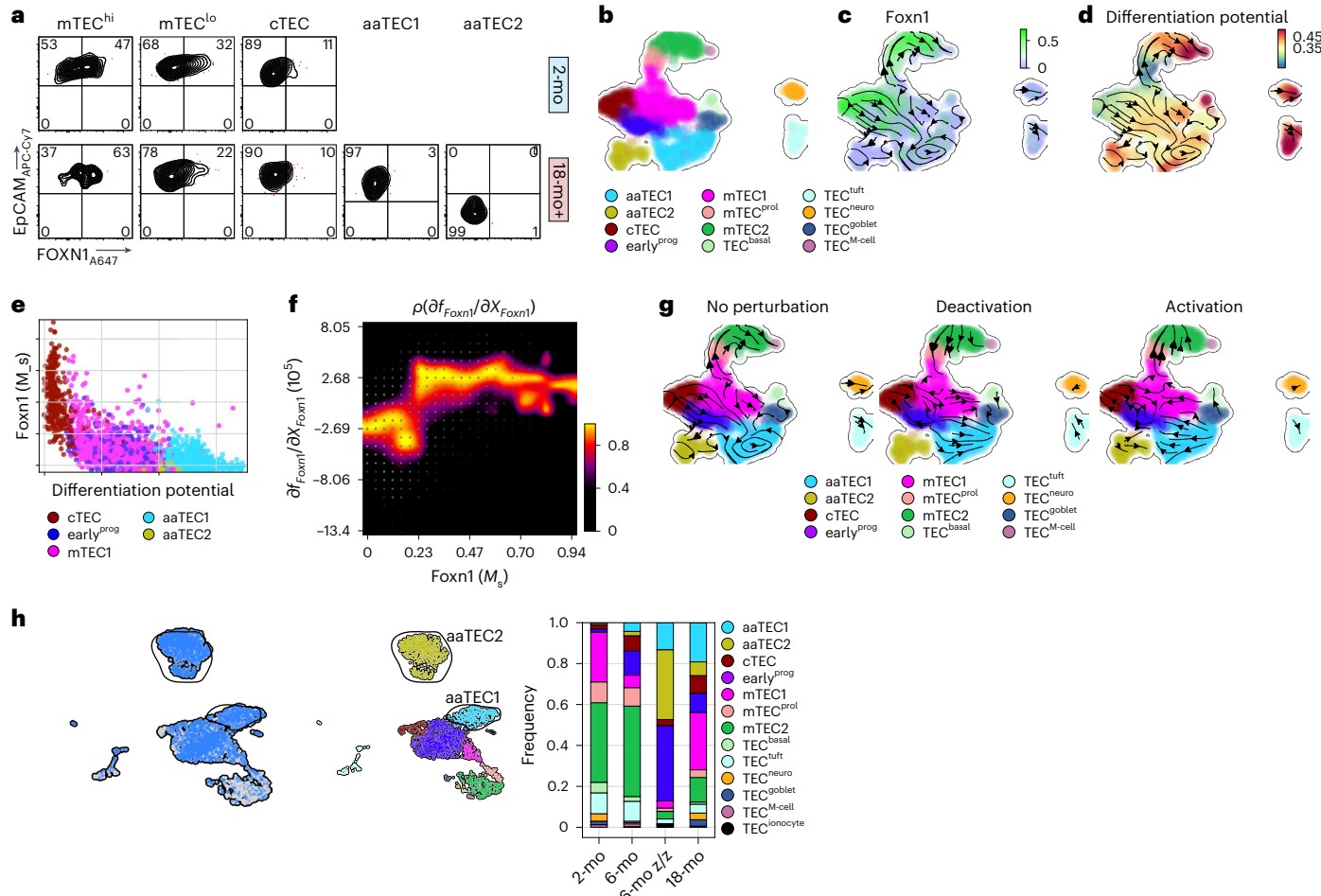

**Fig. 7 | FOXN1 loss of function accelerates aaTEC emergence with age.**
**a**, FOXN1 expression across TEC subsets by flow cytometry. **b**–**d**, UMAP of 18-mo TEC data at steady state, color-coded for TEC subset (**b**), *Foxn1* expression (**c**) and differentiation potential (**d**) with projected RNA velocity data. **e**, Scatter-plot of differentiation potential (**d**) versus *Foxn1* expression levels color-coded by subset annotation. **f**, Response heatmap showing the RNA Jacobian element for *Foxn1* self-induction (*df*$_{Foxn1}$/*dX*$_{Foxn1}$) versus the *Foxn1* expression (*Foxn1* ($M_s$)).

**g**, In silico perturbation analysis of *Foxn1* in the 18-mo epithelium at steady state and accompanied cell fate diversions. Velocity arrows in UMAPs show cell fate directionality in the unperturbed dataset (left) and after in silico suppression (middle) or induction (right) of *Foxn1*. **h**, scRNA-seq was performed on CD45⁻ cells from 6-mo *Foxn1*^Z/Z mice and controls. UMAPs of 3,594 cells color-coded by sample and mapped to our TEC subsets.

subsets toward the normal differentiation trajectory and away from the aaTEC compartment (Fig. 7g). To test these in silico perturbation predictions, we turned to the *Foxn1*^Z/Z mouse model expressing a hypomorphic allele that causes early loss of *Foxn1* in TECs and premature thymic involution[64]. scRNA-seq of TECs from 6-mo *Foxn1*^Z/Z mice and controls was integrated and compared to our 2-mo and 18-mo datasets. The TEC subset frequency in 6-mo *Foxn1*^WT/WT mice was comparable to 2-mo WT mice for the majority of populations, with the emergence of small but detectable populations of aaTEC1 and aaTEC2 subsets (Fig. 7h and Extended Data Fig. 10b); however, in 6-mo *Foxn1*^Z/Z mice we observed a large expansion of aaTEC1s and aaTEC2s collectively composing approximately 50% of TEC; an expansion even greater than 18-mo WT mice (Fig. 7h and Extended Data Fig. 10b). Collectively, these data suggest that loss of FOXN1 expression in TECs favors differentiation into an aaTEC fate.

## Discussion

Here, we define age-associated changes to the thymic microenvironment in the involuting thymus that impairs function in two ways. First, atypical aaTECs form high-density epithelial clusters, devoid of thymocytes. The accretion of aaTEC regions directly contributes to the loss of functional thymic tissue with age and, given that these regions

expand after damage, exacerbates injury-induced atrophy. Second, we found evidence that the emergence of aaTECs perturbs the network of growth factors supporting stromal cell function and thymocyte differentiation, likely constituting an additional impediment to thymic function. Notably, similar features of epithelial-rich, thymocyte-devoid regions can also be observed in the human thymus[34] and our data offer strategies for further interrogation of aaTECs, including the validation of claudin-3 expression by aaTEC1s and podoplanin by aaTEC2s.

Despite the relatively early emergence of aaTECs, their genesis seems to be linked to hallmarks of aging. Genetic approaches demonstrated that aaTEC populations were derived from *Foxn1*⁺ precursors, yet both had lost expression of canonical markers of cTECs and mTECs. RNA velocity analysis suggest that both aaTEC1s and aaTEC2s derive from mTEC1s, consistent with its progenitor-like phenotype[5,7,65], as well as a mTEC1-like cell that shares a signature with a recently described progenitor[8]; however, more sophisticated lineage-tracing approaches will be required to more directly test the precursor–progeny relationships. Nevertheless, our data strongly suggest that loss of FOXN1 is a key driver for the emergence of aaTECs. FOXN1 is crucial for many aspects of TEC biology including their generation, maintenance and regeneration[33]. FOXN1 expression declines with age, a process that has been implicated in contributing to thymic involution[33,64]. Age-associated

TECs do not express FOXN1 nor its downstream targets. Moreover, in mice expressing a FOXN1 hypomorph[64] we found that accelerated emergence of aaTECs accompanies early thymic atrophy; however, although our velocity analysis suggests that aaTECs are the downstream products of differentiation from mTEC1s and early[prog] and may represent end-stage epithelial cells arising and accumulating in the involuted thymus, an alternative hypothesis could be that aaTECs are progenitor cells that have become blocked. In fact, expression of markers known to be expressed by TEC precursors such as claudin-3 and Plet1 could support this alternative interpretation and further studies will be needed to test these hypotheses[66–68].

The molecular drivers of aaTECs likely also involve additional factors. For instance, we found considerable changes in the stromal microenvironment beyond TECs, including significant changes in fibroblasts that mirror many of the main hallmarks of aging[39,41], reflected by a loss in mitochondrial, metabolic and proteostasis programs, and increase in pathways associated with inflammaging or the SASP[39]. This latter finding likely also reflects the broader role of inflammaging and SASP in driving EMT in aged tissues[58,59]. This link is especially notable given that one of the main features of the aging thymus is the replacement of functional tissue with fat[69,70], which can directly drive loss of thymic function[71]. Furthermore, there is evidence that the emergence of fat in the human thymus may be triggered by EMT[72]. These data also support recent observations that age-associated changes in tissues are organ- and cell-lineage specific (for example senescence-associated inflammaging impacting on muscle regeneration)[32,39,40]. Our findings are therefore consistent with the concept that aaTECs represent a thymus-specific manifestation of these programs, and that age-related changes in other cells of the thymic microenvironment, in particular fibroblasts, could also contribute to aaTEC emergence; however, our data also suggest that the emergence of aaTECs impairs thymic function beyond the replacement of functional tissue, by competing with conventional TECs for growth signals. This effect seemed especially important after acute damage where expansion of aaTECs correlated with impairment to thymic regeneration, a significant clinical problem given the relationship between T cell reconstitution and clinical outcomes following HCT[4]. Notably, there is also evidence of epithelial-rich areas devoid of thymocytes in human thymus found only in aged and dysregulated tissue (such as myasthenia gravis)[34]; however, without a direct way to deplete these cells (or block their emergence to begin with) further studies will be needed to elucidate the specific contributions of aaTECs in thymic aging and responses after damage.

These observations suggest that the defective response and recovery to acute damage with age could be due to: (1) failure of aged stromal subsets to orchestrate regenerative programs; (2) upregulation of a partial EMT program leading to expansion of aaTEC; and (3) the co-opting of regenerative factors by aaTEC from typical differentiated mTEC subsets. These features seem to be at least partially driven by changes within the fibroblast compartment, and particularly their upregulation of genes associated with inflammaging/SASP. In summary, these studies highlight unique stromal cell responses to age- and stress-related thymic atrophy. Furthermore, the discovery of aaTECs, along with the functional changes in fibroblasts with age consistent with inflammaging/SASP, provide therapeutic targets for improving T cell immunity more broadly. Age-associated TECs therefore constitute a nexus of stromal cell dysfunction in thymic involution and impaired regeneration.

## Online content

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

Anastasia I. Kousa [1,2,3,17], Lorenz Jahn [1,17], Kelin Zhao[4,5,17], Angel E. Flores[6], Dante Acenas II[2,7], Emma Lederer [2,7], Kimon V. Argyropoulos[8], Andri L. Lemarquis[1,3], David Granadier[2], Kirsten Cooper [2], Michael D'Andrea [4,5], Julie M. Sheridan [4], Jennifer Tsai [1], Lisa Sikkema[9,10], Amina Lazrak[1], Katherine Nichols[1], Nichole Lee[1], Romina Ghale[1], Florent Malard[1,11], Hana Andrlova[1], Enrico Velardi [12], Salma Youssef [1], Marina Burgos da Silva[1], Melissa Docampo[1], Roshan Sharma[9], Linas Mazutis [9], Verena C. Wimmer[4,5], Kelly L. Rogers [4,5], Susan DeWolf[13], Brianna Gipson [1], Antonio L. C. Gomes[1], Manu Setty [9,14], Dana Pe'er [9], Laura Hale[15], Nancy R. Manley[6,16], Daniel H. D. Gray [4,5,18] ✉, Marcel R. M. van den Brink [1,3,13,18] ✉ & Jarrod A. Dudakov [2,7,18] ✉

[1]Program in Immunology, Memorial Sloan Kettering Cancer Center, New York, NY, USA. [2]Translational Science and Therapeutics Division, and Immunotherapy Integrated Research Center, Fred Hutchinson Cancer Center, Seattle, WA, USA. [3]City of Hope Los Angeles and National Medical Center, Duarte, CA, USA. [4]The Walter and Eliza Hall Institute of Medical Research, Parkville, Victoria, Australia. [5]Department of Medical Biology, The University of Melbourne, Parkville, Victoria, Australia. [6]Department of Genetics, University of Georgia, Athens, GA, USA. [7]Department of Immunology, University of Washington, Seattle, WA, USA. [8]Department of Pathology and Laboratory Medicine, Memorial Sloan Kettering Cancer Center, New York, NY, USA. [9]Computational and Systems Biology Program, Memorial Sloan Kettering Cancer Center, New York, NY, USA. [10]Institute of Computational Biology, Helmholtz Center Munich, Munich, Germany. [11]Sorbonne Université, Centre de Recherche Saint-Antoine INSERM UMRs938, Service d'Hématologie Clinique et de Thérapie Cellulaire, Hôpital Saint Antoine, AP-HP, Paris, France. [12]Division of Pediatric Hematology and Oncology, Bambino Gesù Children's Hospital, IRCCS, Rome, Italy. [13]Department of Medicine, Memorial Sloan Kettering Cancer Center, New York, NY, USA. [14]Basic Sciences Division & Translational Data Science Integrated Research Center, Fred Hutchinson Cancer Center, Seattle, WA, USA. [15]Human Vaccine Institute, Duke University, Durham, NC, USA. [16]School of Life Sciences, Arizona State University, Phoenix, AZ, USA. [17]These authors contributed equally: Anastasia I. Kousa, Lorenz Jahn, Kelin Zhao. [18]These authors jointly supervised this work: Daniel H. D. Gray, Marcel R. M. van den Brink, Jarrod A. Dudakov. ✉e-mail: dgray@wehi.edu.au; mvandenbrink@coh.org; jdudakov@fredhutch.org

## Methods

### Experimental methods

**Tissue collection.** Inbred male and female C57BL/6J mice were obtained from The Jackson Laboratories or through the National Institute of Aging mouse colony. $Foxn1^{tdTomato}$ mice were generated by crossing $Foxn1$-cre (Jax 018448) with B6.Cg-Gt(ROSA) 26Sor$^{tm14(CAG-tdTomato)Hze}$/J mice (Jax 007914). $Foxn1^{nTnG}$ mice were generated by crossing $Foxn1$-cre (Jax 018448) with ROSA$^{nT-nG}$ mice (Jax 023537). $Foxn1^{z/z}$ mice were generated as previously described[64]. As a model of thymic injury, mice were given a sublethal dose of TBI (550 cGy) with a Cs-137 γ-radiation source. Mice were maintained at The Memorial Sloan Kettering Cancer Center, Fred Hutchinson Cancer Center, Walter and Eliza Hall Institute or University of Georgia animal houses. All experiments were performed according to Institutional Animal Care and Use Committee guidelines.

Human thymus tissue was obtained from the archives of the Duke University Department of Pathology as formalin-fixed paraffin-embedded sections. All tissues were used anonymously, only recording patient age, sex and surgical diagnosis. We show images from one 50-year-old female patient. All human tissues were collected according to a protocol approved by the Duke University Institutional Review Board.

**Isolation of cells and flow cytometry.** The thymus was enzymatically digested following an adapted protocol[73,74]. In brief, thymi were mechanically dissociated into 1–2-mm pieces. Tissue pieces were incubated with a digestion buffer (either RPMI with 10% FCS, 62.5 µm ml⁻¹ liberase TM, 0.4 mg ml⁻¹ DNase I; or RPMI with 25 mM HEPES, 20 µg µl⁻¹ DNase1 and 1 mg ml⁻¹ collagenase/dispase). Between incubation steps, supernatant containing dissociated cells were transferred to tubes equipped with a 100-µm filter. Cells were pelleted by centrifugation at $400g$ for 5 min. All steps were performed at 4 °C unless indicated. For sequencing experiments, cell pellets were incubated with anti-mouse CD45 microbeads and CD45$^+$ cells were depleted from cell suspension using magnetic-associated cell sorting (MACS) on LS columns according to the manufacturer's protocol. Following red blood cell lysis using ACK buffer, the CD45-depleted cell fraction was incubated with an antibody cocktail for 15 min at 4 °C and cells of interest were purified by fluorescent-associated cell sorting (FACS) on a BD Biosciences Aria II using a 100-µm nozzle. Cells were sorted into tubes containing RPMI supplemented with 2% BSA. FACS-purified cells were spun down at $400g$ for 5 min and resuspended in PBS supplemented with 0.04% BSA for generation of single-cell suspensions.

For flow cytometry and cell sorting, surface antibodies against CD45 (30-F11), CD31 (390 or MEC13.3), TER-119 (TER-119), MHCII IA/IE (M5/114.15.2), EpCAM (G8.8), Ly51 (6C3), PDGFRα (APA5), CD104 (346-11A), L1CAM (555), Ly6D (49-H4), Gp38 (8.1.1), CD26 (H194-112), CD62P (RB40.34), podoplanin (8.1.1), CD62P (RB40.34), CD9 (KMC8) and CD309 (Avas12a) were purchased from BD Biosciences, BioLegend or eBioscience. Ulex europaeus agglutinin 1 (UEA1) was purchased from Vector Laboratories. Antibody against phospho-AKT was purchased from Cell Signaling Technologies; claudin-3 and anti-rabbit secondary antibodies were purchased from Invitrogen (Thermo Fisher); DCLK1 (aa690-720) was purchased from LSBio; GP2 (2F11-C3) was purchased from MBL Life Science; and anti-GFP (Aves GFP-1020) was purchased from AvesLabs. Anti-FOXN1 antibody was a gift from H.-R. Rodewald[75]. Flow cytometry was performed on a Fortessa X50 or Symphony A6 (BD Biosciences) and cells were sorted on an Aria II (BD Biosciences) using FACSDiva (BD Biosciences). Analysis was performed by FlowJo (Treestar Software). Detail of specific vendors, fluorochromes, catalog numbers, lot numbers dilutions and gating can be found in Supplementary Tables 11, 12.

Cells were isolated as described and depleted of CD45$^+$ cells by MACS depletion (Miltenyi Biotech). CD45$^-$ cells were incubated for 5 min with recombinant mouse KGF (100 ng ml⁻¹) when phospho-AKT was assessed. For phospho-AKT staining, cells were fixed and permeabilized in 1.6% paraformaldehyde at 37 °C followed by 90% methanol at 4 °C. After thorough washing to remove all methanol, cells were stained for both intracellular and extracellular antigens simultaneously.

**Thymic tissue clearing and immunofluorescence.** After euthanasia, mice were transcardially perfused with PBS followed by 4% PFA. Thymi were dissected and post-fixed in 4% PFA for 4 h at 4 °C. For confocal imaging, fixed tissue was sectioned at 200 µm using a Leica VT1000 S vibratome. Tissue clearing was performed as previously described[76] with some modifications. In brief, tissue was immersed in monomer buffer (4% acrylamide and 0.25% ($w/v$) azo-initiator (Wako Pure Chemical Industries) in PBS) and incubated at 4 °C overnight. The solution was transferred to a vacuum tube and bubbled with nitrogen gas for 15 min. The gel was set for 3 h at 37 °C with gentle rotation, after which the tissue was transferred to clearing buffer (8% SDS and 50 mM sodium sulfite in PBS) and cleared at 37 °C until turning semi-transparent. To remove SDS, samples were transferred to the following buffers to wash for 1 h each with rotation: (1) 1% SDS, 0.5% Triton-X in PBS; (2) wash buffer (1% BSA and 0.5% Triton-X in PBS) for two washes; and (3) blocking buffer (4% normal serum, 1% BSA and 0.3% Triton-X in PBS) for two washes. Antibodies were diluted in blocking buffer at the dilutions indicated below. The antibodies used were rabbit anti-pan-cytokeratin (Dako, cat. no. Z0622), anti-K5 (BioLegend, cat. no. poly19055), rat anti-mouse K8/18 (Troma-1; Developmental Studies Hybridoma Bank), rabbit anti-K14 (Abcam, cat. no. EPR17350), rat anti-mouse AIRE (WEHI, clone 5H12), rabbit anti-human/mouse DCLK1 (LSBio, cat. no. LS-C100746) and biotinylated UEA1 lectin (Vector Labs, cat. no. B-1065). The secondary antibodies used were Alexa Fluor 647 donkey anti-rabbit IgG (H+L) (Invitrogen, cat. no. A31573), Alexa Fluor 647 goat anti-rat IgG (H+L) (Invitrogen, cat. no. A-21247) and Alexa Fluor 647 streptavidin conjugate (Invitrogen, cat. no. S21374). After staining, samples were washed in PBS with 0.3% Triton-X. For imaging, samples were incubated in EasyIndex optical clearing solution (refractive index, RI = 1.46) (LifeCanvas Technology) at room temperature until turning fully transparent. A table of antibodies and vendors can be found in Supplementary Table 13.

*Confocal microscopy imaging.* Tissue sections were imaged on a Zeiss LSM 880 confocal microscope using a Plan-Apochromat ×25/0.8 multi-immersion objective at a voxel size of 0.22 µm in $XY$ and 2 µm in $Z$.

*Light-sheet microscopy imaging.* Whole thymic lobes were scanned using a Zeiss Z.1 Light-sheet microscope. The detection objective was an EC Plan-Neofluar ×5/0.16. Stacks were acquired at a resolution of 0.915 µm in $XY$ and approximately 4.9 µm in $Z$. Dual-side images were fused using the maximum intensity option.

*Image presentation.* All images shown are processed using Imaris v.9.7.1 (Bitplane). Regions of interest in tissue sections are presented as 10-µm $Z$-projections.

*Volume calculation of thymic regions.* The total volumes of entire right thymic lobes were calculated using Imaris by generating the lobe surface from the tdTomato channel. Medullary regions were defined by high GFP$^+$ cell density and Krt14$^+$ and HD-TEC regions in aged thymus were defined by compacted GFP$^+$ TECs. To identify medullary and HD-TEC regions in the images, we developed a pipeline in ImageJ (v.2.3.0/1.53f)[77]. For medulla, GFP and keratin 14 channels were combined. For HD-TEC regions, only the GFP$^+$ channel was used. The images were filtered using two-dimensional (2D) median and three-dimensional (3D) Gaussian filtering and then binarized using a 2D min−max filter with thresholds set according to fluorescence intensity. The resulting image was used to extract the medullary or HD-TEC surface. The cortical volume was calculated as total thymic lobe volume minus the medullary and HD-TEC volume.

*Segmentation of TEC nuclei.* TEC nuclei were identified in confocal images of thymus sections using the spot detection function in Imaris. Total TEC spots were filtered and then TEC subsets were segmented according to the shortest distance to the indicated surface or the section edge. Medullary or HD-TECs were defined by the shortest distance to indicated surface ≤0 μm, subcapsular TECs were defined as shortest distance to section edge ≥−25 μm and the remaining spots were defined as cTECs.

Mean cell density was calculated by dividing the number of specific TEC subsets to the volume of different thymic regions.

*Quantification of TECs.* The number of nuclei in the various TEC subsets in the right lobe was calculated by multiplying the mean cell densities ascertained by confocal analysis of the slices from the left lobe by the volumes determined by light-sheet imaging of the right lobe.

**Tissue preparation for sequencing.** The scRNA-seq of FACS-sorted cell suspensions was performed on Chromium instrument (10x Genomics) following the user-guide manual (CG00052 Rev E) and using the Single Cell 3′ Reagent kit (v2). The viability of cells before loading onto the encapsulation chip was 73–98%, as confirmed with 0.2% (*w/v*) Trypan blue stain. Each sample, containing approximately 8,000 cells, was encapsulated in microfluidic droplets at a final dilution of 66–70 cells per μl (a multiplet rate ~3.9%). Following a reverse transcription step, the emulsion droplets were broken, barcoded cDNA purified with DynaBeads and amplified by 12 cycles of PCR: 98 °C for 180 s, 12× (98 °C for 15 s, 67 °C for 20 s, 72 °C for 60 s) and 72 °C for 60 s. The 50 ng PCR-amplified barcoded cDNA was fragmented with the reagents provided in the kit, purified with SPRI beads and the resulting DNA library was ligated to the sequencing adaptor followed by indexing PCR: 98 °C for 45 s; 12 × 98 °C for 20 s, 54 °C for 30 s, 72 °C for 20 s and 72 °C for 60 s. The final DNA library was double-size purified (0.6–0.8×) with SPRI beads and sequenced on an Illumina NovaSeq platform. Sequencing for Foxn1$^{lacz}$ and Foxn1$^{tdTom}$ was performed on an Illumina NextSeq.

Visium spatial gene expression slides were permeabilized at 37 °C for 12–18 min and polyadenylated. Messenger RNA was captured by primers bound to the slides. Reverse transcription, second-strand synthesis, cDNA amplification and library preparation proceeded using the Visium Spatial Gene Expression Slide and Reagent kit (10x Genomics, PN 1000184) according to the manufacturer's protocol. After evaluation by real-time PCR, cDNA amplification included 11–12 cycles; sequencing libraries were prepared with eight cycles of PCR. Indexed libraries were pooled equimolar and sequenced on a NovaSeq 6000 in a PE28/120 run using the NovaSeq 6000 S1 Reagent kit (200 cycles; Illumina).

**Library preparation and sequencing.** After preparing our single-cell suspension solution, we utilized the library preparation and next-generation sequencing services offered by the University of Georgia's Genomics and Bioinformatics Core to generate our scRNA-seq library. Ten thousand thymic stromal cells were loaded onto a 10x Genomics Chromium 3′ Single Cell Gene Expression Solution v3 microfluidics chip (10x Genomics) to generate an Illumina sequencer-ready library. Sequencing was then performed on an Illumina NextSeq 500/550, using four flow lanes that resulted in four BCL files that were shared with us using Illumina's Basespace online platform.

**Computational analysis**
**Mapping of single-cell and spatial transcriptome libraries.** The scRNA-seq FASTQ files were processed with Cell Ranger (v.7.0.1) and Visium libraries were processed with Space Ranger (v.1.3.1) from 10x Genomics. All samples were mapped to the mouse mm10-2020-A genome assembly, except for the *Foxn1*$^{tdTom}$ dataset that was mapped to a custom mouse mm10-2020-A, including the sequences for the tdTomato gene and WPRE element (custom genome FASTA and index files for the tdTomato-WPRE sequence were downloaded from GSE125464).

**Single-cell RNA-seq and spatial transcriptomics quality control and initial analysis.** The Cell Ranger and Space Ranger-generated filtered_feature_bc_matrix.h5 files were processed following the guidelines on the shunPykeR GitHub repository[78], an assembled pipeline of publicly available single-cell analysis packages put in coherent order, which allow data analysis in a reproducible manner and seamless usage of Python and R code. Genes that were not expressed in any cell, and also ribosomal and hemoglobin genes, were removed from downstream analysis. Each cell was then normalized to a total library size of 10,000 reads and gene counts were log-transformed using a pseudo-count of 1. Principal-component analysis (PCA) was applied to reduce noise before data clustering. To select the optimal number of principal components to retain for each dataset, the knee point (eigenvalues smaller radius of curvature) was used as a guide. Leiden clustering[79] was used to identify clusters within the PCA-reduced data.

*CD45$^−$ TBI series.* The quality of the single cells was computationally assessed based on total counts, number of genes and mitochondrial and ribosomal fraction per cell, with low total counts, low number of genes (≤1,000) and high mitochondrial content (≥0.2) as negative indicators of cell quality (Supplementary Fig. 1). Cells characterized by more than one negative indicator were considered as low-quality cells. Although cells were negatively sorted before sequencing for the CD45 marker, a small number of CD45$^+$ cells (expressing *Ptprc*), and also a few parathyroid cells (expressing *Gcm2*), were detected within our dataset (Supplementary Fig. 1). To remove bad-quality cells and contaminants in an unbiased way, we assessed the metrics at the cluster level rather than on individual cells. Leiden clusters with a low-quality profile and/or a high number of contaminating cells were removed. Finally, cells marked as doublets by scrublet[80] were also filtered out. Overall, a total of 12,497 cells, representing 13.3% of our data, were excluded from further analysis (Supplementary Fig. 1).

After removal of low-quality and doublet cells, PCA ($n_{comps}$=45) and unsupervised clustering analysis was applied to the steady-state CD45$^−$ slice of the data using top highly variable genes ($n_{hvgs}$ = 3,500) and using Leiden (resolution = 0.3). Batch effect correction was performed using harmony[81] with default parameters and using sample (Supplementary Fig. 1) as the batch key. Major cell lineages (epithelium, endothelium and fibroblast) were annotated based on canonical markers (Extended Data Fig. 2). Each major lineage was then sliced and reanalyzed in a similar fashion (epithelium, $n_{hvgs}$ = 3,500, $n_{comps}$ = 50, harmony_key = 'sample', resolution = 1.4; endothelium, $n_{hvgs}$ = 3,500, $n_{comps}$ = 30, harmony_key = 'sample', resolution = 0.1; fibroblast, $n_{hvgs}$ = 3,500, $n_{comps}$ = 50, harmony_key = 'sample', resolution = 0.5) to interrogate these linages heterogeneity to a higher degree (Fig. 1a,c). Similarly for the steady-state CD45$^−$ data, the TBI CD45$^−$ slice of the data ($n_{hvgs}$ = 3,500, $n_{comps}$ = 65, harmony_key = 'sample', resolution = 0.7) and their subsequent epithelial ($n_{hvgs}$ = 3,500, $n_{comps}$ = 35, harmony_key = 'sample', resolution = 1.3), endothelial ($n_{hvgs}$ = 3,500, $n_{comps}$ = 35, harmony_key = 'sample', resolution = 0.2) and fibroblast ($n_{hvgs}$ = 3,500, $n_{comps}$ = 30, harmony_key = 'sample', resolution = 0.4) lineages were reanalyzed and annotated separately (Extended Data Fig. 10); however, when highly variable genes were calculated in the TBI setting, the day 1 TBI part of the data was excluded from the calculation due to the presence of a high number of inflammatory response genes. Finally, the steady-state and TBI slice annotations were transferred on the complete dataset ($n_{hvgs}$ = 3,500 (no day 1), $n_{comps}$ = 65, harmony_key = 'sample') shown in Fig. 5i.

*Foxn1$^{tdTom}$ data.* The quality of the single cells was computationally assessed as described for the CD45$^−$ TBI series ($n_{hvgs}$ = 3,500, $n_{comps}$ = 25, harmony_key = 'sample'). A total of 4,062 cells, representing 23.0% of the data were excluded from further analysis. The epithelial cell lineage was sliced and reanalyzed further ($n_{hvgs}$ = 3,500, $n_{comps}$ = 45, harmony_key = 'sample') to allow identification of smaller epithelial

cell populations present in the epithelial compartment of the CD45⁻ TBI series.

*Foxn1^LacZ data.* The quality of the single cells was computationally assessed as described for the CD45⁻ TBI series ($n_{hvgs}$=3,500, $n_{comps}$ = 35, harmony_key = 'sample'). A total of 5,594 cells, representing 40.0% of the data, were excluded from further analysis. The epithelial cell lineage was sliced and reanalyzed further ($n_{hvgs}$ = 3,500, $n_{comps}$ = 45) to allow identification of smaller epithelial cell populations present in the epithelial compartment of the CD45⁻ TBI series.

**Differential expression analysis.** Differential expression analysis for comparisons of interest was performed using the sc.tl.rank_gene_groups() function from scanpy[82] with the Wilcoxon rank-sum method[83]. In all cases, differentially expressed genes (DEGs) were considered statistically significant if the FDR-adjusted *P* value was ≤0.05. No fold change threshold was applied.

**Generation of public gene signatures to characterize our steady-state subsets.** We used the sc.tl.score_genes() function from scanpy[81] (that calculates averaged scores based on cluster specific genes; scores are subtracted with a randomly sampled reference gene set) to generate gene signatures based on markers provided in the literature[23,26,27] (Extended Data Fig. 2 and Supplementary Table 1) to assist annotation of our steady-state epithelial, endothelial and fibroblast subsets in the CD45⁻ TBI series data (Supplementary Fig. 3).

**Mapping of our steady-state subsets onto the TBI, *Foxn1*^tdTom and *Foxn1*^LacZ data.** We used scanpy's sc.tl.score_genes() function with the top 20 DEGs from the steady-state defined subsets (Wilcoxon, FDR ≤ 0.05, sorted in descending order by Wilcoxon z-score; Supplementary Table 3) to generate unique cell type subset signatures, which we mapped to the respective lineage subsets in the TBI (days 1, 4 and 7; Extended Data Fig. 10), *Foxn1*^tdTom and *Foxn1*^LacZ data.

**Public datasets reanalysis.** Re-analysis of single-cell transcriptome datasets from public nonhematopoietic mouse and human thymic samples (CD45⁻ populations) were processed as described in the 'Single-cell RNA-seq and spatial transcriptomics analysis' section of the Methods. The Data Availability section provides a complete list of the raw count data files used as the entry point for each dataset reanalysis.

**ThymoSight.** ThymoSight is an R Shiny app that we have developed to allow interactive exploration of all mouse and human publicly available single-cell datasets of the nonhematopoietic thymic stroma. Mouse datasets included are from refs. 5–7,10–18 and our own data are from this manuscript. Human datasets included are from refs. 9,19,20. ThymoSight also provides dataset metadata fields (if available/applicable) such as tissue, age, stage, sorted cell population, gender, genotype, treatment, linked publication, mapped annotation based on our own subset signatures and original annotation. The app.R code that launches the app together with the Python notebooks used to create consistent annotation fields, reanalyze and integrate the public datasets with ours have been submitted on GitHub (https://github.com/FredHutch/thymosight). The server hosting the interactive app can be accessed at www.thymosight.org.

**Single-cell RNA-seq meta-analysis.** *Integration of CD45⁻ steady-state data with the Visium data.* We used scanorama (v.1.7.2)[84] to integrate our scRNA-seq datasets with our spatial transcriptomic data. Integration was performed between age-matched data at steady state and with default parameters using scanpy's example tutorial (https://scanpy.readthedocs.io/en/stable/tutorials/spatial/integration-scanorama.html#integrating-spatial-data-with-scrna-seq-using-scanorama).

*RNA velocity analysis.* Velocyto (v.0.17.17)[85] was used to generate loom files, which we subsequently merged with our already-annotated single-cell object. We performed RNA velocity analysis within the thymic epithelium compartment of our data using scVelo (v.0.2.4)[86] in stochastic mode.

*Pseudotime analysis.* We used Dynamo's (v.1.4.0)[62] VectorField() function with the given parameters (basis = 'umap', M = 1,000, MaxIter = 170, pot_curl_div = True) to calculate the vector field and to estimate the negative of the single-cell potential (ddhodge potential; Dynamo's version of pseudotime) of the thymic epithelia in 18 mo mice at steady state.

*Cell fate prediction analysis.* We used Dynamo's topography (basis = 'umap') and fate (interpolation_num = 100, direction = 'forward', inverse_transform = false, average = false) functions to create fate prediction animations for our 18-mo epithelial dataset at steady state, setting each of our epithelial subsets as the progenitor population each time. To visualize the fate transition animation results in a static format we leveraged CellRank's (v.2.0.3.dev10+g4ae88b9)[87] built plot_single_flow() module using the already-calculated Dynamo's ddhodge potential (binned) to create vein plots resembling fate transition and relative frequency of the epithelial subsets.

*Pathway enrichment analysis.* Pathway enrichment analysis was performed with GSEA (v.4.3.2)[37] according to the gene list and rank metric provided. The GSEA preranked module was used to predict pathway enrichment in threshold-free comparisons: (1) 18-mo versus 2-mo subsets at steady-state and (2) aaTEC1s and aaTEC2s versus other TECs. We created rankings for all DEGs using the Wilcoxon z-score in descending order. Predicted pathways with an FDR ≤ 0.05 were considered significantly enriched.

*Network analysis.* Network analysis of the significantly enriched GSEA pathways from comparisons of interest was performed using Cytoscape (v.3.10.0)[38]. We used the EnrichmentMap() function to organize enriched pathways (FDR ≤ 0.05) with a high overlap of genes (default cutoff similarity of 0.375) in the same network allowing for a simplified and intuitive visualization of the distinct processes that are significantly represented in each subset at steady state. This facilitated interpretation of the enriched pathways from the plethora of comparisons and allowed categorization of all resulting pathways into networks based on the overlap of the genes contributing to the pathway's enrichment (Supplementary Table 5). Manual inspection of the resulting networks allowed allocation of network-related annotations. Individual pathways that were not part of an existing network were manually annotated to the existing categories based on their biological function or grouped under 'singlet'.

*Cell–cell interaction analysis.* CellChat (v.1.4.0)[50] was used with default parameters to predict cell–cell interactions between all CD45⁻ subsets within the 2-mo and 18-mo cohorts at steady state and at days 1, 4 and 7 after damage against the complete CellChat database. Cell subsets with fewer than 20 cells were excluded from the interactome analysis. For comparisons between the individual TBI time points and age cohorts, individual CellChat objects were integrated using the mergeCellChat() function. For the circus plots shown in Fig. 4h, some of the cell type subsets were grouped together: ECs (aEC, capEC and vEC), FBs (capsFB, intFB, medFB, nmSC and Fat), mTEC^prol/mTEC2s and mimetic (basal, tuft, neuro, goblet and M cell).

**CD45⁻ bulk RNA-seq preprocessing and downstream data analysis.** *Quality control, alignment and gene count quantification.* Quality control of the raw read files (FASTQ) was performed using the FastQC tool[88]. Low-quality reads and adaptor contaminants were removed

using Trimmomatic[89] (default parameters for paired-end reads) and post-trimmed reads were reassessed with FastQC to verify adaptor removal and potential bias introduced by trimming. The quality control-approved reads were aligned to the GRCm38.p5 (mm10) mouse genome assembly (GENCODE; M12 release) with STAR[90] aligner using default parameters and –runThreadN set to 32 to increase execution speed. The STAR-aligned files were then used as input to the featureCounts[91] tool (default parameters) to quantify gene expression levels and construct the count matrix.

*Low gene count removal and library size normalization.* The raw count matrix was converted to a DGEList object in R using the readDGE() function from the edgeR[92] package. Lowly expressed genes were removed using the filterByExpr() function for the groups of interest before comparison with the scRNA-seq datasets.

*Bulk RNA-seq versus scRNA-seq.* scRNA-seq sample reproducibility was verified using bulk RNA-seq data for the CD45⁻ sorted populations. Comparison between bulk and scRNA-seq CD45⁻ transcriptional profiles was performed by computing Pearson's correlation between $\log_{10}$-transformed raw bulk counts (per biological replicate) and $\log_{10}$-transformed averaged raw single-cell counts (per technical replicate) for the relevant datasets across the TBI timeframe (Supplementary Fig. 2).

**Statistics and reproducibility.** All statistics were calculated, and display graphs were generated, in GraphPad Prism.

Specific statistical tests used have been highlighted in the figure legends but briefly, statistical analysis between two groups were performed on biological replicates (individual mice) with a two-tailed Mann–Whitney or two-tailed *t*-test and, where appropriate, a two-tailed paired *t*-test. Statistical comparison between three or more groups was performed on biological replicates (individual mice) with a Kruskall–Wallis test with Dunn's correction, one-way analysis of variance with Dunnett's correction or two-way analysis of variance with Šídák correction.

The imaging studies in Fig. 2a,b were performed independently three times (*n* = 1–3 mice per experiment). For the images in Fig. 2e, PanK was performed independently three times (*n* = 1–3 mice per experiment), Krt5 was performed independently four times (*n* = 1–3 mice per experiment), Krt8 was performed once (*n* = 2 mice), Krt14 was performed independently seven times (*n* = 1–3 mice per experiment) and claudin-3 was performed independently four times (*n* = 1–4 mice per experiment); with one section imaged for each mouse. The studies described in Fig. 3i were performed independently ten times (*n* = 1–3 mice per experiment) with one section imaged per animal. Figure 3b was performed independently three times (*n* = 4 per group). In Extended Data Fig. 7b, staining for FOXN1 was performed independently twice (*n* = 7 per experiment). In Extended Data Fig. 9a, DCLK and UEA1 were performed once (*n* = 2 mice per experiment) with one section imaged per animal.

### Reporting summary
Further information on research design is available in the Nature Portfolio Reporting Summary linked to this article.

### Data availability
Sequencing data generated in this study have been deposited in NCBI's Gene Expression Omnibus and can be accessed through the SuperSeries accession number GSE240020. For ThymoSight, accession numbers for the publicly available raw count data that have been reanalyzed for this study are provided here. For public mouse data, please see Kernfeld et al. (2018) (GSE107910)[10]; Bornstein et al. (2018) (GSE103967)[6]; Dhalla et al. (2019) (https://www.ebi.ac.uk/biostudies/arrayexpress/studies/E-MTAB-8105#)[7]; Baran-Gale et al. (2020)

(https://bioconductor.org/packages/release/data/experiment/html/MouseThymusAgeing.html)[5]; Wells et al. (2020) (GSE137699)[11]; Rota et al. (2021) (GSE162668)[12]; Nusser et al. (2022) (GSE106856)[8]; Michelson et al. (2022) (GSE194253)[13]; Klein et al. (2023) (GSE215418)[15]; Farley et al. (2023) (GSE232765)[16]; Givony et al. (2023) (GSE236075)[17]; Michelson et al. (2023) (GSE225661)[14]; and Horie et al. (2023) (GSE228198)[18]. For public human data please see Park et al. (2020) (https://zenodo.org/records/3711134)[9]; Bautista et al. (2021) (GSE147520)[19]; and Ragazzini et al. (2023) (GSE220830, GSE220206 and GSE220829)[20]. Re-analyzed public datasets with added metadata can be accessed at Zenodo at https://doi.org/10.5281/zenodo.12516405 (ref. 93).

### Code availability
The shunPykeR adapted Jupyter notebooks, R notebooks and the assorted conda and renv files to reproduce analyses and figure creation for this manuscript can be found on GitHub at https://github.com/kousaa/Kousa-et-al-2024-NI. The app.R code that launches the ThymoSight app, together with the Python notebooks used to create consistent annotation fields, reanalyze and integrate the public datasets with ours have been submitted on GitHub at https://github.com/FredHutch/thymosight. The server hosting the interactive app can be accessed at www.thymosight.org.

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

## Acknowledgements

We thank K. Manova-Todorova, E. Chan, R. Junka, N. Fan and A. Barlas from the Molecular Cytology Core Facility at Memorial Sloan Kettering Cancer Center for their support and expert advice in the preparation and analysis of immunohistochemistry data. We thank D. Tenenbaum and the Scientific Computing group at Fred Hutchinson Cancer Center for help with deployment of the ThymoSight Shiny application. We also thank the Flow Cytometry and Comparative Medicine Core Facilities, as well as support of the Immunotherapy Integrated Research Center at the Fred Hutchinson Cancer Center. We are also grateful for the support of the WEHI Centre for Dynamic Imaging, Flow Cytometry facility and Bioservices facility, in particular L. Whitehead, P. Rajasekhar, T. Boudier, S. Monard, Z. Arnold, H. Marks, T. Ballinger and S. Holloway. Finally, we thank H.-R. Rodewald for the provision of the anti-FOXN1 antibody. This research was supported by National Institutes of Health award numbers R01-CA228358 (M.R.M.vdB.), R01-CA228308 (M.R.M.vdB.), R01-HL123340 (M.R.M.vdB.), R01-HL147584 (M.R.M.vdB.), P01-CA023766 (M.R.M.vdB.), P01-AG052359 (J.A.D. and M.R.M.vdB.), R35⁻HL-171556 (J.A.D.), R01-HL145276 (J.A.D.), R01-HL165673 (J.A.D.), U01-AI70035 (J.A.D.); as well as the NCI Cancer Center Support Grants P30-CA015704 (Fred Hutchinson Cancer Center) and P30 CA008748 (Memorial Sloan Kettering Cancer Center). D.H.D.G. was funded by Australian National Health and Medical Research Council (NHMRC) Fellowships 1090236 and 1158024; NHMRC Project/Ideas Grants 1078763, 1121325 and 1187367 and Cancer Council of Victoria Grants-in-Aid 1146518 and 1102104 and through the Victorian State Government Operational Infrastructure Support and Australian Government NHMRC IRIISS. M.R.M.vdB. also received support from the Starr Cancer Consortium, the Tri-Institutional Stem Cell Initiative, The Lymphoma Foundation, The Susan and Peter Solomon Divisional Genomics Program, Cycle for Survival and the Parker Institute for Cancer Immunotherapy. J.A.D. received additional support from a Scholar Award from the American Society of Hematology; the Mechtild Harf (John Hansen) Award from the DKMS Foundation for Giving Life; the Cuyamaca Foundation; and the Bezos Family Foundation. L.J. received support from the European Molecular Biology Organization (ALTF-431-2017) and the MSK Sawiris Foundation. K.Z. is supported by a University of Melbourne Research Training Program Scholarship. D.A. received support from T32-GM007270 and D.G. was supported by F30-HL165761.

## Author contributions

A.F., D.A., E.L. and K.A. contributed equally. A.I.K., L.J., K.Z., E.V., D.H.D.G., J.A.D. and M.R.M.vdB. conceived and designed the study. N.R.M., K.L.R., D.H.D.G., M.R.M.vdB. and J.A.D. acquired funding. L.J. and A.I.K. planned research activities. A.I.K., L.S., L.J. and L.M. curated data. A.I.K., L.S. and L.J. performed initial analysis. A.I.K., L.S., L.J., M.S., R.S. and A.L.C.G. provided code and performed computational and statistical analysis of single-cell and bulk RNA-seq data. A.I.K. created the R shiny interactive app and K.C. assisted with website development. L.J., K.Z., J.T., E.V., K.V.A., F.M. and M.D.'A. analyzed flow cytometric data. A.E.F. and N.R.M. provided scRNA-seq on *Foxn1*^tdTom and *Foxn1*^Z/Z. K.V.A. and L.H. analyzed and interpreted H&E sections. L.H. analyzed and provided human imaging. L.J., K.Z., A.E.F., D.A., E.L., D.G., K.C., J.M.S., E.V., K.V.A., J.T., A.L., K.N., N.L., R.G., F.M., H.A., S.Y., M.B.dS., M.D., V.C.W., K.L.R., S.D.W. and B.G. planned and performed experiments. D.P., V.C.W., N.R.M., D.H.D.G., M.R.M.vdB. and J.A.D. supervised study. L.J., A.I.K., K.Z., S.Y., D.H.D.G. and J.A.D. prepared figures and visualized data. L.J., A.I.K., K.Z., D.H.D.G., J.A.D. and M.R.M.vdB. wrote the initial draft. All authors reviewed and approved the manuscript.

## Competing interests

M.R.M.vdB. has received research support and stock options from Seres Therapeutics and stock options from Notch Therapeutics and Pluto Therapeutics; has consulted, received honorarium from or participated in advisory boards for Seres Therapeutics, WindMIL Therapeutics, Rheos Medicines, Merck & Co, Magenta Therapeutics, Frazier Healthcare Partners, Nektar Therapeutics, Notch Therapeutics, Forty Seven, Priothera, Ceramedix, Lygenesis, Pluto Therapeutics, GlaskoSmithKline, Da Volterra, Vor BioPharma, Novartis (Spouse), Synthekine (Spouse) and Beigene (Spouse); he has IP Licensing with Seres Therapeutics and Juno Therapeutics; and holds a fiduciary role on the Foundation Board of DKMS (a nonprofit organization). J.A.D. and M.R.M.vdB. are founders of, and receive stock options, from ThymoFox; and both have received royalties from Wolters Kluwer. The Walter and Eliza Hall Institute of Medical Research receives milestone and royalty payments related to venetoclax. Employees are entitled to receive benefits related to these payments; D.H.D.G. reports receiving benefits. D.H.D.G. has received research funding from Servier. The remaining authors declare no competing interests.

## Additional information

**Extended data** is available for this paper at https://doi.org/10.1038/s41590-024-01915-9.

**Correspondence and requests for materials** should be addressed to Daniel H. D. Gray, Marcel R. M. van den Brink or Jarrod A. Dudakov.

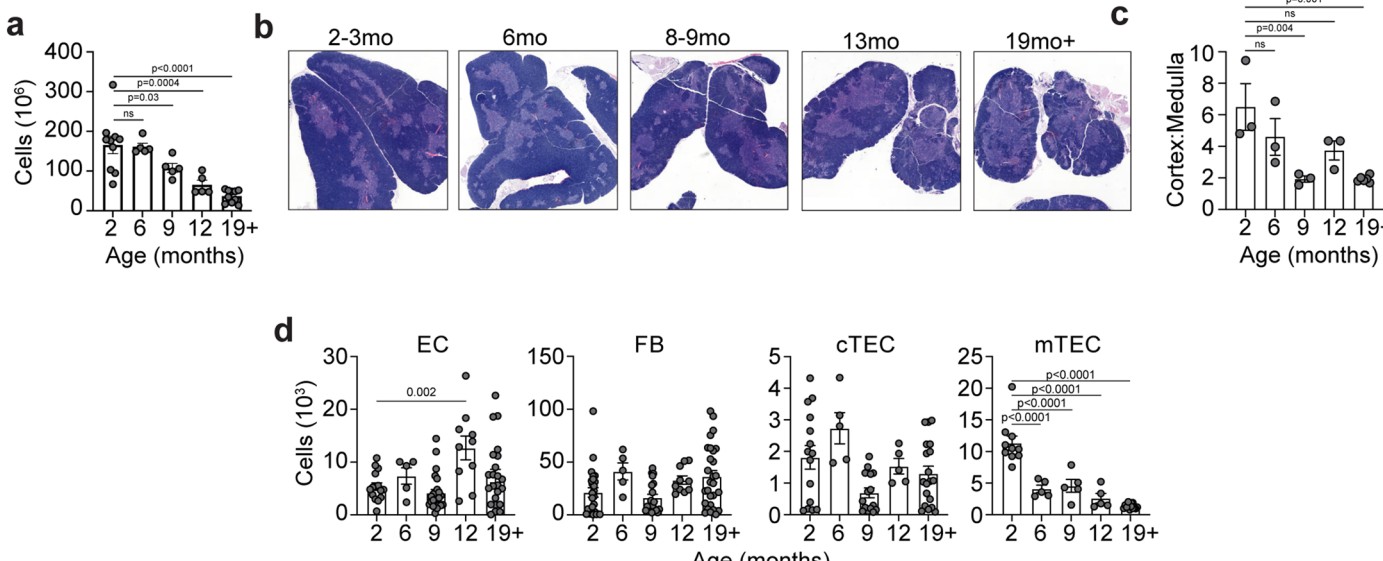

**Extended Data Fig. 1 | General stromal features of thymic aging. a,** Thymic cellularity of female C57BL/6 mice at 2, 6, 9, 12 or 19+ months of age. **b-c,** Representative images of hematoxylin and eosin (H&E) stained mouse thymi from 2, 6, 8-9, 13, and 19+mo female C57BL/6 mice (**b**) used to calculate ratio of cortical (dark) to medullary (light) region (**c**). In (**c**), each dot represents a biological replicate. **d,** Flow cytometric analysis of enzymatically digested thymus and absolute cell numbers for major cell types (TECs; cTECs and mTECs; ECs and FBs) in 2, 6, 9, 12, and 18+ mo female C57BL/6 mice. Summary data represents mean ± SEM; each dot represents an individual biological replicate; statistics were generated for **a**, **c**, and **d** using one-way ANOVA with the Dunnett correction for multiple comparisons.

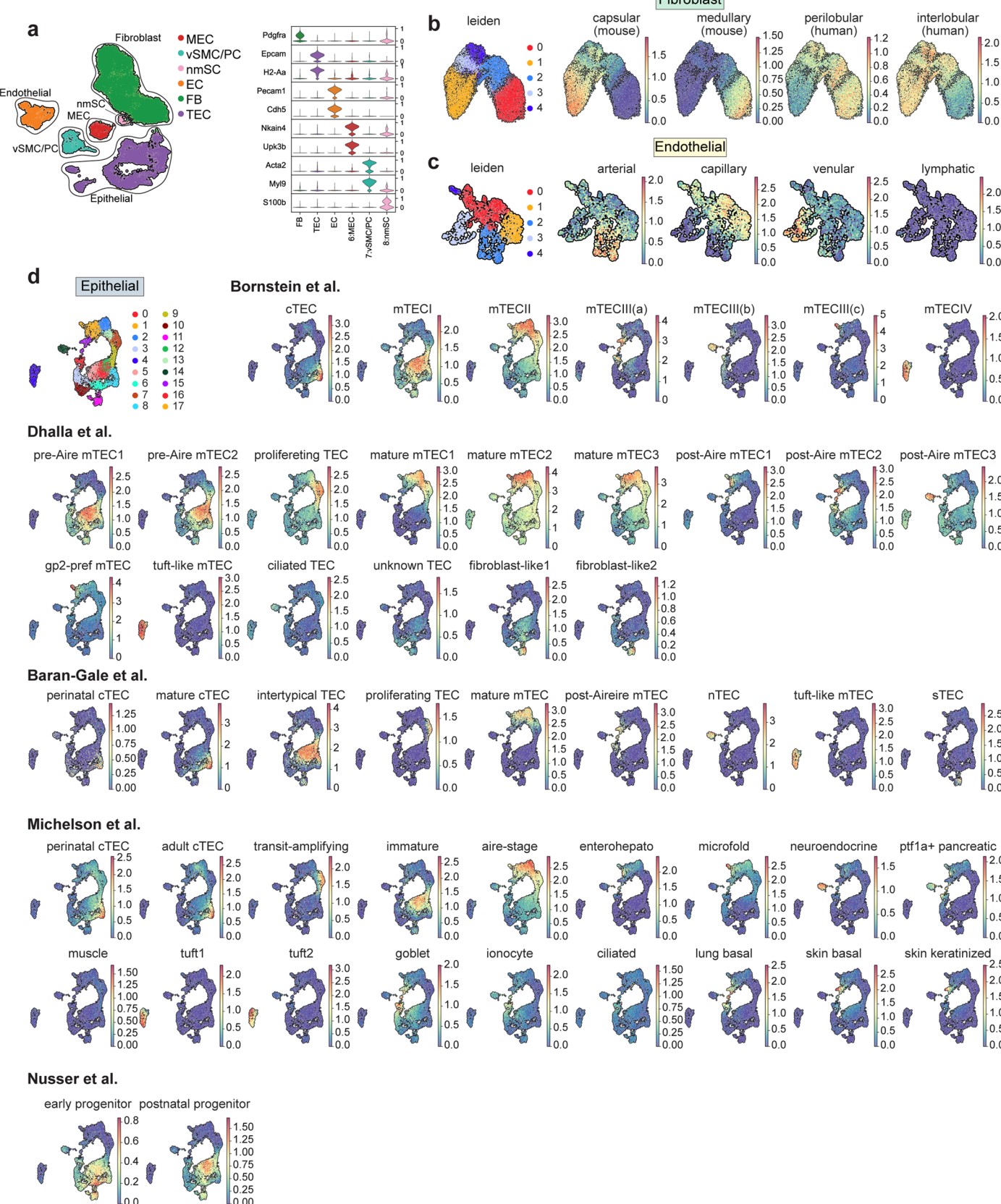

**Extended Data Fig. 2 | Mapping of pre-existing thymic stromal sequencing datasets. a**, Broad structural cell subsets were annotated based on expression of canonical markers such as *Pdgfra, Epcam, H2-aa, Pecam*, and *Cdh5*. **b**, Leiden clustering of our fibroblast population (n$_{FB}$=13,240) and signatures for murine capsular-medullary and human perilobular-interlobular fibroblasts based on previously published datasets[9,26]. **c**, Leiden clustering of our endothelial population (n$_{EC}$=1,661) and signatures for arterial, capillary, venular, and lymphatic endothelial cells based on previously published datasets[27]. **d**, Leiden clustering of our thymic epithelial population (n$_{TEC}$=6,175) and signatures of previously published literature and overlaid on our sequencing dataset[5–7,13].

**a** THYMOSIGHT (INTEGRATED MOUSE DATASET - ALL CELLS)

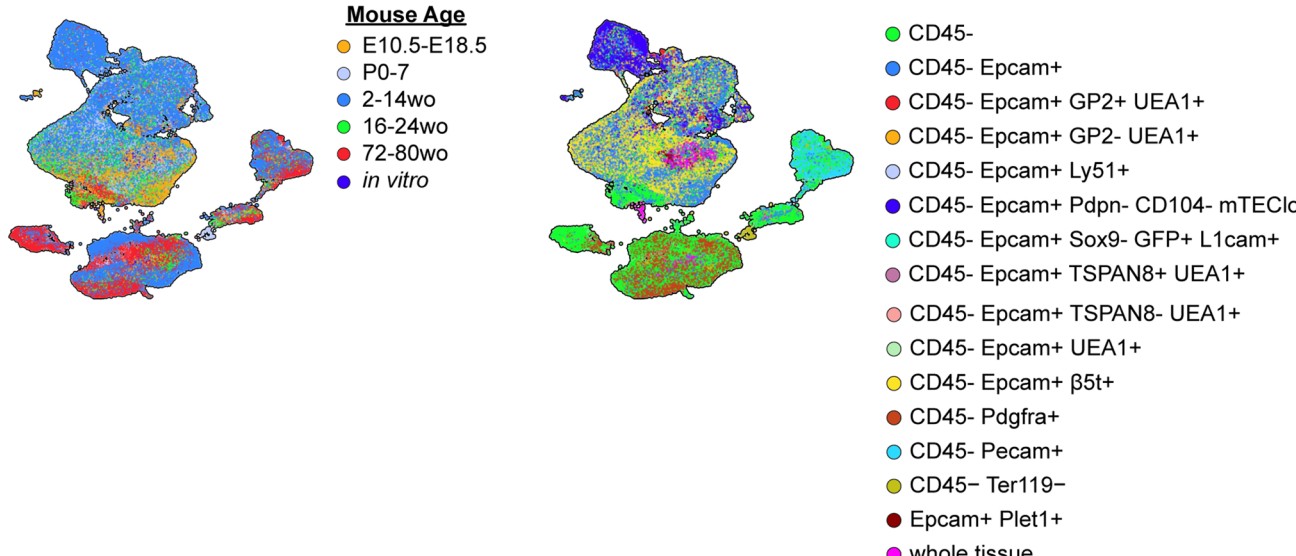

**b** THYMOSIGHT (INTEGRATED MOUSE DATASET - EPITHELIAL)

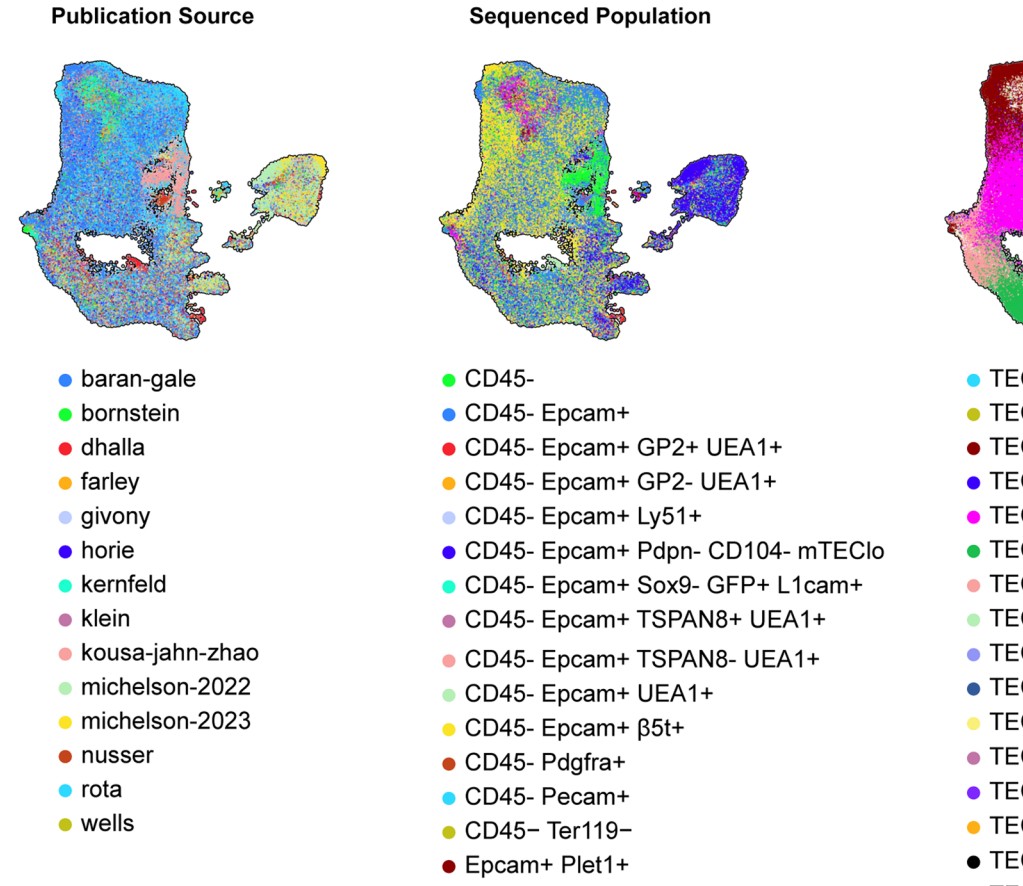

**Extended Data Fig. 3 | Thymosight: Integration of thymic sequencing datasets. a-b**, UMAPs of (**a**) all mouse non-hematopoietic thymic stroma cells (ThymoSight integration of public data[5–18] and ours; n = 297,988) annotated by age and sequenced population, and (**b**) all mouse thymic epithelial cells (n = 205,625; subset of CD45− ThymoSight data) annotated by publication source, sequenced population, and TEC subset.

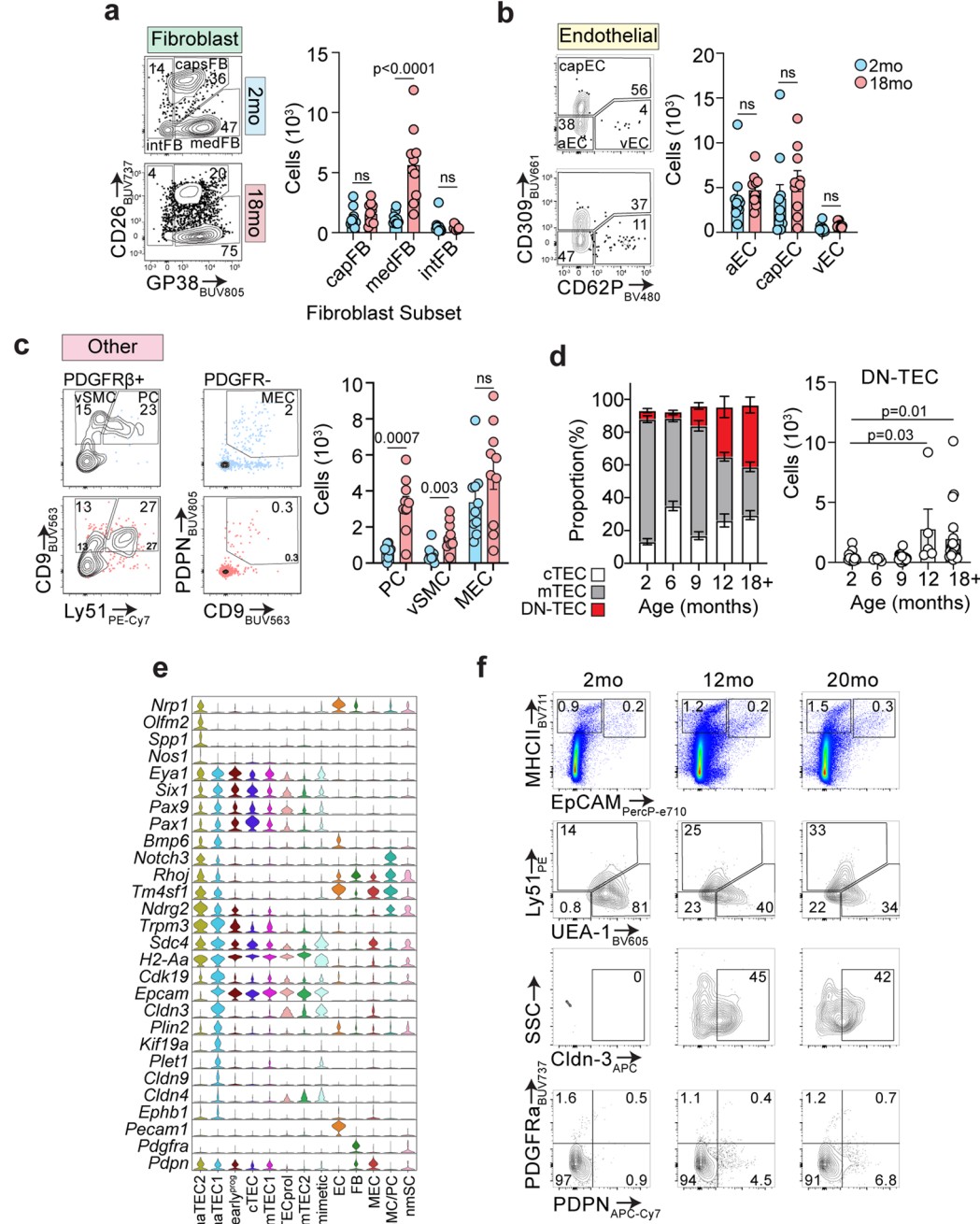

**Extended Data Fig. 4 | Quantification of non-epithelial stromal cells subsets and aaTEC gating. a–c**, Concatenated flow cytometry plots and quantities for cell populations within the fibroblast (**a**), endothelial (**b**), and "other" (**c**) cell lineages in 2-mo (n = 10) and 18-mo (n = 10) mice. **c**, Concatenated flow cytometry plots and quantities for pericytes (PC), vascular smooth muscle cells (vSMC) and mesothelial cells (MEC) (n = 10/age). **d**, Frequency and numbers of DN-TEC across lifespan: 2-mo (n = 14), 6-mo (n = 5), 9mo (n = 15), 12-mo (n = 5), and 18+mo (n = 18). **e**, Violin plots with extensive list of aaTEC1 and aaTEC2 markers. **f**, Gating strategy for aaTECs. aaTEC1 were first gated on CD45⁻TER119⁻ then PDGFRα⁻

CD31⁻ cells. EpCAM⁺MHCII⁺ cells were gated as the whole TEC compartment, then mTECs and cTECs were excluded by taking the UEA1⁻Ly51⁻ double negative fraction and gating on CLDN3. aaTEC2 were also first gated on CD45⁻TER119⁻ then PDGFRα⁻CD31⁻ cells. EpCAM⁻MHCII⁺ cells were then gated and PDPN⁺PDGFRβ⁻ were classed at aaTEC2. Summary data represents mean ± SEM; each dot represents an individual biological replicate. Statistics for **a–c** were generated using two-tailed Mann–Whitney tests comparing within individual populations and for **d** using the Kruskal–Wallis test with Dunns correction.

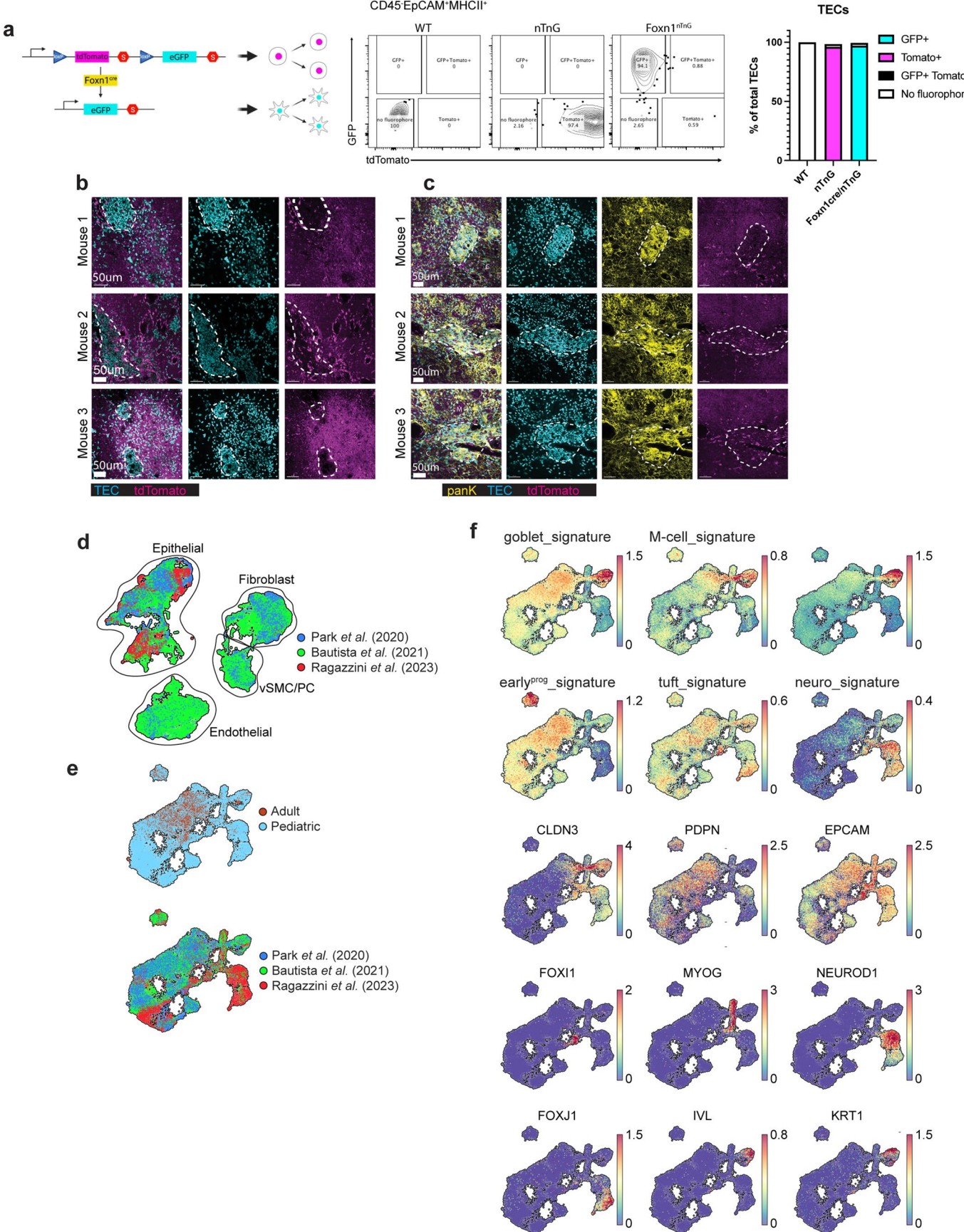

**Extended Data Fig. 5 | See next page for caption.**

**Extended Data Fig. 5 | Validation of aaTEC identification and imaging.**
**a**, Generation of *Foxn1^nTnG mice*. ROSA^nT-nG (nT/nG) mice[73] were intercrossed with *Foxn1^Cre* mice[74]. Representative flow cytometric plots of TEC from 11 weeks old WT, nT/nG and *Foxn1^nTnG* show specific detection of GFP in nearly all TEC only in the latter strain. Quantification of the relative proportions of TEC expressing the reporters are shown in the bar graph on the right (n = 2 to 3 from 2 experiments). **b**, Representative confocal images of thymic sections from 12-mo *Foxn1^nTnG* mice with high-density TECs located in peri-medullary region. **c**, Representative

confocal images of thymic sections from 12-mo *Foxn1^nTnG* mice stained with anti-pan-keratin, with high-density TEC regions highlighted. **d**, UMAPs integrating all human CD45⁻ non-hematopoietic cells from published datasets[9,19,20] (ThymoSight) annotated by age and dataset (n = 115,536). **e**, UMAPs integrating all human thymic epithelial cells from public datasets[9,19,20] (ThymoSight) annotated by age and dataset (n = 40,144). **f**, Signatures of our mouse epithelial cell subsets (Supplementary Table 3), including aaTEC, overlaid onto the integrated human TEC data derived from[9,19,20].

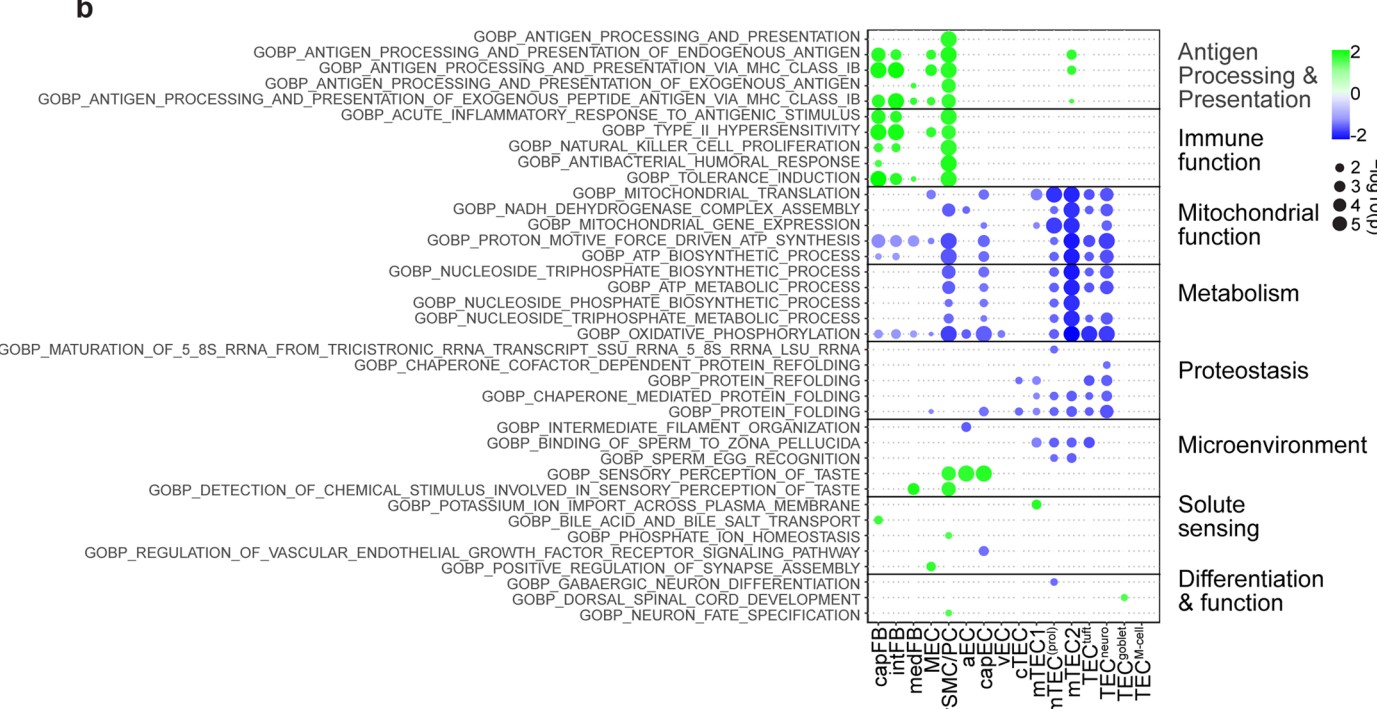

**Extended Data Fig. 6 | Age related changes in gene expression in thymic stromal cells. a**, Differential expression of key epithelial genes and thymopoietic factors with age. **b**, As in Fig. 4a, GSEA pathway analysis was performed for each subset based on differentially expressed genes within each population between 2-mo and 18-mo mice (Supplementary Table 4-5) and Cytoscape network analysis was used to integrate enriched pathways (FDR≤0.05) sharing a core set of genes. Dotplot of top 5 pathways within each category. Individual pathways are listed.

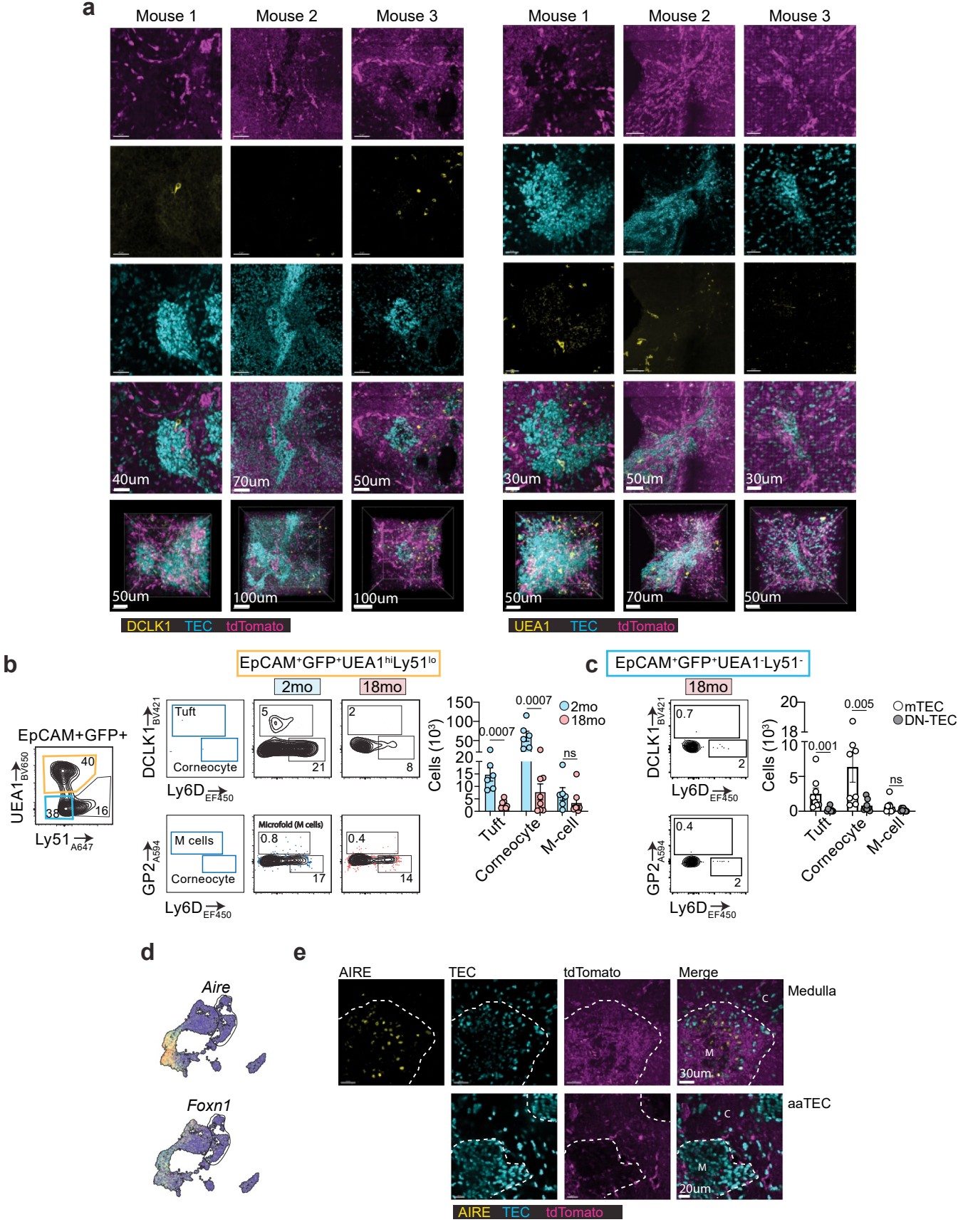

**Extended Data Fig. 7 | See next page for caption.**

**Extended Data Fig. 7 | Comparison of aaTECs with conventional TECs and mimetic cells. a**, 3D reconstruction and representative images of high-density TEC region from 12-mo *Foxn1^{nTnG}* mice stained with DCLK1 or UEA1 to highlight tuft cells and M-like cells, respectively. **b**, Flow cytometry plots showing proportion of selected mimetic cells (tuft, corneocyte and M-cells) in 2-mo (n = 6) and 18-mo (n = 8) *Foxn1^{nTnG}* mice. Mimetic cells were first gated on EpCAM⁺GFP⁺ cells, then mTECs (UEA1^{hi}Ly51^{lo}) were assessed for the mimetic cell markers DCLK1 (tuft cells), GP2 (microfold cells), and Ly6D (corneocytes). Bar graph shows quantification of mimetic cell numbers. **c**, Flow cytometry

plots showing mimetic cell frequency in 18-mo *Foxn1^{nTnG}* mice (n = 8) gated on EpCAM⁺GFP⁺UEA1⁻Ly51⁻ DN-TECs. Bar graph shows quantification of mimetic cells comparing mTECs (as in Extended Data Fig. 7b) and DN-TECs. **d**, *Aire* and *Foxn1* expression in TEC subsets. **e**, Representative confocal images of thymic sections from 12-mo *Foxn1^{nTnG}* mice stained with anti-AIRE, with the medulla or high-density TECs highlighted. Scale bar: 50μm. Summary data represents mean ± SEM; each dot represents an individual biological replicate. Statistics for **b-c** were generated using two-tailed Mann–Whitney tests comparing within individual mimetic cell subsets.

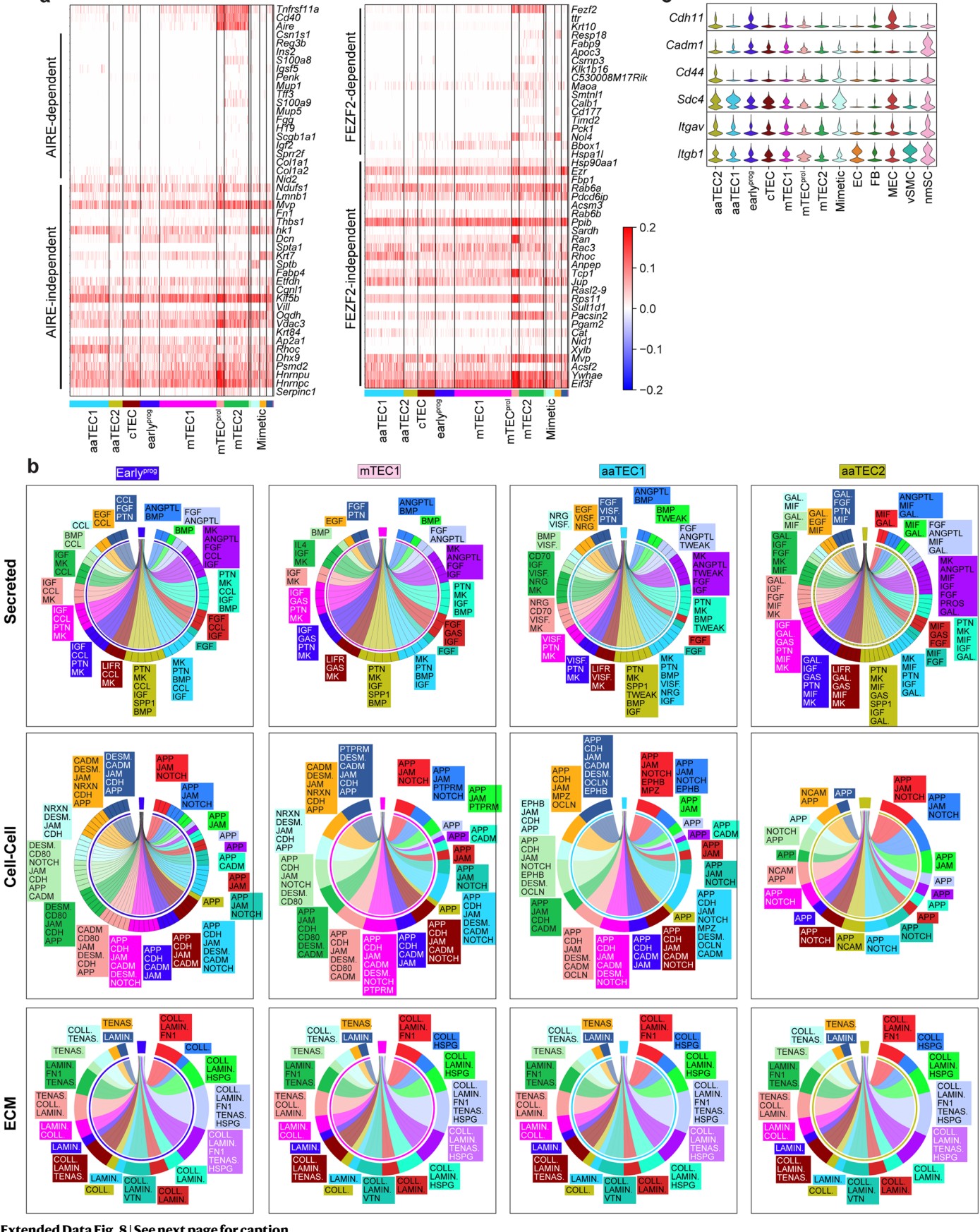

**Extended Data Fig. 8 | See next page for caption.**

**Extended Data Fig. 8 | aaTEC function and interactome with age. a,** As in Fig. 4g, heatmap of AIRE- (left) and FEZF2- (right) dependent/independent genes from[72]. Heatmap shows scaled normalized gene expression. Individual genes are listed. **b,** As in Fig. 4i, CellChat chord diagrams showing outgoing signals from all stromal cell populations towards early[prog], mTEC1, aaTEC1 or aaTEC2 cellular receivers. Chord diagrams are color-coded by the sender population and split by the type of CellChat signaling (secreted, cell-cell and ECM). Specific outgoing signals per sender are listed in color-matching boxes on the side of each plot. **c,** Violin plot of receptors for putative EMT factors Midkine and Pleiotrophin.

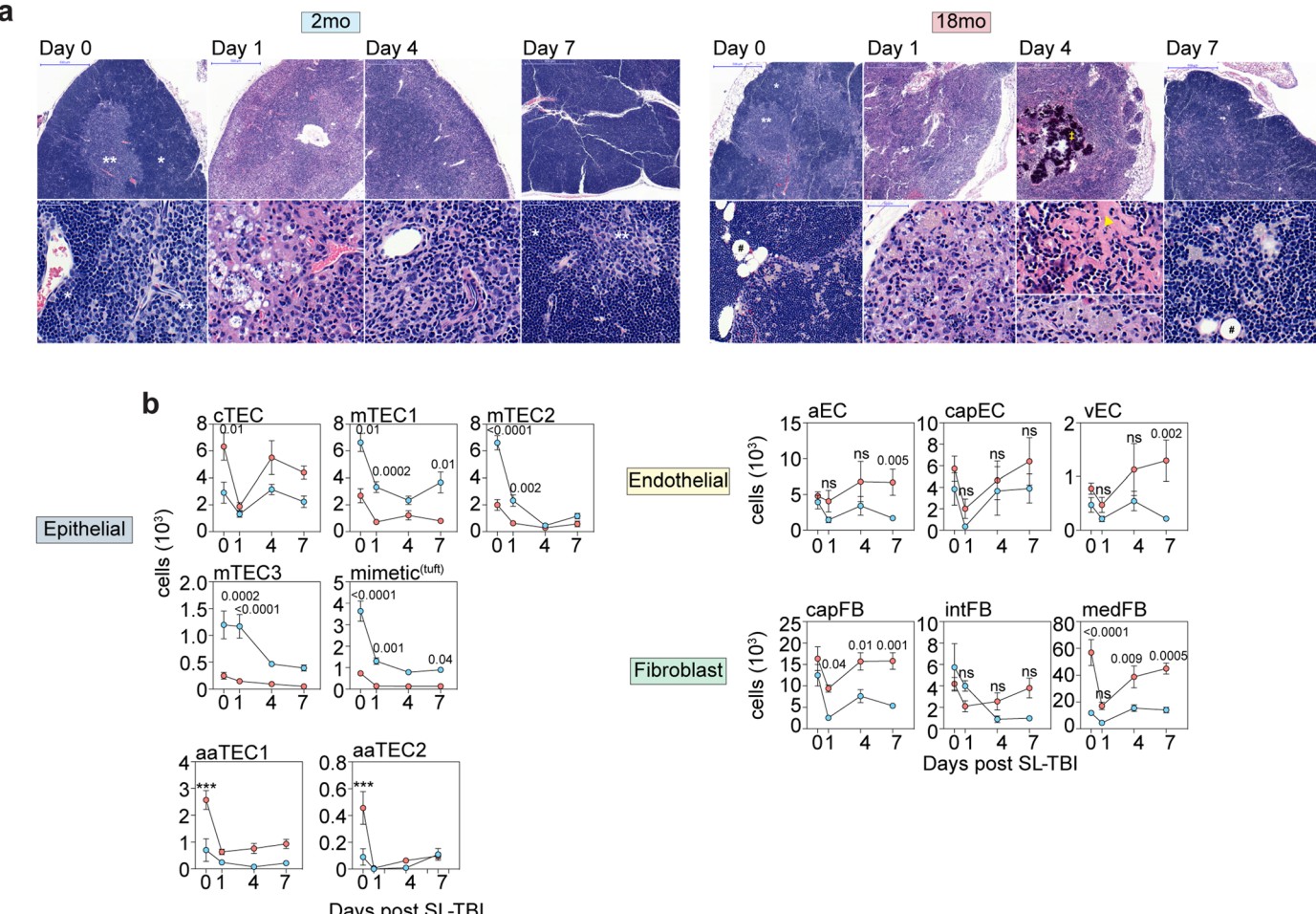

**Extended Data Fig. 9 | Acute thymic recovery following ionizing radiation.**
**a**, Morphologic alterations in the context of acute thymic involution after TBI and thymic reconstitution in 2-mo and 18-mo mice. Top and bottom rows represent low- and high-power images from each timepoint, respectively. Annotations: cortex (*), medulla (**), adipocytes (#), areas of dystrophic calcification (†), areas of dense fibrosis (arrowhead). **b**, Kinetics of recovery for the epithelial, endothelial and fibroblast defined subsets on day 0 (n = 10, 2-mo and 10, 18-mo),

1 (n = 10, 2-mo and 10, 18-mo), 4 (n = 10, 2-mo and 10, 18-mo) and 7 (n = 10, 2-mo and 10, 18-mo) after TBI in 2-mo and 18-mo mice. Total cellularity for each subset. Statistics compare across ages for each timepoint. Asterisks denote when recovery in either age cohort on a specific timepoint was significantly different to the other cohort. Summary data represents mean ± SEM. Statistics for **b** were generated using a two-way ANOVA with Šídák correction.

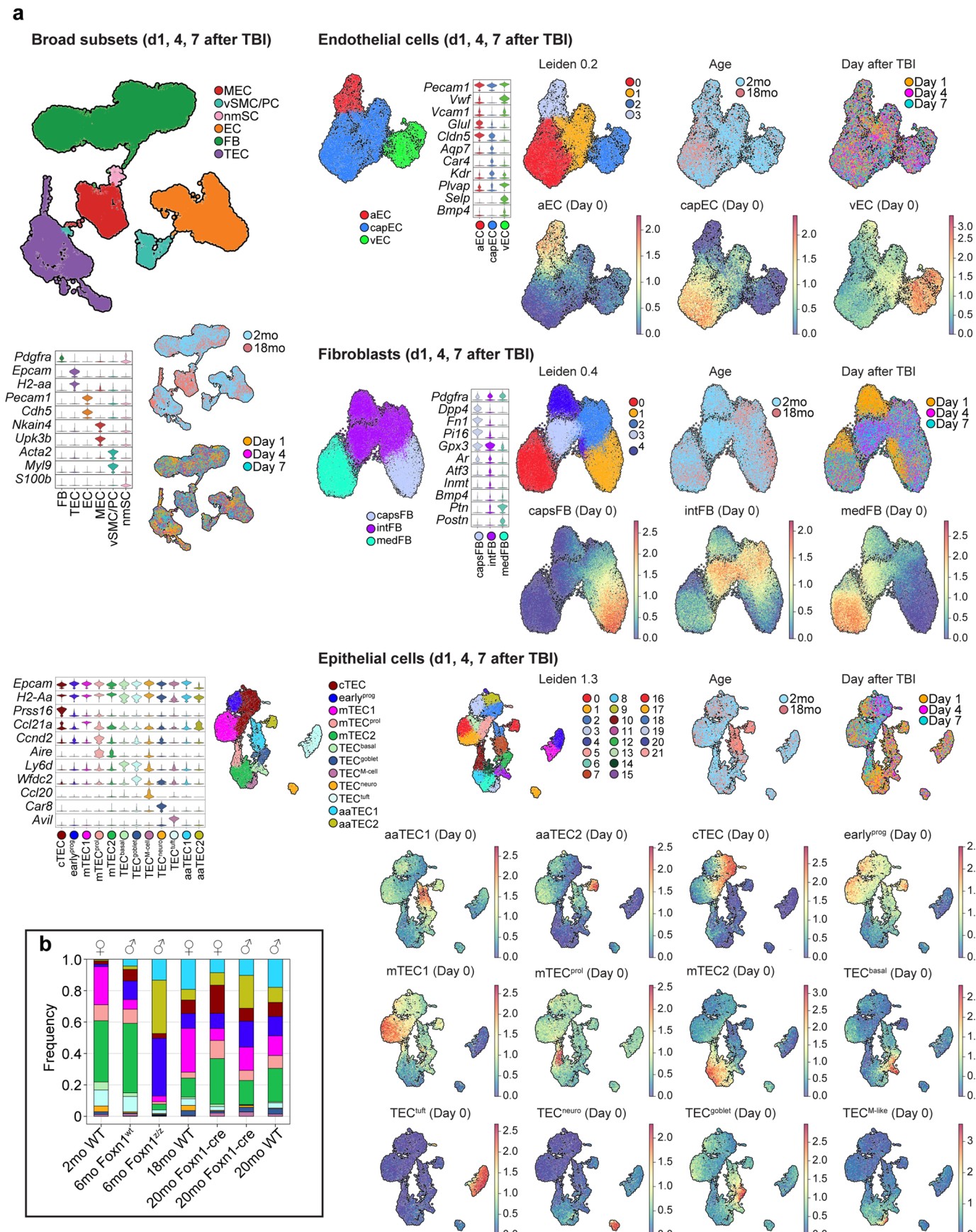

**Extended Data Fig. 10 | See next page for caption.**

**Extended Data Fig. 10 | Mapping of pre-existing thymic stromal sequencing datasets after acute damage. a**, Broad structural cell subsets were annotated based on expression of canonical markers such as *Pdgfra, Epcam, H2-aa, Pecam*, and *Cdh5* and steady-state subset signatures (top 20 marker genes). Leiden clustering and cell subset signatures derived from Supplementary Table 3 across endothelial, fibroblast and epithelial cells isolated at days 1, 4, and 7 after TBI. **b**, Stacked barplots comparing TEC subsets frequency in the 6-mo *Foxn1$^{Z/Z}$* mice and controls to our own TEC subsets in 2-mo and 18-mo wild-type mice at steady state. ♀: female mice, ♂: male mice.

Daniel Gray,
Marcel van den Brink

# Reporting Summary

## Statistics

For all statistical analyses, confirm that the following items are present in the figure legend, table legend, main text, or Methods section.

| n/a | Confirmed | |
|---|---|---|
| ☐ | ☒ | The exact sample size (*n*) for each experimental group/condition, given as a discrete number and unit of measurement |
| ☐ | ☒ | A statement on whether measurements were taken from distinct samples or whether the same sample was measured repeatedly |
| ☐ | ☒ | The statistical test(s) used AND whether they are one- or two-sided<br>*Only common tests should be described solely by name; describe more complex techniques in the Methods section.* |
| ☐ | ☒ | A description of all covariates tested |
| ☐ | ☒ | A description of any assumptions or corrections, such as tests of normality and adjustment for multiple comparisons |
| ☐ | ☒ | A full description of the statistical parameters including central tendency (e.g. means) or other basic estimates (e.g. regression coefficient) AND variation (e.g. standard deviation) or associated estimates of uncertainty (e.g. confidence intervals) |
| ☒ | ☐ | For null hypothesis testing, the test statistic (e.g. *F*, *t*, *r*) with confidence intervals, effect sizes, degrees of freedom and *P* value noted<br>*Give P values as exact values whenever suitable.* |
| ☒ | ☐ | For Bayesian analysis, information on the choice of priors and Markov chain Monte Carlo settings |
| ☒ | ☐ | For hierarchical and complex designs, identification of the appropriate level for tests and full reporting of outcomes |
| ☒ | ☐ | Estimates of effect sizes (e.g. Cohen's *d*, Pearson's *r*), indicating how they were calculated |

*Our web collection on statistics for biologists contains articles on many of the points above.*

## Software and code

Policy information about availability of computer code

| Data collection | Cells were sorted on an Aria II (BD Biosciences) using FACSDiva (BD Biosciences, Franklin Lakes, NJ). Flow cytometry analysis was performed on a Fortessa X-50 or Symphony A6 using FACSDiva Software (BD Biosciences). Sequencing data were collected on Illumina Nextseq or Novaseq platforms. Tissue sections were imaged on a Zeiss LSM 880 confocal microscope. Whole thymic lobes were scanned using a Zeiss Z.1 Lightsheet microscope. |
|---|---|
| Data analysis | #1. Flow cytometry data were analyzed using FlowJo (v10.7.1, TreeStar Software) and GraphPad Prism (v10; GraphPad Software, LLC). Data was collected and cells sorted on BD cytometers using FACSDiva software.<br><br>#2. All single cell RNA-seq data were analyzed using Cell Ranger (v7.0.1; 10x Genomics). The shunPykeR adapted Jupyter notebooks, R notebooks, and the assorted conda (.yml) and renv (renv.lock) environment files to reproduce analyses and figure creation for this manuscript can be found at https://github.com/kousaa/Kousa-et-al-2024-NI. The app.R code that launches the ThymoSight app, together with the python notebooks used to create consistent annotation fields, reanalyze and integrate the public datasets with ours have been submitted on GitHub at https://github.com/FredHutch/thymosight. The server hosting the interactive app can be accessed at www.thymosight.org. Environments and packages can be found listed below.<br><br>#3. Other software: GSEA (v4.3.2); Cytoscape (v3.10.0);<br><br>#4. Bulk RNA-seq data were analyzed with FastQC (v0.11.9); Trimmomatic (); STAR aligner (v2.7.0e); featureCounts (v1.6.3)<br><br>#5. All images shown are processed using Imaris 9.7.1 (Bitplane). |

Environment names and requirements for scRNAseq analsysis:

name: Kousa-et-al-2024-NI (main), channels: conda-forge, defaults, dependencies: -alabaster=0.7.12, -anndata=0.8.0, -appnope=0.1.2, -argon2-cffi=21.3.0, -argon2-cffi-bindings=21.2.0, -arpack=3.7.0, -asttokens=2.0.5, -attrs=21.4.0, -babel=2.10.3, -backcall=0.2.0, -beautifulsoup4=4.11.1, -bleach=4.1.0, -brotli=1.0.9, -brotli-bin=1.0.9, -brotlipy=0.7.0, -ca-certificates=2022.6.15.1, -certifi=2022.6.15.1, -cffi=1.15.1, -charset-normalizer=2.1.1, -colorama=0.4.5, -cryptography=37.0.4, -cycler=0.11.0, -debugpy=1.5.1, -decorator=5.1.1, -defusedxml=0.7.1, -docutils=0.19, -entrypoints=0.4, -executing=0.8.3, -fonttools=4.37.1, -freetype=2.10.4, -glpk=4.65, -gmp=6.2.1, -h5py=3.7.0, -hdf5=1.10.6, -icu=58.2, -igraph=0.9.10, -imagesize=1.4.1, -importlib-metadata=4.11.4, -importlib_metadata=4.11.4, -importlib_resources=5.2.0, -ipykernel=6.15.2, -ipython=8.4.0, -ipython_genutils=0.2.0, -jedi=0.18.1, -jinja2=3.0.3, -joblib=1.1.0, -jpeg=9e, -jsonschema=4.4.0, -jupyter_client=7.3.5, -jupyter_core=4.10.0, -jupyterlab_pygments=0.1.2, -kiwisolver=1.4.4, -lcms2=2.12, -leidenalg=0.8.10, -libblas=3.9.0, -libbrotlicommon=1.0.9, -libbrotlidec=1.0.9, -libbrotlienc=1.0.9, -libcblas=3.9.0, -libcxx=14.0.6, -libffi=3.3, -libgfortran=4.0.0, -libgfortran4=7.5.0, -libiconv=1.17, -liblapack=3.9.0, -libllvm11=11.1.0, -libopenblas=0.3.12, -libpng=1.6.37, -libsodium=1.0.18, -libtiff=4.2.0, -libwebp-base=1.2.4, -libxml2=2.9.14, -llvm-openmp=14.0.4, -llvmlite=0.38.0, -lz4-c=1.9.3, -markupsafe=2.1.1, -matplotlib-base=3.5.3, -matplotlib-inline=0.1.6, -metis=5.1.0, -mistune=0.8.4, -mpfr=4.1.0, -munkres=1.1.4, -natsort=8.2.0, -nbclient=0.5.13, -nbconvert=6.4.4, -nbformat=5.3.0, -ncurses=6.3, -nest-asyncio=1.5.5, -networkx=2.8.6, -notebook=6.4.12, -numba=0.55.1, -olefile=0.46, -openssl=1.1.1q, -packaging=21.3, -pandas=1.4.4, -pandocfilters=1.5.0, -parso=0.8.3, -patsy=0.5.2, -pexpect=4.8.0, -pickleshare=0.7.5, -pip=22.1.2, -prometheus_client=0.14.1, -prompt-toolkit=3.0.20, -psutil=5.9.0, -ptyprocess=0.7.0, -pure_eval=0.2.2, -pycparser=2.21, -pygments=2.11.2, -pynndescent=0.5.7, -pyopenssl=22.0.0, -pyparsing=3.0.9, -pyrsistent=0.18.0, -pysocks=1.7.1, -python=3.8.13, -python-dateutil=2.8.2, -python-fastjsonschema=2.16.2, -python-igraph=0.9.11, -python_abi=3.8, -pytz=2022.2.1, -pyzmq=23.2.0, -readline=8.1.2, -scanpy=1.9.1, -scikit-learn=1.1.2, -scipy=1.5.3, -seaborn=0.12.0, -seaborn-base=0.12.0, -send2trash=1.8.0, -session-info=1.0.0, -setuptools=63.4.1, -six=1.16.0, -snowballstemmer=2.2.0, -soupsieve=2.3.1, -sphinx=5.1.1, -sphinxcontrib-applehelp=1.0.2, -sphinxcontrib-devhelp=1.0.2, -sphinxcontrib-htmlhelp=2.0.0, -sphinxcontrib-jsmath=1.0.1, -sphinxcontrib-qthelp=1.0.3, -sphinxcontrib-serializinghtml=1.1.5, -sqlite=3.39.2, -stack_data=0.2.0, -statsmodels=0.13.2, -stdlib-list=0.7.0, -suitesparse=5.10.1, -tbb=2021.5.0, -terminado=0.13.1, -testpath=0.6.0, -texttable=1.6.4, -threadpoolctl=3.1.0, -tk=8.6.12, -tornado=6.2, -tqdm=4.64.1, -traitlets=5.1.1, -typing-extensions=4.3.0, -typing_extensions=4.3.0, -umap-learn=0.5.3, -unicodedata2=14.0.0, -urllib3=1.26.11, -wcwidth=0.2.5, -webencodings=0.5.1, -wheel=0.37.1, -xz=5.2.5, -zeromq=4.3.4, -zipp=3.8.0, -zlib=1.2.12, -zstd=1.4.9, -pip:, -aniso8601==9.0.1, -anndata2ri==1.1, -annoy==1.17.1, -backports-zoneinfo==0.2.1, -biomart==0.9.2, -chardet==4.0.0, -click==8.1.3, -cmake==3.24.1.1, -cython==0.29.32, -dunamai==1.13.0, -et-xmlfile==1.1.0, -fa2==0.3.5, -fbpca==1.0, -fcsparser==0.2.4, -feather-format==0.4.1, -flask==1.1.2, -flask-cors==3.0.10, -flask-restful==0.3.9, -geosketch==1.2, -get-version==3.5.4, -harmonypy==0.0.6, -harmonyts==0.1.4, -idna==2.10, -imageio==2.22.1, -intervaltree==3.1.0, -itsdangerous==2.1.2, -loompy==3.0.7, -mousipy==0.0.5, -nbstripout==0.6.1, -numexpr==2.8.3, -numpy==1.20.2, -numpy-groupies==0.9.19, -openpyxl==3.0.10, -palantir==1.0.1, -phenograph==1.5.7, -pillow==9.2.0, -pyarrow==9.0.0, -pyreadr==0.4.6, -python-slugify==5.0.2, -pytz-deprecation-shim==0.1.0.post0, -pywavelets==1.3.0, -requests==2.25.1, -rpy2==3.5.4, -scanorama==1.7.2, -scikit-image==0.19.3, -scrublet==0.2.3, -scvelo==0.2.4, -sklearn==0.0, -sortedcontainers==2.4.0, -tables==3.7.0, -text-unidecode==1.3, -tifffile==2022.8.12, -tzdata==2022.2, -tzlocal==4.2, -watermark==2.3.1, -werkzeug==2.2.2, -wrapt==1.15.0, -xlrd==2.0.1, -xlsxwriter==3.0.3

name: Kousa-et-al-2024-NI (dynamo), channels: defaults, dependencies: -anyio=4.2.0, -appnope=0.1.2, -argon2-cffi=21.3.0, -argon2-cffi-bindings=21.2.0, -asttokens=2.0.5, -async-lru=2.0.4, -babel=2.11.0, -beautifulsoup4=4.12.2, -bleach=4.1.0, -brotli-python=1.0.9, -bzip2=1.0.8, -ca-certificates=2023.12.12, -certifi=2024.2.2, -cffi=1.16.0, -charset-normalizer=2.0.4, -comm=0.1.2, -debugpy=1.6.7, -decorator=5.1.1, -defusedxml=0.7.1, -executing=0.8.3, -ipykernel=6.28.0, -ipython=8.20.0, -jedi=0.18.1, -json5=0.9.6, -jsonschema=4.19.2, -jsonschema-specifications=2023.7.1, -jupyter-lsp=2.2.0, -jupyter_client=8.6.0, -jupyter_core=5.5.0, -jupyter_events=0.8.0, -jupyter_server=2.10.0, -jupyter_server_terminals=0.4.4, -jupyterlab=4.0.11, -jupyterlab_pygments=0.1.2, -jupyterlab_server=2.25.1, -libcxx=14.0.6, -libffi=3.4.4, -libsodium=1.0.18, -matplotlib-inline=0.1.6, -mistune=2.0.4, -nbclient=0.8.0, -nbconvert=7.10.0, -nbformat=5.9.2, -ncurses=6.4, -nest-asyncio=1.6.0, -notebook=7.0.8, -notebook-shim=0.2.3, -openssl=3.0.13, -overrides=7.4.0, -pandocfilters=1.5.0, -parso=0.8.3, -pexpect=4.8.0, -pip=23.3.1, -prometheus_client=0.14.1, -prompt-toolkit=3.0.43, -prompt_toolkit=3.0.43, -psutil=5.9.0, -ptyprocess=0.7.0, -pure_eval=0.2.2, -pycparser=2.21, -pysocks=1.7.1, -python=3.11.8, -python-fastjsonschema=2.16.2, -python-json-logger=2.0.7, -pyyaml=6.0.1, -pyzmq=25.1.2, -readline=8.2, -referencing=0.30.2, -requests=2.31.0, -rfc3339-validator=0.1.4, -rfc3986-validator=0.1.1, -rpds-py=0.10.6, -send2trash=1.8.2, -setuptools=68.2.2, -six=1.16.0, -sniffio=1.3.0, -soupsieve=2.5, -sqlite=3.41.2, -stack_data=0.2.0, -terminado=0.17.1, -tinycss2=1.2.1, -tk=8.6.12, -tornado=6.3.3, -traitlets=5.7.1, -typing_extensions=4.9.0, -urllib3=2.1.0, -wcwidth=0.2.5, -webencodings=0.5.1, -websocket-client=0.58.0, -wheel=0.41.2, -xz=5.4.6, -yaml=0.2.5, -zeromq=4.3.5, -zlib=1.2.13, -pip:, -absl-py==2.1.0, -aiohttp==3.9.3, -aiosignal==1.3.1, -anndata==0.10.5.post1, -array-api-compat==1.4.1, -attrs==23.2.0, -cellrank==2.0.3.dev5+g721c59f, -cfgv==3.4.0, -chex==0.1.85, -click==8.1.7, -colorcet==3.1.0, -contextlib2==21.6.0, -contourpy==1.2.0, -cycler==0.12.1, -distlib==0.3.8, -docrep==0.3.2, -dunamai==1.19.2, -dynamo-release==1.4.0, -et-xmlfile==1.1.0, -etils==1.7.0, -filelock==3.13.1, -flax==0.8.1, -fonttools==4.49.0, -frozenlist==1.4.1, -fsspec==2024.2.0, -get-version==3.5.5, -h5py==3.10.0, -identify==2.5.35, -idna==3.6, -igraph==0.10.8, -importlib-resources==6.1.2, -jax==0.4.25, -jaxlib==0.4.25, -jinja2==3.0.3, -joblib==1.3.2, -kiwisolver==1.4.5, -lightning==2.1.4, -lightning-utilities==0.10.1, -llvmlite==0.42.0, -loompy==3.0.7, -louvain==0.8.0, -markdown-it-py==3.0.0, -markupsafe==2.1.5, -matplotlib==3.6.3, -mdurl==0.1.2, -ml-collections==0.1.1, -ml-dtypes==0.3.2, -mpmath==1.3.0, -msgpack==1.0.8, -mudata==0.2.3, -multidict==6.0.5, -multipledispatch==1.0.0, -natsort==8.4.0, -networkx==3.2.1, -nodeenv==1.8.0, -numba==0.59.0, -numdifftools==0.9.41, -numpy==1.26.4, -numpy-groupies==0.10.2, -numpyro==0.13.2, -openpyxl==3.1.2, -opt-einsum==3.3.0, -optax==0.1.9, -orbax-checkpoint==0.5.3, -packaging==23.2, -pandas==2.2.1, -patsy==0.5.6, -pillow==10.2.0, -platformdirs==4.2.0, -pre-commit==3.6.2, -progressbar2==4.4.1, -protobuf==4.25.3, -pygam==0.9.1, -pygments==2.17.2, -pygpcca==1.0.4, -pynndescent==0.5.11, -pyparsing==3.1.1, -pyro-api==0.1.2, -pyro-ppl==1.9.0, -python-dateutil==2.9.0, -python-utils==3.8.2, -pytorch-lightning==2.2.0.post0, -pytz==2024.1, -rich==13.7.1, -scanpy==1.9.8, -scikit-learn==1.1.3, -scipy==1.11.4, -scvelo==0.3.1, -scvi-tools==1.1.1, -seaborn==0.13.2, -session-info==1.0.0, -statsmodels==0.14.1, -stdlib-list==0.10.0, -sympy==1.12, -tensorstore==0.1.54, -texttable==1.7.0, -threadpoolctl==3.3.0, -toolz==0.12.1, -torch==2.2.1, -torchmetrics==1.3.1, -tqdm==4.66.2, -typing-extensions==4.10.0, -tzdata==2024.1, -umap-learn==0.5.5, -virtualenv==20.25.1, -wrapt==1.16.0, -yarl==1.9.4, -zipp==3.17.0

name: Kousa-et-al-2024-NI (thymosight), channels: -conda-forge, -defaults, dependencies: -anndata=0.10.5.post1, -anyio=4.2.0, -appnope=0.1.2, -argon2-cffi=21.3.0, -argon2-cffi-bindings=21.2.0, -arpack=3.8.0, -array-api-compat=1.4.1, -asttokens=2.0.5, -async-lru=2.0.4, -attrs=23.1.0, -babel=2.11.0, -beautifulsoup4=4.12.2, -bleach=4.1.0, -brotli=1.1.0, -brotli-bin=1.1.0, -brotli-python=1.0.9, -bzip2=1.0.8, -c-ares=1.27.0, -ca-certificates=2024.2.2, -cached-property=1.5.2, -cached_property=1.5.2, -certifi=2024.2.2, -cffi=1.16.0, -charset-normalizer=2.0.4, -colorama=0.4.6, -contourpy=1.2.0, -cycler=0.12.1, -debugpy=1.6.7, -decorator=5.1.1, -defusedxml=0.7.1, -exceptiongroup=1.2.0, -executing=0.8.3, -fonttools=4.25.0, -freetype=2.12.1, -get-annotations=0.1.2, -glpk=5.0, -gmp=6.3.0, -h5py=3.10.0, -hdf5=1.14.3, -icu=73.2, -idna=3.4, -igraph=0.10.10, -ipykernel=6.28.0, -ipython=8.20.0, -jedi=0.18.1, -jinja2=3.1.3, -joblib=1.3.2, -json5=0.9.6, -jsonschema=4.19.2, -jsonschema-specifications=2023.7.1, -jupyter-lsp=2.2.0, -jupyter_client=8.6.0, -jupyter_core=5.5.0, -jupyter_events=0.8.0, -jupyter_server=2.10.0, -jupyter_server_terminals=0.4.4, -jupyterlab=4.0.11, -jupyterlab_pygments=0.1.2, -jupyterlab_server=2.25.1, -kiwisolver=1.4.5, -krb5=1.21.2, -lcms2=2.16, -leidenalg=0.10.2, -lerc=4.0.0, -libaec=1.1.2, -libblas=3.9.0, -libbrotlicommon=1.1.0, -libbrotlidec=1.1.0, -libbrotlienc=1.1.0, -libcblas=3.9.0, -libcurl=8.5.0, -libcxx=16.0.6, -libdeflate=1.19, -

libedit=3.1.20191231, -libev=4.33, -libexpat=2.5.0, -libffi=3.4.4, -libgfortran=5.0.0, -libgfortran5=13.2.0, -libhwloc=2.9.3, -libiconv=1.17, -libjpeg-turbo=3.0.0, -liblapack=3.9.0, -libleidenalg=0.11.1, -libllvm14=14.0.6, -libnghttp2=1.58.0, -libopenblas=0.3.26, -libpng=1.6.43, -libsodium=1.0.18, -libsqlite=3.45.1, -libssh2=1.11.0, -libtiff=4.6.0, -libwebp-base=1.3.2, -libxcb=1.15, -libxml2=2.12.5, -libzlib=1.2.13, -llvm-openmp=17.0.6, -llvmlite=0.42.0, -markupsafe=2.1.3, -matplotlib-base=3.8.3, -matplotlib-inline=0.1.6, -mistune=2.0.4, -munkres=1.1.4, -natsort=8.4.0, -nbclient=0.8.0, -nbconvert=7.10.0, -nbformat=5.9.2, -ncurses=6.4, -nest-asyncio=1.6.0, -networkx=3.2.1, -notebook=7.0.8, -notebook-shim=0.2.3, -numba=0.59.0, -numpy=1.26.4, -openjpeg=2.5.0, -openssl=3.2.1, -overrides=7.4.0, -packaging=23.1, -pandas=2.2.1, -pandocfilters=1.5.0, -parso=0.8.3, -patsy=0.5.6, -pexpect=4.8.0, -pillow=10.2.0, -pip=23.3.1, -platformdirs=3.10.0, -prometheus_client=0.14.1, -prompt-toolkit=3.0.43, -prompt_toolkit=3.0.43, -psutil=5.9.0, -pthread-stubs=0.4, -ptyprocess=0.7.0, -pure_eval=0.2.2, -pycparser=2.21, -pygments=2.15.1, -pynndescent=0.5.11, -pyparsing=3.1.1, -pysocks=1.7.1, -python=3.11.8, -python-dateutil=2.8.2, -python-fastjsonschema=2.16.2, -python-igraph=0.11.4, -python-json-logger=2.0.7, -python-tzdata=2024.1, -python_abi=3.11, -pytz=2023.3.post1, -pyyaml=6.0.1, -pyzmq=25.1.2, -readline=8.2, -referencing=0.30.2, -requests=2.31.0, -rfc3339-validator=0.1.4, -rfc3986-validator=0.1.1, -rpds-py=0.10.6, -scanpy=1.9.8, -scikit-learn=1.4.1.post1, -scipy=1.12.0, -seaborn=0.13.2, -seaborn-base=0.13.2, -send2trash=1.8.2, -session-info=1.0.0, -setuptools=68.2.2, -six=1.16.0, -sniffio=1.3.0, -soupsieve=2.5, -sqlite=3.41.2, -stack_data=0.2.0, -statsmodels=0.14.1, -stdlib-list=0.10.0, -tbb=2021.11.0, -terminado=0.17.1, -texttable=1.7.0, -threadpoolctl=3.3.0, -tinycss2=1.2.1, -tk=8.6.13, -tornado=6.3.3, -tqdm=4.66.2, -traitlets=5.7.1, -typing-extensions=4.9.0, -typing_extensions=4.9.0, -tzdata=2024a, -umap-learn=0.5.5, -urllib3=2.1.0, -wcwidth=0.2.5, -webencodings=0.5.1, -websocket-client=0.58.0, -wheel=0.41.2, -xorg-libxau=1.0.11, -xorg-libxdmcp=1.1.3, -xz=5.4.5, -yaml=0.2.5, -zeromq=4.3.5, -zlib=1.2.13, -zstd=1.5.5, -pip:, -annoy==1.17.3, -biomart==0.9.2, -comm==0.2.1, -cython==3.0.8, -et-xmlfile==1.1.0, -harmonypy==0.0.6, -imageio==2.34.0, -ipywidgets==8.1.2, -jupyterlab-widgets==3.0.10, -lazy-loader==0.3, -openpyxl==3.1.2, -scikit-image==0.22.0, -scrublet==0.2.3, -tifffile==2024.2.12, -widgetsnbextension==4.0.10

name: Kousa-et-al-NI (cellchat), -R=4.2.2, -Bioconductor=3.15, Python: 3.8.2 (virtualenv), packages: -BBmisc=1.13, -BH=1.78.0-0, -Biobase=2.56.0, -BiocGenerics=0.42.0, -BiocManager=1.30.19, -BiocParallel=1.30.4, -BiocVersion=3.15.2, -CellChat=1.4.0, -ComplexHeatmap=2.12.1, -DBI=1.1.3, -DelayedArray=0.22.0, -DelayedMatrixStats=1.18.2, -DropletUtils=1.16.0, -EnhancedVolcano=1.14.0, -FNN=1.1.3.1, -Formula=1.2-4, -GenomeInfoDb=1.32.4, -GenomeInfoDbData=1.2.8, -GenomicRanges=1.48.0, -GetoptLong=1.0.5, -GlobalOptions=0.1.2, -HDF5Array=1.24.2, -Hmisc=4.7-1, -IRanges=2.30.1, -KernSmooth=2.23-20, -MASS=7.3-58.1, -MAST=1.22.0, -Matrix=1.5-3, -MatrixGenerics=1.8.1, -MatrixModels=0.5-1, -NMF=0.24.0, -R.methodsS3=1.8.2, -R.oo=1.25.0, -R.utils=2.12.2, -R6=2.5.1, -RANN=2.6.1, -RColorBrewer=1.1-3, -RCurl=1.98-1.9, -ROCR=1.0-11, -RSpectra=0.16-1, -Rcpp=1.0.9, -RcppAnnoy=0.0.20, -RcppArmadillo=0.11.4.2.1, -RcppEigen=0.3.3.9.3, -RcppProgress=0.4.2, -RcppTOML=0.1.7, -Rhdf5lib=1.18.2, -Rtsne=0.16, -S4Vectors=0.34.0, -Seurat=4.3.0, -SeuratObject=4.1.3, -SingleCellExperiment=1.18.1, -SparseM=1.81, -SummarizedExperiment=1.26.1, -TSP=1.2-1, -UpSetR=1.4.0, -XVector=0.36.0, -abind=1.4-5, -anndata=0.7.5.3, -arm=1.13-1, -arrow=10.0.0, -askpass=1.1, -assertthat=0.2.1, -backports=1.4.1, -base64enc=0.1-3, -beachmat=2.12.0, -bit=4.0.5, -bit64=4.0.5, -bitops=1.0-7, -blob=1.2.3, -boot=1.3-28.1, -brew=1.0-7, -brio=1.1.3, -broom=1.0.1, -bslib=0.4.1, -ca=0.71.1, -caTools=1.18.2, -cachem=1.0.6, -callr=3.7.3, -car=3.1-1, -carData=3.0-5, -cellranger=1.1.0, -checkmate=2.1.0, -circlize=0.4.15, -cli=3.4.1, -clipr=0.8.0, -clue=0.3-63, -cluster=2.1.4, -coda=0.19-4, -codetools=0.2-18, -colorspace=2.0-3, -commonmark=1.8.1, -corrplot=0.92, -cowplot=1.1.1, -cpp11=0.4.3, -crayon=1.5.2, -credentials=1.3.2, -crosstalk=1.2.0, -curl=4.3.3, -data.table=1.14.6, -dbplyr=2.1.1, -deldir=1.0-6, -dendextend=1.16.0, -desc=1.4.2, -devtools=2.4.3, -dichromat=2.0-0.1, -diffobj=0.3.5, -digest=0.6.31, -doParallel=1.0.17, -dplyr=1.0.10, -dqrng=0.3.0, -dtplyr=1.2.2, -edgeR=3.38.4, -egg=0.4.5, -ellipsis=0.3.2, -emmeans=1.8.2, -estimability=1.4.1, -evaluate=0.19, -expm=0.999-6, -fansi=1.0.3, -farver=2.1.1, -fastmap=1.1.0, -feather=0.3.5, -figpatch=0.2, -fitdistrplus=1.1-8, -fontawesome=0.4.0, -forcats=0.5.2, -foreach=1.5.2, -foreign=0.8-83, -formatR=1.12, -formattable=0.2.1, -fs=1.5.2, -futile.logger=1.4.3, -futile.options=1.0.1, -future=1.29.0, -future.apply=1.10.0, -gargle=1.2.1, -gclus=1.3.2, -generics=0.1.3, -gert=1.6.0, -ggalluvial=0.12.3, -ggcorrplot=0.1.4, -ggplot2=3.4.0, -ggplotify=0.1.0, -ggpubr=0.5.0, -ggrepel=0.9.2, -ggridges=0.5.4, -ggsci=2.9, -ggsignif=0.6.4, -gh=1.3.0=0.1.1, -globals=0.16.2, -glue=1.6.2, -goftest=1.2-3, -googledrive=2.0.0, -googlesheets4=1.0.1, -gplots=3.1.3, -gprofiler2=0.2.1, -gridBase=0.4-7, -gridExtra=2.3, -gridGraphics=0.5-1, -gtable=0.3.1, -gtools=3.9.4, -haven=2.5.1, -heatmaply=1.4.0, -here=1.0.1, -highr=0.9, -hms=1.1.2, -htmlTable=2.4.1, -htmltools=0.5.4, -htmlwidgets=1.5.4, -httpuv=1.6.6, -httr=1.4.4, -ica=1.0-3, -ids=1.0.1, -igraph=1.3.5, -ini=0.3.1, -interp=1.1-3, -irlba=2.3.5.1, -isoband=0.2.6, -iterators=1.0.14, -jpeg=0.1-9, -jquerylib=0.1.4, -jsonlite=1.8.4, -knitr=1.41, -labeling=0.4.2, -lambda.r=1.2.4, -later=1.3.0, -lattice=0.20-45, -latticeExtra=0.6-30, -lazyeval=0.2.2, -leiden=0.4.3, -lifecycle=1.0.3, -limma=3.52.4, -listenv=0.8.0, -lme4=1.1-31, -lmtest=0.9-40, -locfit=1.5-9.6, -lubridate=1.8.0, -magick=2.7.3, -magrittr=2.0.3, -mapproj=1.2.9, -maps=3.4.1, -matrixStats=0.63.0, -memoise=2.0.1, -mgcv=1.8-41, -mime=0.12, -miniUI=0.1.1.1, -minqa=1.2.5, -modelr=0.1.9, -munsell=0.5.0, -mvtnorm=1.1-3, -naniar=0.6.1, -network=1.17.2, -nlme=3.1-160, -nloptr=2.0.3, -nnet=7.3-18, -norm=1.0-10.0, -numDeriv=2016.8-1.1, -openssl=2.0.5, -openxlsx=4.2.5.1, -pacman=0.5.1, -pals=1.7, -parallelly=1.32.1, -patchwork=1.1.2, -pbapply=1.6-0, -pbkrtest=0.5.1, -pheatmap=1.0.12, -pillar=1.8.1, -pkgbuild=1.3.1, -pkgconfig=2.0.3, -pkgload=1.3.2, -pkgmaker=0.32.2, -plotly=4.10.1, -plyr=1.8.8, -png=0.1-8, -polyclip=1.10-4, -polynom=1.4-1, -praise=1.0.0, -prettyunits=1.1.1, -processx=3.8.0, -progress=1.2.2, -progressr=0.12.0, -promises=1.2.0.1, -ps=1.7.2, -purrr=0.3.5, -qap=0.1-2, -quantreg=5.94, -rappdirs=0.3.3, -rcmdcheck=1.4.0, -readr=2.1.3, -readxl=1.4.1, -registry=0.5-1, -rematch=1.0.1, -rematch2=2.1.2, -remotes=2.4.2, -renv=0.15.5, -reprex=2.0.2, -reshape2=1.4.4, -reticulate=1.26, -rhdf5=2.40.0, -rhdf5filters=1.8.0, -rjson=0.2.21, -rlang=1.0.6, -rmarkdown=2.17, -rngtools=1.5.2, -roxygen2=7.1.2, -rpart=4.1.19, -rprojroot=2.0.3, -rstatix=0.7.1, -rstudioapi=0.13, -rversions=2.1.1, -rvest=1.0.3, -sass=0.4.4, -scCB2=1.6.0, -scales=1.2.1, -scattermore=0.8, -sctransform=0.3.5, -scuttle=1.6.3, -selectr=0.4-2, -seriation=1.4.0, -sessioninfo=1.2.2, -shape=1.4.6, -shiny=1.7.3, -sitmo=2.0.2, -sna=2.7, -snow=0.4-4, -sourcetools=0.1.7, -sp=1.5-1, -sparseMatrixStats=1.8.0, -spatstat.data=3.0-0, -spatstat.explore=3.0-5, -spatstat.geom=3.0-3, -spatstat.random=3.0-1, -spatstat.sparse=3.0-0, -spatstat.utils=3.0-1, -statnet.common=4.6.0, -stringi=1.7.8, -stringr=1.5.0, -survival=3.4-0, -svglite=2.1.0, -sys=3.4.1, -systemfonts=1.0.4, -tensor=1.5, -testthat=3.1.6, -tibble=3.1.8, -tidyr=1.2.1, -tidyselect=1.2.0, -tidyverse=1.3.2, -tinytex=0.42, -tzdb=0.3.0, -usethis=2.1.5, -utf8=1.2.2, -uuid=1.1-0, -uwot=0.1.14, -vctrs=0.5.1, -viridis=0.6.2, -viridisLite=0.4.1, -visdat=0.5.3, -vroom=1.6.0, -waldo=0.4.0, -webshot=0.5.4, -whisker=0.4, -withr=2.5.0, -xfun=0.35, -xml2=1.3.3, -xopen=1.0.0, -xtable=1.8-4, -yaml=2.3.6, -yulab.utils=0.0.5, -zip=2.2.0, -zlibbioc=1.42.0, -zoo=1.8-11

For manuscripts utilizing custom algorithms or software that are central to the research but not yet described in published literature, software must be made available to editors and reviewers. We strongly encourage code deposition in a community repository (e.g. GitHub). See the Nature Portfolio guidelines for submitting code & software for further information.

# Data

All manuscripts must include a data availability statement. This statement should provide the following information, where applicable:
- Accession codes, unique identifiers, or web links for publicly available datasets
- A description of any restrictions on data availability
- For clinical datasets or third party data, please ensure that the statement adheres to our policy

Sequencing data generated in this study have been deposited in NCBI's Gene Expression Omnibus (GEO) and can be accessed through the GEO SuperSeries accession number GSE240020. Accession numbers for publicly available raw count data that have been re-analyzed for this study are listed here: (i) [Mouse] > Kernfeld et al. (2018) [GSE107910]; Bornstein et al. (2018) [GSE103967]; Dhalla et al. (2019) [https://www.ebi.ac.uk/biostudies/arrayexpress/studies/E-MTAB-8105#]; Baran-Gale et al. (2020) [https://bioconductor.org/packages/release/data/experiment/html/MouseThymusAgeing.html]; Wells et al. (2020) [GSE137699]; Rota et al. (2021) [GSE162668]; Nusser et al. (2022) [GSE106856]; Michelson et al. (2022) [GSE194253]; Klein et al. (2023) [GSE215418]; Farley et al. (2023) [GSE232765]; Givony et al. (2023) [GSE236075]; Michelson et al. (2023) [GSE225661]; Horie et al. (2023) [GSE228198] and (ii) [Human] > Park et al. (2020) [https://zenodo.org/records/3711134]; Bautista et al. (2021) [GSE147520]; Ragazzini et al. (2023) [GSE220830, GSE220206, GSE220829]. The re-analyzed public datasets with added metadata can be accessed at 10.5281/zenodo.12516405.

# Research involving human participants, their data, or biological material

| | |
|---|---|
| Reporting on sex and gender | Thymus tissues were obtained from the archives of the Duke University Department of Pathology as FFPE sections. All tissues were used anonymously, with recording of only patient age, gender, and surgical diagnosis. We show imagies from one 50 year old female patient. |
| Reporting on race, ethnicity, or other socially relevant groupings | N/A |
| Population characteristics | Histology analysis of thymus derived from the Duke University human thymus tissue bank of a de-identified 50 year old female was used in the study |
| Recruitment | N/A |
| Ethics oversight | All human tissues were collected according to a protocol approved by the Duke University Institutional Review Board. |

Note that full information on the approval of the study protocol must also be provided in the manuscript.

# Field-specific reporting

Please select the one below that is the best fit for your research. If you are not sure, read the appropriate sections before making your selection.

☒ Life sciences ☐ Behavioural & social sciences ☐ Ecological, evolutionary & environmental sciences

For a reference copy of the document with all sections, see nature.com/documents/nr-reporting-summary-flat.pdf

# Life sciences study design

All studies must disclose on these points even when the disclosure is negative.

| | |
|---|---|
| Sample size | Sample sizes were based on previously published work and preliminary studies, the effect size (difference between groups/standard deviation) for each study was estimated, and sample sizes were calculated using G*Power (v3.1.9.2) (Faul et al. 2009 Behavior Res Meth. 41:1149-1160) for an alpha=0.05 and beta=0.8. |
| Data exclusions | No samples were excluded from scRNA-seq, visium or flow cytometry datasets.<br><br>scRNA-seq cell exclusions: Quality of the single cells was computationally assessed based on total counts, number of genes, mitochondrial and ribosomal fraction per cell, with low total counts, low number of genes (≤1000) and high mitochondrial content (≥0.2) as negative indicators of cell quality. Cells characterized by more than one negative indicator were considered as "bad" quality cells. Although cells were negatively sorted prior to sequencing for the CD45 marker, a small amount of CD45+ cells (expressing Ptprc), and also a few parathyroid cells (expressing Gcm2), were detected within our dataset. To remove bad quality cells and contaminants in an unbiased way, we assessed them in a cluster basis rather than individually. Leiden clusters with a "bad" quality profile and/or a high number of contaminating cells were removed. Finally, cells marked as doublets by scrublet were also filtered out. |
| Replication | In general variation within groups and between experiments was low, however to take into account inter-experimental variation all experiments were performed at least twice and in no instance were experiments discarded due to conflicting findings. To account for intra- |

experimental variation, particularly for in vitro studies, several wells per conditions were assessed with primary sample material coming from at least two different mice. In vitro experiments were performed independently at least three times.

| Randomization | No specific method of randomization was used. However, for all experiments, mouse cages were randomly allocated to each group after balancing of age and sex. |
|---|---|
| Blinding | Due to practical considerations in experimental design and the fact that aged mice are typically easy to distinguish from young mice, the performing investigator was not typically blinded to the group allocation during the experiments. |

# Reporting for specific materials, systems and methods

We require information from authors about some types of materials, experimental systems and methods used in many studies. Here, indicate whether each material, system or method listed is relevant to your study. If you are not sure if a list item applies to your research, read the appropriate section before selecting a response.

## Materials & experimental systems

| n/a | Involved in the study |
|---|---|
| ☐ | ☒ Antibodies |
| ☒ | ☐ Eukaryotic cell lines |
| ☒ | ☐ Palaeontology and archaeology |
| ☐ | ☒ Animals and other organisms |
| ☒ | ☐ Clinical data |
| ☒ | ☐ Dual use research of concern |
| ☒ | ☐ Plants |

## Methods

| n/a | Involved in the study |
|---|---|
| ☒ | ☐ ChIP-seq |
| ☐ | ☒ Flow cytometry |
| ☒ | ☐ MRI-based neuroimaging |

## Antibodies

| Antibodies used | The antibodies used were rabbit anti-pan-Cytokeratin (Dako, Cat# Z0622), anti-K5 (BioLegend, Cat# poly19055), rat anti-mouse K8/18 (Troma-1; Developmental Studies Hybridoma Bank, Iowa City), rabbit anti-K14 (AbCam, Cat# EPR17350), rat anti-mouse AIRE (WEHI, Clone# 5H12), rabbit anti-human/mouse DCLK1 (LSBio, Cat# LS-C100746), biotinylated UEA-1 lectin (Vector labs, USA, Cat# B-1065). The secondary antibodies used were Alexa Fluor® 647 Donkey anti-rabbit IgG (H+L) (Invitrogen, Cat# A31573), Alexa Fluor® 647 Goat anti-rat IgG (H+L) (Invitrogen, Cat# A-21247), Alexa Fluor™ 647 Streptavidin conjugate (Invitrogen, Cat# S21374).<br><br>For flow cytometry and cell sorting, surface antibodies against CD45 (30-F11), CD31 (390 or MEC13.3), TER-119 (TER-119), MHC-II IA/IE (M5/114.15.2), EpCAM (G8.8), Ly51 (6C3), PDGFRα (APA5), CD104 (346-11A), L1CAM (555), Ly6D (49-H4), Gp38 (8.1.1), CD26 (H194-112), CD62P (RB40.34), podoplanin (8.1.1), CD62P (RB40.34), CD9 (KMC8), and CD309 (Avas12a) were purchased from BD Biosciences (Franklin Lakes, NJ), BioLegend (San Diego, CA) or eBioscience (San Diego, CA). Ulex europaeus agglutinin 1 (UEA1), was purchased from Vector Laboratories (Burlingame, CA). Antibodies against phosphoAKT was purchased from Cell Signaling Technologies (Danvers, MA); Claudin-3 and anti-rabbit secondary were purchased from Invitrogen (Thermo-Fisher, Waltham, MA); DCLK1 (aa690-720) was purchased from LSBio (Seattle, WA); GP2 (2F11-C3) was purchased from MBL Life Science; and anti-GFP (Aves GFP-1020) was purchased from AvesLabs (Davis, CA). Anti-FOXN1 antibody was a gift from Hans Reimer-Rodewald. A detailed table outlining specific vendors, fluorochromes, catalog numbers, lot numbers and dilutions has been included in a separate data table. |
|---|---|
| Validation | Isotype and fluorescent-minus one controls were used to set gates. |

## Animals and other research organisms

Policy information about studies involving animals; ARRIVE guidelines recommended for reporting animal research, and Sex and Gender in Research

| Laboratory animals | Inbred male and female C57BL/6J mice were obtained from Jackson Laboratories or through the National Institute of Aging mouse colony. Foxn1tdTomato mice were generated by crossing Foxn1-cre (Jax 018448) with B6.Cg-Gt(ROSA)26Sortm14(CAG-tdTomato)Hze/J mice (Jax 007914). Foxn1nTnG mice were generated by crossing Foxn1-cre (Jax 018448) with ROSAnT-nG mice (Jax 023537). Foxn1z/z mice were generated as previously described. As an aging study, we used mice across the lifespan at 2 months-old, 6mo, 9mo, 12mo, and 18-24mo. Animal rooms were kept in a constant temperature of 72 degrees (+/- 3 degrees) Fahrenheit with humidity set between 30%-70% as recommended in the "Guide for the Care and Use of Lab Animals". Light cycles were set for 12 hours on and 12 hours off. |
|---|---|
| Wild animals | No wild animals were used in this study. |
| Reporting on sex | All mouse ages and sexes are reported in figure legends |
| Field-collected samples | No field collected samples were used in this study. |
| Ethics oversight | All studies were performed under approved Institutional Animal Care and Use Committee (or equivalent) protocols at Memorial |

| Ethics oversight | Sloan Kettering Cancer Center, Fred Hutchinson Cancer Center, University of Georgia, or Walter and Eliza Hall Institute. |

Note that full information on the approval of the study protocol must also be provided in the manuscript.

# Flow Cytometry

## Plots

Confirm that:

☒ The axis labels state the marker and fluorochrome used (e.g. CD4-FITC).

☒ The axis scales are clearly visible. Include numbers along axes only for bottom left plot of group (a 'group' is an analysis of identical markers).

☒ All plots are contour plots with outliers or pseudocolor plots.

☒ A numerical value for number of cells or percentage (with statistics) is provided.

## Methodology

| Sample preparation | Thymus was enzymatically digested following and adapted protocol. Briefly, thymi were mechanically dissociated into 1-2 mm pieces. Tissue pieces were incubated with a digestion buffer (either, RPMI with 10% FCS, 62.5 um/mL liberase TM, 0.4 mg/ml DNase I; or RPMI with 25mM HEPES, 20µg/µL DNAse1, and 1mg/mL Collagenase/Dispase). Between incubation steps, supernatant containing dissociated cells were transferred to tubes equipped with 100 um filter. Cells were pelleted by centrifugation at 400g for 5 min. All steps were performed at 4ºC unless indicated. For sequencing experiments, cell pellets were incubated with anti-mouse CD45 microbeads and CD45+ cells were depleted from cell suspension using magnetic-associated cell sorting (MACS) on LS columns according to manufacturer's protocol. Following red blood cell lysis using ACK buffer, the CD45-depleted cell fraction was incubated with an antibody cocktail for 15 min at 4 C and cells of interest were purified by fluorescent-associated cell sorting (FACS) on a BD Biosciences Aria II using a 100 ⊠m nozzle. Cells were sorted into tubes containing RPMI supplemented with 2% BSA. FACS-purified cells were spun down at 400g for 5 min and resuspended in PBS supplemented with 0.04 % BSA for generation of single-cell suspensions. |
| Instrument | Cells sorting was performed on a BD Biosciences Aria II or S6 using a 100 um nozzle. Cells were analysed on a ???? |
| Software | Flow cytometry was analyzed using FACSDiva (BD Biosciences) or FlowJo (Treestar Software). |
| Cell population abundance | For sort experiments, cell purity was typically >99% as checked using a post-sort purity check by flow cytometry. |
| Gating strategy | All gating have been outlined in the manuscript. |

☒ Tick this box to confirm that a figure exemplifying the gating strategy is provided in the Supplementary Information.

