## [Peer review file · Nature Immunology]

Age-related epithelial defects limit thymic function and regeneration

Corresponding Author: Dr Jarrod Dudakov

Version 0:

Decision Letter:

28th Jun 2023

Dear Dr Dudakov,

Thank you for providing your response to reviewers comments for your article, "Age-related epithelial defects limit thymic function and regeneration". We would be very interested in considering a revised version that addresses reviewers serious concerns as you outlined.

We hope you will find the referees' comments useful as you decide how to proceed. If you wish to submit a substantially revised manuscript, please bear in mind that we will be reluctant to approach the referees again in the absence of major revisions.

If you choose to revise your manuscript taking into account all reviewer and editor comments, please highlight all changes in the manuscript text file [OPTIONAL: in Microsoft Word format].

* If you have not done so already please begin to revise your manuscript so that it conforms to our Article format instructions at <http://www.nature.com/ni/authors/index.html>. Refer also to any guidelines provided in this letter.

The Reporting Summary can be found here:

When submitting the revised version of your manuscript, please pay close attention to our

<https://www.nature.com/nature-portfolio/editorial-policies/image-integrity>>Digital Image Integrity Guidelines. and to the following points below:

Link Redacted

If you wish to submit a suitably revised manuscript we would hope to receive it within 6 months. If you cannot send it within this time, please let us know. We will be happy to consider your revision so long as nothing similar has been accepted for publication at Nature Immunology or published elsewhere.

Nature Immunology is committed to improving transparency in authorship. As part of our efforts in this direction, we are now requesting that all authors identified as 'corresponding author' on published papers create and link their Open Researcher and Contributor Identifier (ORCID) with their account on the Manuscript Tracking System (MTS), prior to acceptance. ORCID helps the scientific community achieve unambiguous attribution of all scholarly contributions. You can create and link your ORCID from the home page of the MTS by clicking on 'Modify my Springer Nature account'. For more information please visit please visit www.springernature.com/orcid.

Thank you for the opportunity to review your work.

Sincerely,

Stephanie Houston
Editor
Nature Immunology

Reviewers' Comments:

Reviewer #1:

Remarks to the Author:

In this manuscript, Anastasia et al. examined age-related changes in thymic stromal cells using scRNA-seq, spatial transcriptome, lineage-tracing and light-sheet imaging. They found that two atypical thymic epithelial cells (TECs) emerged in aged mice. These age-associated TECs (aaTECs) form highly dense clusters in the peri-medullary region that were devoid of thymocytes. Based on the GSEA analysis, the authors concluded that aaTECs exhibit features of senescence and EMT, and thus form non-functional microenvironment in the involuted thymus. Following sublethal irradiation, the aaTECs expanded substantially, which may perturb normal TEC differentiation in the early stages of regeneration.

The data obtained from the present study are considered to be of high value as a resource. In addition, identification of aaTECs, previously undescribed thymic stromal cells, is novel.

However, this study has several weak points. (1) The mechanism of age-related TEC development is unclear. (2) The role of aaTEC in the involuted thymus is unclear. (3) There is little data on age-related and damage-induced changes in cTECs. To address issues related to (1) and (2), the authors need to identify transcriptional factor(s) as well as signaling pathway that regulate development of aaTECs. Then, the authors should establish mice lacking aaTECs and examine the significance of aaTECs. It is, however, understandable that it will take several years to address these issues. Thus, at least the authors should add more detailed data on aaTECs as described below and clearly refer to the limitation of this study in the "Discussion".

In general, cTECs are essential for the development and maintenance of DP cells, the most abundant thymocyte and a prominent indicator of thymic regeneration. Thus, cTEC may play a key role in thymic regeneration after insult. Despite this, the authors only show limited data on cTECs throughout the manuscript. So, the authors should provide the following data on cTEC and describe the association of cTEC with thymic regeneration.

Major concerns

1. The authors should identify the candidates of key factors such as lineage-defining transcription factors for aaTEC1 and aaTEC2.
2. The authors should provide the possible signaling pathway that drives aaTEC differentiation using transcriptomic data. Are the TNFR superfamily receptors and NFκB signaling still involved in the differentiation of aaTECs?
3. The authors showed that aaTECs can be defined as Ly51- UEA- double negative TEC (Fig.1 i), but it is not enough. The authors should identify molecular markers that are specifically expressed in aaTECs but not in cTEC and mTEC, and confirm this by flow cytometric analysis. This experiment is necessary to enable other researchers to detect aaTECs. If possible, it would be better to perform immunohistochemical staining to confirm the marker is specifically expressed in aaTECs.
4. The authors claim that aaTEC form non-functional microenvironment. However, I am not convinced that peri-medullary region formed by aaTECs is non-functional since the authors do not perform analysis using mice lacking aaTECs. The

authors should hence tone down this statement and improve this subject.

5. Do aaTECs express tissue restricted antigens? Are there possibilities that aaTECs function as source of self-antigens?
6. The authors clearly showed that aaTECs are originated from conventional TECs using Foxn1 nTnG mice. However, it remains unclear which TEC subset(s) give rise to aaTECs. Could you predict progenitor cells of aaTECs using scRNA-seq data?
7. The authors should show the age-related and damage-induced changes in cTECs. How does cTEC signature genes change with age or damage induction (e.g. Foxn1, Psmb11, Dll4, Ly51, CD205, IL-7, CCL25, Prss16, etc.)? If there is a change, could it explain the reason why thymic regeneration is delayed in old mice?
8. The authors have previously reported that BMP4 produced by endothelial cells is critical for thymic regeneration. Which EC subset produce BMP4? Does BMP4 expression increase after irradiation?
9. The Discussion is very tightly written. I feel that the authors should provide discussion and perspective related to not only aaTEC but also fibroblast, EC, and cTEC.
10. The authors should describe the limitation of this study in the Discussion.

Minor concerns

11. Typos : Lines 212, 298, 404, 405.

Reviewer #2:

Remarks to the Author:

The authors sought to identify changes in the murine thymic stromal compartment associated with age and injury. They used single-cell RNA sequencing to identify two populations that accumulate in involuted thymic tissue and used immunostaining to demonstrate the presence of areas dense in TECs but lacking thymocytes in older thymus. They also present bioinformatic data suggesting that age-associated TECs perturbs normal thymic function by acting as sinks for TEC growth factors. The work is interesting and potentially relevant to human thymic involution since TECs lacking characteristics of cTECs and mTECs have been identified in adult thymic tissue. However, the authors need to address a number of concerns to strengthen their conclusions.

Major concerns:

Although the presence of aaTECs is demonstrated by transcriptomics, the lack of markers to clearly identify them by flow cytometry or histology is a major caveat. This would be particularly important when quantifying changes in subsets of cells using flow cytometry. It is indeed not clear which subsets are included when gating on UEA-1 negative cells. If such populations include other cells such as mimetic cells, the data might represent shifts in these other populations instead of aaTECs.

The idea that aaTECs are end-stage populations and not precursors blocked in their differentiation capacity would be greatly strengthened by lineage tracing experiments.

The hypothesis that 'emergence of aaTEC perturbs the network of growth factors supporting stromal cell function and thymocyte differentiation' is based on bioinformatics data only and should be confirmed using biological experiments.

Minor concerns:

Fig. 1H – what is non-mTEC? DN-TEC+cTECs? Please label clearly. What is the rationale for using CD104? Which subset of cells is it marking?

Fig. 1i – What is the gating strategy? Do the quantification of DN-TECs include tuft cells and other mimetic cells?

Fig. 2a - What is the scale representing?

Fig. 2b - What is the gating strategy? Do the quantification of DN-TECs include tuft cells and other mimetic cells?

The sentence '...clear lineage trajectory stemming from the mTEC_{prol} and mTEC₁ populations and continuing into the more differentiated mTEC lineages (mTEC₂ and mTEC₃) in young mice' is not accurate since mTEC_{prol} give rise to mTEC₁ or mTEC₂/mTEC₃, not mTEC₁ continuing to differentiate into mTEC₂/mTEC₃.

Fig. 2c - why not include some of the mimetic cells (goblet, M cells) in the RNA velocity?

Fig. 3B – how is the medulla marked?

Fig. 3D – Krt5, Krt8, and Krt14 are not very specific to any TEC subsets and are definitely not specific to aaTECs. Based on the transcriptome data, was there a cytokeratin that is more specific to aaTECs? Co-staining of GFP and some of these cytokeratins with UEA-1 would help compare the data with the flow cytometry results.

Fig. 3E – please include scale for expression levels.

Figure 4 - Where are aaTEC2 located relative to HD-TEC regions? Co-labeling of GFP and vimentin (or other marker of EMT) would be very informative.

Fig. 5B – the way the graph is scaled is a bit misleading. The x axis should be scaled to better reflect time points and it might be better to use a broken axis instead of log scale for the y axis.

Fig. 5D – What is the gating strategy for each population? Same as Fig. 1H? An example of how the recovery index is calculated would help understand the data for this panel.

Fig. 5E – legend says 12mo while figure is labelled as 18mo. Which one is it? How are aaTECs marked?

Fig. 5F – What is the age of tissue and what time point after TBI? How are the cortex and medulla marked? And how are the subsets of cells identified?

The sentence 'RNA velocity analysis implied that, in 2mo mice, the reemergence of differentiated mTEC populations stemmed largely from the mTEC1 and mTECprol populations (Fig. 5k).' is misleading since it seems that differentiated mTECs (assuming we are talking about mTEC2/mTEC3) are coming from mTECprol, not mTEC1. The statement about mTECprol potentially giving rise to aaTECs is not supported by the data.

Fig. 6G – What does mTEC diff mean?

How do aaTECs relate to previously described TEC subsets such as intertypical TECs, which have been shown to be progenitor-like and also increase with age (Baran-Gale et al. eLife 2020)?

How do aaTECs compare to the immature TECs populations described by Bautista et al. (Nature Communications 2021)? For example, do aaTECs express some of the markers enriched in the immature population that accumulates in older human tissues?

What is the evidence that aaTECs start emerging at 6mo? Throughout the manuscript, these subsets are identified by transcriptomics from 2mo and 18mo mice while the 6mo data is from flow cytometry and might include other subsets (see comment above).

Why use only females for the analysis of aged mice? Are aaTECs also found in males?

Author Rebuttal letter:

Reviewer #1

In this manuscript, Anastasia et al. examined age-related changes in thymic stromal cells using scRNA-seq, spatial transcriptome, lineage-tracing and light-sheet imaging. They found that two atypical thymic epithelial cells (TECs) emerged in aged mice. These age-associated TECs (aaTECs) form highly dense clusters in the peri-medullary region that were devoid of thymocytes. Based on the GSEA analysis, the authors concluded that aaTECs exhibit features of senescence and EMT, and thus form non-functional microenvironment in the involuted thymus. Following sublethal irradiation, the aaTECs expanded substantially, which may perturb normal TEC differentiation in the early stages of regeneration.

The data obtained from the present study are considered to be of high value as a resource. In addition, identification of aaTECs, previously undescribed thymic stromal cells, is novel. However, this study has several weak points. (1) The mechanism of age-related TEC development is unclear. (2) The role of aaTEC in the involuted thymus is unclear. (3) There is little data on age-related and damage-induced changes in cTECs.

To address issues related to (1) and (2), the authors need to identify transcriptional factor(s) as well as signaling pathway that regulate development of aaTECs. Then, the authors should establish mice lacking aaTECs and examine the significance of aaTECs. It is, however, understandable that it will take several years to address these issues. Thus, at least the authors should add more detailed data on aaTECs as described below and clearly refer to the limitation of this study in the "Discussion".

In general, cTECs are essential for the development and maintenance of DP cells, the most abundant thymocyte and a prominent indicator of thymic regeneration. Thus, cTEC may play

a key role in thymic regeneration after insult. Despite this, the authors only show limited data on cTECs throughout the manuscript. So, the authors should provide the following data on cTEC and describe the association of cTEC with thymic regeneration.

We appreciate the broad enthusiasm for our study and have attempted to address reviewer concerns below. In addition, although there are many changes within the manuscript itself, we have tried to outline changes in blue.

Major concerns

1. The authors should identify the candidates of key factors such as lineage-defining transcription factors for aaTEC1 and aaTEC2.

2. The authors should provide the possible signaling pathway that drives aaTEC differentiation using transcriptomic data. Are the TNFR superfamily receptors and NFkB signaling still involved in the differentiation of aaTECs?

Points 1 and 2 address a similar concern with respect to the drivers of aaTEC emergence so we have addressed them together here. TNFR and NFkB signaling (represented by *Tnfrsf11a*, as well as *Ikbkb*, *Chuk*, and *Aire* as *Nfkb* targets) do not seem to be transcriptionally activated in aaTECs (revised Fig. 4f), therefore, we suggest that these signaling pathways are not necessary for aaTEC maintenance; however, we cannot exclude a potential role for them in aaTEC differentiation. In terms of other drivers of aaTEC differentiation, we now add substantive new data identifying that loss of *FOXN1* is a key contributor to aaTEC emergence. Neither aaTEC subset had detectable *Foxn1* transcripts (Fig. 7b-c) or *FOXN1* protein expression (Fig. 7a). In fact, we found an inverse correlation between *Foxn1* transcription and differentiation potential toward aaTEC derived from RNA velocity analysis (Fig. 7d-e). Using *in silico* perturbation analysis to inform how altered *Foxn1* expression could affect aaTEC cell fate, we observed skewing of the predicted trajectory away from aaTEC when simulating elevated *Foxn1* transcription. By contrast, downregulation of *Foxn1* was predicted to increase aaTEC differentiation (Fig. 7g). Consistent with this prediction, we found markedly accelerated aaTEC emergence in scRNAseq analysis of mice expressing a hypomorphic allele of *Foxn1* that leads to early loss of expression and premature involution (PMID: 18978204) (Fig. 7h).

DATA TO ADDRESS POINTS 1 AND 2:

1. Expression of lymphotoxin and RANKL receptors as well as *NFkb* targets (Fig. 4f)
2. aaTEC expression of *Foxn1* (flow cytometry) (Fig. 7a)
3. aaTEC expression of *Foxn1* (flow cytometry) (Fig. 7b-c)
4. Bioinformatic relationship between *Foxn1* expression and differentiation toward aaTEC fate (Fig. 7d-e)
5. Bioinformatic perturbation and activation of *Foxn1* pathway as it relates to aaTEC differentiation (Fig. 7g)
6. Identification of aaTEC expansion in *Foxn1^{z/z}* mice (scRNAseq) (Fig. 7h)

3. The authors showed that aaTECs can be defined as Ly51- UEA- double negative TEC (Fig. 1 i), but it is not enough. The authors should identify molecular markers that are specifically expressed in aaTECs but not in cTEC and mTEC, and confirm this by flow cytometric analysis. This experiment is necessary to enable other researchers to detect aaTECs. If possible, it would be better to perform immunohistochemical staining to confirm the marker is specifically expressed in aaTECs.

We appreciate that a limitation of our initial submission was the lack of prospective markers to identify aaTECs. To address this concern, we have now reanalyzed our sequencing dataset to identify a list of novel markers that can be used to identify each aaTEC subset. Amongst these, we identified Claudin-3 and podoplanin as putative markers that could be used to identify aaTEC1 and aaTEC2, respectively (Fig. 1i, S5e). We validated Claudin-3 expression by aaTEC by gating on the Ly51-UEA1- DN-TEC fraction of EpCAM+MHCII+ TECs. This enabled exclusion of aaTEC2 cells (that do not express EpCAM) and those mTECs that express *Cldn3* (Fig 1i-k, S5e-f). Further confirmation from imaging studies of the high density aaTEC structures identified in sections of thymic tissue from *Foxn1^{nTnG}* mice. Prominent Claudin-3 expression was observed throughout these high-density TEC regions only detected in aged mice, in contrast to sparse staining observed within the medulla proper (Fig. 2e-f). Moreover, using our transcriptome dataset, we identified podoplanin as a putative marker of aaTEC2 (Fig. 1i, S5e). Using flow cytometry we could identify a population of PDPN+PDGFRa- cells within the EpCAM-MHCII+ fraction that allowed for exclusion of PDPN+ fibroblasts and mesothelial cells that do not express MHCII (Fig 1i-k, S5e-f, Fig 3c). This strategy was also confirmed by flow cytometry in *Foxn1^{nTnG}* mice where EpCAM-GFP+ cells can be readily identified as aaTEC2. Unfortunately, PDPN could not be used effectively in imaging studies due to expression on other cells. Nevertheless, these new data provide comprehensive validation of new markers of aaTEC1 and aaTEC2 that will enable prospective analysis of these novel

populations by the scientific community.

DATA TO ADDRESS POINT 3:

1. Bioinformatic identification of putative aaTEC markers (Fig. 1i, S5e)
 2. aaTEC gating and identification in WT mice (Fig. 1j-k, S5f)
 3. Validation of aaTEC markers by flow cytometry in Foxn1nTnG mice (Fig. 3a-d)
 4. Validation of Claudin-3 in aaTEC regions by imaging of Foxn1nTnG mice (Fig. 2e-f)
4. The authors claim that aaTEC form non-functional microenvironment. However, I am not convinced that peri-medullary region formed by aaTECs is non-functional since the authors do not perform analysis using mice lacking aaTECs. The authors should hence tone down this statement and improve this subject.

We appreciate the reviewer's view and have attempted to tone down our language to address this concern. We should clarify that the function we are referring to is supporting thymocyte differentiation. For instance, in the abstract we have removed reference to an accretion of non-functional thymic tissue and instead refer to these regions as non-productive thymic tissue). Furthermore, we now provide additional evidence for this view showing that, in both mouse and human, aaTEC structures exclude developing thymocytes (Fig. 3h-i, S6b). A new movie showing a 3D reconstruction from the Foxn1nTnG reporter mouse clearly demonstrates this exclusion (Movie S3), supporting our other imaging and spatial transcriptomic analyses. Furthermore, we also now provide evidence in aged human thymus that there are also epithelial-rich regions that also exclude thymocytes (Fig. 3j).

DATA TO ADDRESS POINT 4:

1. Pre-existing data from original submission aaTECs form regions that are devoid of thymocytes, assessed by imaging in Foxn1nTnG mice as well as spatial sequencing (Fig 3h-i, S6b)
2. Human thymus images from 50 year-old female showing epithelial-rich, thymocyte devoid regions (Fig. 3j).
3. Movie showing exclusion of tdTomato+ cells within aaTEC regions (Movie S3)

5. Do aaTECs express tissue restricted antigens? Are there possibilities that aaTECs function as source of self-antigens?

This is an excellent point. Although our original manuscript data suggested that aaTECs do not express AIRE (Figs. 1c, 4c, S9c-d), we did not perform a comprehensive analysis of TRA expression. Therefore, in our revised manuscript we have now included an analysis of AIRE- and FEZF2-driven TRA expression (adapted from PMID: 25224068 and 35061506) across TEC subsets (Figs. 4g, S10). This analysis showed that AIRE- and Fezf2-driven TRA expression was largely restricted to AIRE-expressing mTEC2 and mTECprol (Figs. 4g, S10). Furthermore, our original transcriptome data strongly suggested aaTEC are transcriptionally distinct from known mimetic cell populations (Fig. 1c-d). However, in new flow cytometric analysis we could more definitively demonstrate that there is little to no overlap between aaTEC and prominent mimetic cells, including tuft cells, M-cells and comeocytes (Fig. S9a-b).

DATA TO ADDRESS POINT 5:

1. Pre-existing data from original submission showing lack of Aire expression in aaTEC (Fig. 1c, S9c-d)
2. New bioinformatic reanalysis to show expression of select TRAs (Aire-dependent, Aire-independent/mimetic) across TEC populations (also highlight expression of Aire and Fezf2) (Fig. 4G, S10)
3. New flow cytometric analysis showing negligible mimetic cell overlap with aaTEC (Fig. S9a-b)

FIGURE ON NEXT PAGE>>>

6. The authors clearly showed that aaTECs are originated from conventional TECs using Foxn1 nTnG mice. However, it remains unclear which TEC subset(s) give rise to aaTECs. Could you predict progenitor cells of aaTECs using scRNA-seq data?

We agree with the reviewer that this is an important issue and we have first re-analyzed our sequencing data to show RNA velocity on all cells, including mimetic cells (Fig. 3f). Next, using recent vector field modelling approaches of RNA velocity data using Dynamo, we could show aaTECs derive from mTEC1 and a TEC population enriched for a signature associated with a recently described early progenitor (earlyprog) (PMID: 35614226) (Fig. 3f-g, Movie S2a-b). We have updated and clarified our language in the text to describe these new insights and highlight the lineage relationships between aaTECs and mTEC1/earlyprog and other TECs. In addition to

the description of earlyprog, recent evidence has also emerged (both bioinformatic and experimental) that CCL21-expressing mTEC1 can give rise to other mTECs including mimetic cells and Aire expressing mTEC2 (PMID 38466627, 3280480, 31657037). However, although our bioinformatic analysis suggests that aaTEC are derived from mTEC1 and an earlyprog, more formal lineage tracing studies will need to be performed to formally demonstrate this lineage relationship. Unfortunately, the tools do not yet exist to study this phenomenon (such as CCL21-creERT2 which would allow for temporal control of lineage tracing in aged mice).

DATA TO ADDRESS POINT 6:

1. New bioinformatic analysis of RNA velocity showing all populations including mimetic cells (Fig. 3f)
2. New bioinformatic analysis using RNA velocity modeling techniques to show that just mTEC1 and earlyprog are predicted to be contributing to aaTEC formation directly (Fig. 3f-g, Movie S2a)

7. The authors should show the age-related and damage-induced changes in cTECs. How does cTEC signature genes change with age or damage induction (e.g. Foxn1, Psmb11, Dll4, Ly51, CD205, IL-7, CCL25, Prss16, etc.)? If there is a change, could it explain the reason why thymic regeneration is delayed in old mice?

We strongly agree that cTECs are critical for an effective regenerative response and have previously demonstrated their importance as a target for regenerative factors like BMP4 (via induction of Foxn1 and DLL4). In the current manuscript we present evidence that, in addition to the decreased regenerative response in the production of factors such as FGFs and BMPs with age, one of the mechanisms by which aaTEC may prevent regeneration is by stealing factors from cTECs. To address this point, we performed analysis of gene expression changes in cTECs with age. We did not find changes in the transcription of key genes such as Foxn1, Dll4, Psmb11, Prss16, Ly75 or Enpep (Fig. S8a and not shown). However, this likely reflects a technical limitation as the number of cTECs captured through single cell sequencing of the whole CD45- compartment was low, therefore limiting the statistical power of our analysis of this population. cTECs represent a larger fraction of the epithelial compartment after damage, when we also observe a robust increase in genes such as Dll4, Cxcl12 and Ccl25 in young mice. This upregulation was abrogated in old mice (Fig. 6a-b). Within this specific cTEC analysis we have also included expression of key functional genes not yet implicated in regeneration such as Psmb11, Prss16, Cd205 and Ly51 (Figs. 6b).

DATA TO ADDRESS POINT 7:

1. Added cTEC expression of key factors with age (Fig. S8a).
2. Expression of key cTEC associated genes during the regenerative response (Fig. 6a-b).
3. Clarification of discussion to reflect that cTECs are important for regeneration and the muted response with age contributes to poor regeneration.

8. The authors have previously reported that BMP4 produced by endothelial cells is critical for thymic regeneration. Which EC subset produce BMP4? Does BMP4 expression increase after irradiation?

Our single cell sequencing data of whole non-hematopoietic thymic stroma has allowed the first in-depth characterization of the thymic endothelial cell (EC) compartment. We have identified three main cell subsets within our data: arterial, capillary and venous ECs (Fig. 1c-d, S4c). Using this approach, we could show that Bmp4 is expressed highest by venous ECs and to a lesser degree by arterial ECs (Fig. 1c, S4c). Interestingly, Bmp4 expression correlates with Vwf and Vcam1, higher expression of which represents large EC vessels. Venous ECs also express high levels of P-selectin and low levels of Ly6c1 indicative of thymic portal ECs, inferring that thymic portal ECs are the main producers of Bmp4 (Fig. 1c-d, S4c). In the damage setting, we do indeed find an increase of Bmp4 within venous ECs at days 4 and 7 after TBI, but only in 2mo mice as this is an effect that is absent in 18mo vECs (Fig. 6a).

DATA TO ADDRESS POINT 8:

1. Genes defining EC subsets, leiden clustering for EC compartment, and signatures overlaid onto EC UMAP (Fig. 1c-d, S4c)
2. Changes in gene expression after damage in young and aged mice showing broad upregulation of a regenerative program (including BMP4 by vECs) in young but not old mice (Fig. 6a)

9. The Discussion is very tightly written. I feel that the authors should provide discussion and perspective related to not only aaTEC but also fibroblast, EC, and cTEC.

We appreciate that we were a little too concise in our original submission and have attempted to expand our discussion to address new data showing the involvement of Foxn1 in aaTEC emergence, but have also attempted to do a better job at incorporating some additional discussion of other cell populations, such as endothelial cells and fibroblasts. Specifically, we have attempted to make clearer that we do feel that changes in the fibroblast compartment, and especially acquisition of an "inflammaging" or SASP-like phenotype could play a role in driving aaTEC emergence. Specifically, we discuss:

“However, the molecular drivers of aaTEC likely also involve additional factors. For instance, we found considerable changes in the stromal microenvironment beyond TECs, including significant changes in fibroblasts that mirror many of the main hallmarks of aging, reflected by a loss in mitochondrial, metabolic, and proteostasis programs, and increase in pathways associated with inflammaging or senescence-associated secretory phenotype (SASP)¹. This latter finding likely also reflects the broader role of inflammaging and SASP in driving EMT in aged tissues. This link is especially notable given that one of the main features of the aging thymus is the replacement of functional tissue with fat, which can directly drive loss of thymic function. Furthermore, there is evidence that the emergence of fat in the human thymus may be triggered by EMT. These data also support recent observations that age-associated changes in tissues are organ- and cell-lineage specific (e.g. senescence-associated inflammaging impacting on muscle regeneration). Our findings are therefore consistent with the concept that aaTEC represent a thymus-specific manifestation of these programs, and that age-related changes in other cells of the thymic microenvironment, in particular fibroblasts, could also contribute to aaTEC emergence.”

We go on to mention in our concluding paragraph:

“These features seem to be at least partially driven by changes within the fibroblast compartment and, in particular, their upregulation of genes associated with inflammaging/SASP.”

10. The authors should describe the limitation of this study in the Discussion.

We have incorporated within the discussion section text describing some of the limitations, chief among them that lack of a direct way to deplete these cells we have thus far been unable to conclusively show the impact on aging and regeneration. Specifically, we discuss:

“without a direct way to deplete these cells (or block their emergence to begin with) further studies will be needed to elucidate the specific contributions of aaTEC in thymic aging and responses after damage.”

Minor concerns

11. Typos : Lines 212, 298, 404, 405.

We appreciate the diligence of the reviewer in picking up these errors and have amended them in the text. It appears that in trying to find and replace the word "table" to refer instead to Data Sâ several instances of the word "notable" were inadvertently replaced with Data S causing a typo of noData S!

Reviewer #2

The authors sought to identify changes in the murine thymic stromal compartment associated with age and injury. They used single-cell RNA sequencing to identify two populations that accumulate in involuted thymic tissue and used immunostaining to demonstrate the presence of areas dense in TECs but lacking thymocytes in older thymus. They also present bioinformatic data suggesting that age-associated TECs perturbs normal thymic function by acting as sinks for TEC growth factors. The work is interesting and potentially relevant to human thymic involution since TECs lacking characteristics of cTECs and mTECs have been identified in adult thymic tissue. However, the authors need to address a number of concerns to strengthen their conclusions.

We appreciate the general enthusiasm for our findings and have attempted to address all concerns below. In addition, although there are many significant changes within the manuscript text itself, we have outlined changes in blue.

Major concerns:

Although the presence of aaTECs is demonstrated by transcriptomics, the lack of markers to clearly identify them by flow cytometry or histology is a major caveat. This would be particularly important when quantifying changes in subsets of cells using flow cytometry. It is indeed not

clear which subsets are included when gating on UEA-1 negative cells. If such populations include other cells such as mimetic cells, the data might represent shifts in these other populations instead of aaTECs.

We agree that the Ly51-UEA-1- (DN) TEC gate is an imperfect approach for identifying aaTECs and appreciate the enthusiasm for prospective markers to identify these populations. In our revised manuscript we validated new markers that can be used to identify aaTEC1 and aaTEC2 using imaging and flow cytometry.

As described above to address Reviewer 1, to address this concern we have now reanalyzed our sequencing dataset to identify a list of novel markers that can be used to identify each aaTEC subset. We identified Claudin-3 and podoplanin as putative markers that could be used to identify aaTEC1 and aaTEC2, respectively (Fig. 1i, S5e). Although claudin-3 was not exclusively expressed by aaTECs, and was also expressed by a small subset of mTECs, by excluding expression of Ly51 and UEA1 to exclude mature cTECs and mTECs (as in the double negative TEC gate), allowed for the discrimination of an aaTEC1 population (Fig 1i-k, S5e-f). This gating strategy was confirmed by flow cytometry in Foxn1nTnG mice which showed that claudin-3 was almost exclusively expressed on aaTECs in comparison with mTECs (Fig. 3a-d). Importantly, in these same Foxn1nTnG mice we could also perform imaging studies that showed prominent Claudin-3 expression on high-density TEC regions of the aged thymus with only minimal staining within the medulla proper (Fig. 2e-f).

Using our transcriptome dataset, we identified podoplanin as a putative marker that could identify aaTEC2 (Fig. 1i, S5e). Using flow cytometry we could identify a population of PDPN+PDGFRA- cells within the EpcAM-MHCII+ fraction that allowed for exclusion of PDPN+ fibroblasts (which was only a minor population, Fig 3c) and mesothelial cells that do not express MHCII (Fig 1i-k, S5e-f). This strategy was also confirmed by flow cytometry in Foxn1nTnG mice where EpCAM-GFP+ cells can be readily identified as aaTEC2, however, unfortunately PDPN could not be used effectively in imaging studies due to expression on other cells. With respect to mimetic cells, we could also show that aaTEC regions did not express markers for Tuft cells or M-cells by imaging. In our revised manuscript, we add a flow cytometric survey of the DN-TEC compartment (UEA1-Ly51-EpCAM+GFP+ in Foxn1nTnG mice) and show that this gate does not include substantive Tuft, corneocyte or M-cell thymic mimetic cells (Fig. S9b-c).

DATA:

1. Bioinformatic identification of putative aaTEC markers (Fig. 1i, S5e)
2. Validation by flow cytometry of new aaTEC1 and aaTEC2 markers (Fig. 1j-k, 3a-d, S5f).
3. Corresponding imaging of putative markers in young and aged tissue as well as nTnG mice (Fig. 2e-f).
4. Exclusion of mimetic cells in aaTEC identification (Fig. 9b-c).

The idea that aaTECs are end-stage populations and not precursors blocked in their differentiation capacity would be greatly strengthened by lineage tracing experiments. This is a very good point and perhaps we were too emphatic in our conclusion that aaTEC are end-stage cells. We have revised our bioinformatic analysis, including recent vector field modelling approaches of RNA velocity data using Dynamo. This approach suggests that aaTEC stem from mTEC1 and a TEC population enriched for a signature associated with a recently described early progenitor markers (earlyprog) (PMID: 35614226), but that do not differentiate further from there (Fig. 3f-g, Movie S2a-b). However, it is also entirely possible that instead of being end-stage cells, aaTECs could also be stalled progenitor cells. Therefore, we have revised our manuscript to advance both hypotheses (i.e. that aaTEC may be end-stage or that they are arrested progenitors) and weigh our current evidence for both. Specifically, we highlight our new data demonstrating that reduced FOXN1 activity can drive aaTEC formation, along with our RNA velocity data showing that aaTEC are derived from mTEC and earlyprog, without evidence of downstream products. On the other hand, expression of Claudin-3 and Plet1 (which have been previously implicated as markers for TEC precursors) might suggest that aaTEC retain some features of progenitors. These alternate interpretations and hypotheses are addressed in the discussion. We agree that a genetic fate-mapping approach would be an ideal test of whether aaTEC are capable of differentiating any further. However, the creation and aging of a suitable mouse model (e.g. an aaTEC-specific promoter driving Cre-mediated activation of a reporter allele) to address this point is not feasible within the time constraints.

NEW DATA:

1. RNA velocity analysis including Dynamo which highlights that aaTEC do not become other cells (Fig. 3f-g, Movie S2a-b).

The hypothesis that emergence of aaTEC perturbs the network of growth factors supporting stromal cell function and thymocyte differentiation is based on bioinformatics data only and

should be confirmed using biological experiments.

We acknowledge that, although our transcriptomic data suggest that aaTEC draw tonic signals from conventional TECs, we did not formally show this in our original manuscript submission. To address this concern we have assayed signaling function in stromal subsets purified from aged mice, including TECs and aaTECs, by measuring their response to KGF stimulation (Fig. 6e). We detected AKT phosphorylation in cTECs and mTECs, but not in the negative control, fibroblasts (which do not express the receptor) (Fig. 6c). Both aaTEC1 and aaTEC2 express KGF receptor components and robustly responded to KGF stimulation in increasing AKT phosphorylation (Fig. 6e, 6c), indicating that they retain the capacity to receive and compete for this growth signal, consistent with our hypothesis.

DATA:

1. Stimulation of freshly isolated TECs with KGF, which showed increased AKT phosphorylation in response to KGF stimulation (Fig. 6e).

Minor concerns:

All minor points have been addressed in this revision by better clarifying our statements and/or adding additional context.

Fig. 1H - what is non-mTEC? DN-TEC+cTECs? Please label clearly. What is the rationale for using CD104? Which subset of cells is it marking?

Fig. 1i - What is the gating strategy? Do the quantification of DN-TECs include tuft cells and other mimetic cells?

We apologize for the oversight in not providing the rationale for our gating strategy. We largely adapted the strategy described by Bornstein et al (2018). We have now tried to clarify this strategy in the text of the manuscript and in the figure legend. Specifically, for (new) Fig. 1g, based on a CD45-EpCAM⁺ parent gate, tuft cells were identified by expression of L1CAM, then all other TECs were assessed for expression of conventional TEC markers UEA1 and Ly51. Within the UEA1^{hi}Ly51^{lo} mTEC population, CD104⁺MHCII^{lo} cells were identified as mTEC1. Cells that were deemed as non-mTEC1 were then further fractionated based on MHCII and Ly6D (n=10/age).

While the DN-TEC gate in Fig. 1h does not specifically exclude mimetic cells, we have now included new analysis to demonstrate that this population does not include the major mimetic cell types. In our original manuscript we showed that high-density aaTEC regions did not include DCLK1⁺ tuft UEA1^{high} M-cell mimetics (Fig. S9a). Our new data demonstrates that there is little to no incorporation of Tuft, M-cells or corneocyte mimetic cells in the DN-TEC gate by flow cytometric analyses (Fig. S9a-b). In combination with our new validated markers of aaTEC1 (Claudin-3) and aaTEC2 (PDPN), these data establish a definitive phenotypic characterization of these novel populations that will inform further research.

Fig. 2a - What is the scale representing?

The scale in the revised manuscript Fig. 3e represents the log-transformed average transcription of the td-Tomato+WRPE element. We have updated the figure and associated legend to include this information.

Fig. 2b - What is the gating strategy? Do the quantification of DN-TECs include tuft cells and other mimetic cells?

In (new) Fig. 3a, which shows the identification of aaTEC in Foxn1^{nTnG} mice, the gating strategy has now been more comprehensively outlined in the figure legend. Specifically, we first gate on tdTomato-GFP⁺ cells, which will represent thymic cells that express or have expressed FOXN1 and thus can be broadly characterized as TECs. We then show EpCAM expression which distinguished a novel EpCAM⁻ population only in the involuted thymus (Fig. 3a). Gating on EPCAM⁺ cells, the conventional TEC markers UEA1 and Ly51 discriminate a DN population only in aged mice that includes aaTEC, some of which express Claudin-3 (Fig. 3b). In contrast, cells in the tdTom-GFP⁺EpCAM⁻ group can be identified as aaTEC2, which we have now further characterized based on PDPN expression (Fig. 3c).

The sentence "clear lineage trajectory stemming from the mTEC^{prol} and mTEC1 populations and continuing into the more differentiated mTEC lineages (mTEC2 and mTEC3) in young mice" is not accurate since mTEC^{prol} give rise to mTEC1 or mTEC2/mTEC3, not mTEC1 continuing to differentiate into mTEC2/mTEC3.

We apologize that the language we used to describe our RNA velocity data was not precise. We have updated the text to clarify and to also reflect the new RNA velocity analysis we performed for this revision. Specifically, using a vector field modelling approach of RNA velocity data using Dynamo, we show that mTEC1 and a TEC population enriched for early progenitor markers

(earlyprog) are predicted as the main immediate precursors of the aaTEC lineage (Fig. 3f-g and Movie S2).

Fig. 2c - why not include some of the mimetic cells (goblet, M cells) in the RNA velocity?

We have re-analyzed our sequencing data to show RNA velocity on all cells (including mimetic cells). This analysis shows that mimetic cells do not give rise to aaTECs (Fig. 3f and Movie S2).

Fig. 3B - how is the medulla marked?

In Fig. 3b (what is now Fig. 2b) the medulla surface was defined on the basis of the frequency of GFP+ cells and tdTomato expression (with high tdTom correlating with medullary regions). This assignment was confirmed by Krt14 staining and confocal analysis.

Fig. 3D - Krt5, Krt8, and Krt14 are not very specific to any TEC subsets and are definitely not specific to aaTECs. Based on the transcriptome data, was there a cytokeratin that is more specific to aaTECs? Co-staining of GFP and some of these cytokeratins with UEA-1 would help compare the data with the flow cytometry results.

We agree with the reviewer that, alone, Krt5, Krt8 or Krt14 were not specific for the aaTEC populations. However, in this instance we highlight that the imaging correlated with the a distinctive collective expression profile for these three keratin subunits (i.e. K5+/-; K8+; K14-). In our revised manuscript, we present more refined analysis of these populations that validates new markers for aaTEC, including Claudin-3 and podoplanin, in combination with other canonical markers to resolve both aaTEC subsets.

Fig. 3E - please include scale for expression levels.

We apologize for the oversight and have now included scales for expression levels of the markers in (revised manuscript) Fig. 2f.

Figure 4 - Where are aaTEC2 located relative to HD-TEC regions? Co-labeling of GFP and vimentin (or other marker of EMT) would be very informative.

We agree that finding markers to better identify the location of aaTEC2 would be desirable and complement the spatial transcriptomic data suggesting they are co-localized with aaTEC1. Unfortunately, we have been unable thus far to accomplish this. While we have found that podoplanin can be used (in conjunction with other markers) to identify aaTEC2 by flow cytometry, this has not proved useful as a discriminant in imaging studies due to expression by other thymic stromal cells. Likewise, looking for GFP+EpCAM- cells in Foxn1nTnG mice has not proved a robust method to identify these cells. Finally, given that aaTEC2 only seem to represent a partial EMT phenotype, we have found that they do not express sufficient vimentin to enable their imaging with this marker. We suggest that our initial description of this population, with detailed transcriptomic and phenotypic profiling, will directly inform future efforts to spatially localize aaTEC2.

Fig. 5B - the way the graph is scaled is a bit misleading. The x axis should be scaled to better reflect time points and it might be better to use a broken axis instead of log scale for the y axis. Unfortunately given the wide range of thymic cellularity between baseline, early after damage and late after damage, we have found that the most effective way of showing differences across the timescale is to use a log-scale rather than a broken axis or other approach. However, we did notice that there was a mistake in our original submission and the graph itself was the linear version with the scale marked incorrectly with the log scale. We have now replaced the graph in the panel with the correct log-scale version and feel this highlights the early and late differences effectively (new Fig. 5b).

Fig. 5D - What is the gating strategy for each population? Same as Fig. 1H? An example of how the recovery index is calculated would help understand the data for this panel.

We have now added a comprehensive description in the figure legend about how cells were gated and how the recovery index was calculated. Specifically, depletion and recovery of indicated populations were quantified by flow cytometry over the first 7 days after TBI and area under the curve was calculated (associated with Fig. S8b). Recovery index was generated by calculating the ratio of aged to young AUC for each indicated population.

Fig. 5E - legend says 12mo while figure is labelled as 18mo. Which one is it? How are aaTECs marked?

Fig. 5F - What is the age of tissue and what time point after TBI? How are the cortex and medulla marked? And how are the subsets of cells identified?

We apologize for the confusion. The data in Fig. 5E-F was from 12mo mice untreated or 28 days after a sublethal dose of TBI (550cGy). We have amended the figure and legend to reflect the correct age of mice and treatment. The medulla is marked by tdTomato expression and confirmed by K14. aaTEC is marked by the density of GFP+ cells (also confirmed by K14- and

tdT-). Cortex volume is the volume of lobe minus medulla and aaTEC.

The sentence “RNA velocity analysis implied that, in 2mo mice, the reemergence of differentiated mTEC populations stemmed largely from the mTEC1 and mTECprol populations (Fig. 5k)” is misleading since it seems that differentiated mTECs (assuming we are talking about mTEC2/mTEC3) are coming from mTECprol, not mTEC1. The statement about mTECprol potentially giving rise to aaTECs is not supported by the data.

We apologize for the confusing way we described this analysis and have attempted to rectify this in the manuscript. Specifically, we now highlight that RNA velocity analysis implied that, in 2mo mice, the reemergence of differentiated conventional mTEC populations (mTEC2, mimetic cells) after damage stemmed largely from the mTECprol populations (Fig. 5k). In contrast, after damage in 18mo mice there was a skewing in this inferred direction and rather than mTECprol driving differentiation toward conventional differentiated mTECs, mTEC1 and earlyprog drove differentiation towards aaTEC1 and aaTEC2 (Fig. 5k).

Fig. 6G – What does mTEC diff mean?

We agree that this terminology was not easily understood or described. We used it to simplify some of the figures but have now removed all mention of mTECdiff. In the one place that still contains a collapsed population to include both mTECprol and mTEC2 (Fig. 4h) we have specifically referred to them that way.

How do aaTECs relate to previously described TEC subsets such as intertypical TECs, which have been shown to be progenitor-like and also increase with age (Baran-Gale et al. eLife 2020)? How do aaTECs compare to the immature TECs populations described by Bautista et al. (Nature Communications 2021)? For example, do aaTECs express some of the markers enriched in the immature population that accumulates in older human tissues?

To address the reviewer’s query, we want to first draw their attention to the revised manuscript and rebuttal Fig. S3d, where we overlay TEC subset gene signatures from all current literature, including Baran-Gale et al. eLife 2020, on our own data. Close inspection of these signatures (including the intertypical TEC signature) shows that aaTEC1 and aaTEC2 subsets do not map to any of the currently known TEC populations (Fig. S4d). To further highlight the unique identity of aaTECs, we have also integrated all publicly available sequencing data of the thymic epithelium, including ours, into a master dataset that revealed aaTEC being discrete among the TEC subsets from other datasets (Fig. 1a), with the exception of one (Rota et al., 2021, PMID: 34860543). We believe the main reason for the lack of aaTEC in other datasets (especially in the case of Baran-Gale et al. which does explore thymus aging) is the sorting strategies used to isolate TEC in the first place. Studying the thymic epithelium normally entails sorting on a combination of EPCAM and cortical or medullary markers. Purification of EPCAM+ cells will omit aaTEC2 which lack expression for EPCAM. Similarly, aaTEC1 lack expression of conventional TEC markers such as Ly51 and UEA1, which will exclude aaTEC1 if these additional markers are used, such as in Baran-Gale et al. Consistent with this, the only publication (Rota et al., 2021) that broadly sorted on CD45- cells managed to identify a small fraction of cells that had a transcriptional signature resembling aaTEC1 and aaTEC2 (Fig. S4d). Notably, aaTEC was most prominent in Foxn1 mutant mice, in agreement with our new data showing that Foxn1 impairment leads to accelerated emergence of aaTECs (Fig. 7). In the Rota et al. manuscript, the small numbers of aaTEC1 and aaTEC2 resulted in the authors combining these data under the intertypical TECs that were in close proximity.

What is the evidence that aaTECs start emerging at 6mo? Throughout the manuscript, these subsets are identified by transcriptomics from 2mo and 18mo mice while the 6mo data is from flow cytometry and might include other.

We apologize for this phrasing; we did not mean to imply a sharp threshold at 6 months for when they emerge. Our data suggests a gradual accumulation of these cells but we have not performed an in-depth kinetic characterization in the early stages of life to determine the exact timing of their emergence. We have amended our language to reflect this uncertainty.

Why use only females for the analysis of aged mice? Are aaTECs also found in males?

The original transcriptome dataset was derived from only female mice; however, we also have extensive data showing the presence of aaTECs in both male and female mice. In Fig. 1k, where we calculate the number of aaTEC1 and aaTEC2, we have now highlighted which mice were male and which were female in each of the age groups. This clearly shows that aaTECs are present in both males and females. Furthermore, when assessing emergence of aaTEC in Foxn1^{z/z} mice we have sequencing data from both male and female mice. Although in the main figure these have been combined to highlight the broad differences across age and genotype, we have now included an expanded pilot in Fig. S14 that highlights the representation of different subsets in either male or female mice. We have also now clearly highlighted in the figure legends which sex was used for which experiment and throughout the manuscript we have a mix of both male and female mice.

1 Schaum, N. et al. Ageing hallmarks exhibit organ-specific temporal signatures. Nature 583, 596-602 (2020). <https://doi.org/10.1038/s41586-020-2499-y>

Version 1:

Decision Letter:

RE: MS# NI-A35908A

Dear Dr. Dudakov,

The comments from the referees on your Review entitled "Age-related epithelial defects limit thymic function and regeneration" are now in. We are happy to inform you that the review is almost ready for publication in Nature Immunology's Focus on [FILL IN TOPIC], but first it must be revised in response to the referees' comments and our editorial requirements.

Please note my suggestions in the manuscript. Please note in your revision letter the pages and lines where the changes can be found in the revision (highlighting the changes in a copy of the manuscript is also extremely helpful and helps speed acceptance).

Before sending us your revised manuscript, care should be taken to adjust the format to fit the style of Nature Immunology. Please note the enclosed guidelines, also available on our web site at <http://www.nature.com/ni/authors/submit/index.html>.

When you are ready to submit your revised Review and figures, please use the URL below.

Link Redacted

We hope to receive the revised paper in ten days. Please let us know if circumstances will delay submission beyond this time. If you have any questions please do not hesitate to contact me.

If any of the display items in your manuscript (for example - figures, tables, images, videos or text boxes) include images that are the same as (or are adaptations of) images that have previously been published elsewhere, please fill in the Third Part Rights Table (<http://www.nature.com/documents/thirdpartyrights-table.doc>), and return to us when you resubmit your revised manuscript. This information will enable us to obtain the necessary rights to re-use such previously published material. If we are unable to obtain the necessary rights to use or adapt any of the material that you wish to use in the display items in your manuscript, we will contact you to discuss the sourcing of alternative images.

In recognition of the time and expertise our reviewers provide to Nature Immunology's editorial process, we would like to formally acknowledge their contribution to the external peer review of your manuscript entitled "Age-related epithelial defects limit thymic function and regeneration". For those reviewers who give their assent, we will be publishing their names alongside the published article.

Please note that you and any of your coauthors will be able to order reprints and single copies of the issue containing your article through Springer Nature's reprint website, which is located at <http://www.nature.com/reprints/index.html>. Please let your coauthors and your institutions' public affairs office know that they are also welcome to order reprints by this method.

Nature Immunology has now transitioned to a unified Rights Collection system which will allow our Author Services team to quickly and easily collect the rights and permissions required to publish your work. Once your paper is accepted, you will receive an email in approximately 10 business days providing you with a link to complete the grant of rights. If you choose to publish Open Access, our Author Services team will also be in touch at that time regarding any additional information that may be required to arrange payment for your article.

Please note that you will not receive your proofs until the publishing agreement has been received through our system. If you have any questions please contact ASJournals@springernature.com.

ORCID

Nature Immunology is committed to improving transparency in authorship. As part of our efforts in this direction, we are now requesting that all authors identified as 'corresponding author' create and link their Open Researcher and Contributor Identifier (ORCID) with their account on the Manuscript Tracking System (MTS) prior to acceptance. ORCID helps the scientific community achieve unambiguous attribution of all scholarly contributions. For more information please visit <http://www.springernature.com/orcid>

For all corresponding authors listed on the manuscript, please follow the instructions in the link below to link your ORCID to your account on our MTS before submitting the final version of the manuscript. If you do not yet have an ORCID you will be able to create one in minutes.

IMPORTANT: All authors identified as 'corresponding author' on the manuscript must follow these instructions. Non-corresponding authors do not have to link their ORCIDs but are encouraged to do so. Please note that it will not be possible to add/modify ORCIDs at proof. Thus, if they wish to have their ORCID added to the paper they must also follow the above procedure prior to acceptance.

To support ORCID's aims, we only allow a single ORCID identifier to be attached to one account. If you have any issues attaching an ORCID identifier to your MTS account, please contact the [Platform Support Helpdesk](http://platformsupport.nature.com/).

We hope that you will support this initiative and supply the required information. Should you have any query or comments, please do not hesitate to contact me.

Sincerely,
Stephanie Houston, PhD
Senior Editor
Nature Immunology

Reviewers' comments:

Reviewer #1 (Remarks to the Author):

The authors appropriately addressed my concerns.

Reviewer #2 (Remarks to the Author):

Thank you for the detailed responses and the additional clarifications added to the manuscript. While my previous concerns were addressed in the revised manuscript, I have minor suggestions to improve Figure 3k: it would be helpful to show expression of markers characteristics of TEC subsets to help the readers understand which subsets have highest expression of aaTEC signatures. For example, could the panels showing expression of cTEC_signature, mTEC1_signature, mTECprol_signature, and mTEC2_signature be moved to the main figure? I also suggest showing expression of markers of the keratinocyte/corneocyte subset such as IVL or KRT1 in Figure S7. Finally, it would be interesting to show the expression of the newly identified markers CLDN3 and PDPN in the human TEC dataset.

Author Rebuttal letter:

Reviewer #1 (Remarks to the Author):

The authors appropriately addressed my concerns.

We appreciate that we have sufficiently answered the reviewers concerns and thank them for their work on improving our manuscript.

Reviewer #2 (Remarks to the Author):

Thank you for the detailed responses and the additional clarifications added to the manuscript. While my previous concerns were addressed in the revised manuscript, I have minor suggestions to improve Figure 3k: it would be helpful to show expression of markers characteristics of TEC subsets to help the readers understand which subsets have highest expression of aaTEC signatures. For example, could the panels showing expression of cTEC_signature, mTEC1_signature, mTECprol_signature, and mTEC2_signature be moved to the main figure? I also suggest showing expression of markers of the keratinocyte/corneocyte subset such as IVL or KRT1 in Figure S7. Finally, it would be interesting to show the expression of the newly identified markers CLDN3 and PDPN in the human TEC dataset.

We thank the reviewer for these helpful suggestions and have amended the manuscript accordingly. Specifically, as suggested we have moved some of the main TEC signatures ((mTEC1, mTECprol, and mTEC2) into the main figure and included additional UMAPs with IVL, KRT1, CLDN3, and PDPN in the extended data figure (now Ext Fig. 5).

Version 2:

Decision Letter:

Our ref: NI-A35908B

13th May 2024

Dear Dr. Dudakov,

Thank you for submitting your revised manuscript "Age-related epithelial defects limit thymic function and regeneration" (NI-A35908B). We'll be happy in principle to publish it in Nature Immunology, pending minor revisions to comply with our editorial and formatting guidelines.

We will now perform detailed checks on your paper and will send you a checklist detailing our editorial and formatting requirements in about a week. Please do not upload the final materials and make any revisions until you receive this additional information from us.

If you had not uploaded a Word file for the current version of the manuscript, we will need one before beginning the editing process; please email that to immunology@us.nature.com at your earliest convenience.

Thank you again for your interest in Nature Immunology Please do not hesitate to contact me if you have any questions.

Sincerely,

Stephanie Houston, PhD
Senior Editor
Nature Immunology

Version 3:

Decision Letter:

In reply please quote: NI-A35908C

Dear Dr. Dudakov,

I am delighted to accept your manuscript entitled "Age-related epithelial defects limit thymic function and regeneration" for publication in an upcoming issue of Nature Immunology.

Over the next few weeks, your paper will be copyedited to ensure that it conforms to Nature Immunology style. Once your paper is typeset, you will receive an email with a link to choose the appropriate publishing options for your paper and our Author Services team will be in touch regarding any additional information that may be required.

Please note that *Nature Immunology* is a Transformative Journal (TJ). Authors may publish their research with us through the traditional subscription access route or make their paper immediately open access through payment of an article-processing charge (APC). Authors will not be required to make a final decision about access to their article until it has been accepted. Find out more about Transformative Journals.

Authors may need to take specific actions to achieve compliance with funder and institutional open access mandates. If your research is supported by a funder that requires immediate open access (e.g. according to 

Also, if you have any spectacular or outstanding figures or graphics associated with your manuscript - though not necessarily included with your submission - we'd be delighted to consider them as candidates for our cover. Simply send an electronic version (accompanied by a hard copy) to us with a possible cover caption enclosed.

If you have not already done so, we strongly recommend that you upload the step-by-step protocols used in this manuscript to protocols.io. protocols.io is an open online resource that allows researchers to share their detailed experimental know-how. All uploaded protocols are made freely available and are assigned DOIs for ease of citation. Protocols can be linked to any publications in which they are used and will be linked to from your article. You can also establish a dedicated workspace to collect all your lab Protocols. By uploading your Protocols to protocols.io, you are enabling researchers to more readily reproduce or adapt the methodology you use, as well as increasing the visibility of your protocols and papers. Upload your Protocols at <https://protocols.io>. Further information can be found at <https://www.protocols.io/help/publish-articles>.

Please note that we encourage the authors to self-archive their manuscript (the accepted version before copy editing) in their institutional repository, and in their funders' archives, six months after publication. Nature Portfolio recognizes the efforts of funding bodies to increase access of the research they fund, and strongly encourages authors to participate in such efforts. For information about our editorial policy, including license agreement and author copyright, please visit www.nature.com/ni/about/ed_policies/index.html

Sincerely,

Stephanie Houston, PhD
Senior Editor
Nature Immunology

Click here if you would like to recommend Nature Immunology to your librarian
<http://www.nature.com/subscriptions/recommend.html#forms>

** Visit the Springer Nature Editorial and Publishing website at http://editorial-jobs.springernature.com?utm_source=ejP_NImm_email&utm_medium=ejP_NImm_email&utm_campaign=ejP_NImm for more information about our career opportunities. If you have any questions please click [here](mailto:editorial.publishing.jobs@springernature.com).
